# Variable efficiency of nonsense-mediated mRNA decay across human tissues, tumors and individuals

Guillermo Palou-Márquez[1] and Fran Supek[1,2,3]*

*Correspondence:
fran.supek@bric.ku.dk

[1] Institute for Research in Biomedicine (IRB Barcelona), The Barcelona Institute for Science and Technology (BIST),  Barcelona 08028, Spain
[2] Biotech Research and Innovation Centre, Faculty of Health and Medical Sciences, University of Copenhagen, Copenhagen 2200, Denmark
[3] Catalan Institution for Research and Advanced Studies (ICREA), Barcelona, Spain

## Abstract

**Background:**  Nonsense-mediated mRNA decay (NMD) is a quality-control pathway that degrades mRNA bearing premature termination codons (PTCs) resulting from mutation or mis-splicing, and that additionally participates in gene regulation of unmutated transcripts. While NMD activity is known to differ between examples of PTCs, it is less well studied if human tissues differ in NMD activity, or if individuals differ.

**Results:**  We analyzed exomes and matched transcriptomes from Human tumors and healthy tissues to quantify individual-level NMD efficiency, and assess its variability between tissues, tumors, and individuals. This was done by monitoring mRNA levels of endogenous NMD target transcripts, and additionally supported by allele-specific expression of germline PTCs. Nervous system and reproductive system tissues have lower NMD efficiency than other tissues, such as the digestive tract. Next, there is systematic inter-individual variability in NMD efficiency, and we identify two underlying mechanisms. First, somatic copy number alterations can robustly associate with NMD efficiency, prominently the commonly-occurring gain at chromosome 1q that encompasses two core NMD genes: *SMG5* and *SMG7* and additional functionally interacting genes such as *PMF1* and *GON4L*. Second, deleterious germline variants in genes such as the *KDM6B* chromatin modifier can associate with higher or lower NMD efficiency in individuals. Variable NMD efficiency modulates positive selection upon somatic nonsense mutations in tumor suppressor genes, and is associated with cancer patient survival and immunotherapy responses.

**Conclusions:**  NMD efficiency is variable across human tissues, and it is additionally variable across individuals and tumors thereof due to germline and somatic genetic alterations.

## Background

Nonsense-mediated mRNA decay (NMD) is a quality-control mechanism that degrades mRNAs containing premature termination codons (PTCs), which can arise from nonsense mutations, frameshift mutations, or aberrant splicing events [1–3]. Traditionally recognized for its role in preventing the translation of defective transcripts that could lead to truncated proteins with potentially detrimental functions, NMD also plays a common role in the regulation of non-mutated, natural transcripts. NMD regulates approximately 10% of the normal transcriptome, impacting physiological gene expression [4–6]. This broader regulatory role is attributed to the presence of NMD-triggering features in many endogenous transcripts, even in the absence of PTCs [7]. An additional regulatory role of NMD stems from being coupled with alternative splicing that retains introns or skips exons and thus introduces NMD-triggering features as a means to regulate gene expression [7].

In mammals, the NMD pathway primarily distinguishes between PTCs and normal stop codons through a mechanism linked to the splicing process and translation. Based on various targeted experiments that yielded this mechanistic understanding, as well as on large-scale genomic analyses of PTCs in transcriptomes, genomic rules for NMD detection or evasion were identified [8–10]. Most prominently, the PTCs located in the last exon of a transcript escape NMD, as well as PTCs within the ∼50 nt upstream of the last splice site: the exon-junction complex (EJC) proteins deposited onto splice sites promote NMD, but only if the EJC was not displaced by the elongating ribosome. Start-proximal PTCs are also less likely to initiate NMD, likely in part through reinitiation of translation, and PTCs in long exons partially evade NMD through less clear mechanisms [8, 11]. These rules notwithstanding, the majority of PTCs do trigger NMD to some extent.

Furthermore, non-mutated endogenous NMD targets exhibit specific features that can trigger NMD [5, 6, 12–14]. For example, an upstream open reading frame (uORF) in the 5' untranslated region (UTR) can cause NMD activation [15, 16], mimicking a PTC with a downstream EJC deposited, and similarly so an intron within the 3' UTR [12, 17]. High GC content in 3'UTR may also be associated with increased NMD efficiency [18].

However, the known NMD features are not perfect predictors of observed NMD efficiency of PTC [11, 12]. It appears that additional features contribute to NMD variation, and these might not pertain only to genomic context differences between PTCs and/or transcripts, but were also proposed to stem from NMD action varying among individuals [19, 20]. Particular examples underlying this hypothesized inter-individual NMD variability were put forth, such as links with individual examples of germline variation, somatic alterations, or changes in expression of NMD factors [20–22]. Furthermore, though less extensively studied, variations in NMD efficiency have been observed between different tissues and cell lines in mice [23, 24] and in human cells [25, 26]. For example, considering the same PTCs across various Human tissues, 18% of variants exhibited heterogeneous effects across tissues and 8% were tissue-specific [27]. These variations in NMD efficiency between tissues or across individuals can lead to distinct pathologies in genetic diseases and affect therapeutic outcomes. This is illustrated in the case of two Duchenne muscular dystrophy patients with identical PTCs but differing disease severities [28] apparently associated with NMD efficiency. Another related case

involved two fetuses harboring the same PTC for Roberts syndrome [29]. Thus, particular examples suggest individuals can differ in NMD efficiency, but it is not clear what is the magnitude of this variation, its extent across populations and what are the common underlying mechanisms.

One genetic disease with many proposed links with NMD is cancer. NMD may exhibit either pro-tumor or tumor suppressor functions, as a function of cell type, the genetic context and tumor microenvironment [30]. The key NMD factor *UPF1* was reported as downregulated in various cancers[31–37], in comparison with matched normal tissue; however opposite trends were also noted [38, 39]. An interesting case of NMD in association with mutation burden was found in colorectal adenocarcinoma (CRAD), where tumors of patients with microsatellite-instability (MSI) hypermutation overexpressed essential NMD factor genes *UPF1/2* and *SMG1/6/7*, in contrast to microsatellite-stable (MSS) primary CRAD[40]. Moreover, inhibiting NMD impaired cell proliferation and tumor growth in MSI tumors specifically[40].

Approximately 30% of inherited diseases, as well as the majority of cancers (due to mutations in tumor suppressor genes (TSGs)), are caused by PTCs resulting from frameshift mutations or nonsense variants [41, 42], and thus may be modulated by NMD. NMD has been recognized as a significant factor influencing disease severity and clinical phenotype [11, 28, 43–45], and a population genomics analysis suggests that it can either exacerbate or alleviate disease phenotypes, depending on the disease, with the former case being more common [11].

Additionally, NMD was considered in therapeutic contexts: it is a determinant in the efficacy of cancer immunotherapy, where only those frameshift mutations that evade NMD predict a positive immunotherapy response [11, 44]. Mouse cancer models of genetic NMD targeting show favorable responses [46]. In the context of genetic diseases, it has been shown that NMD inhibition markedly increases the levels of mutated mRNAs [47], conceivably ameliorating the clinical phenotype severity, especially in combination with stop codon read-through therapy drugs [48–50].

If there were variability in NMD efficiency across individuals and/or tissues, it would impact the genetic disease phenotype as well as cancer evolution, and potentially impact on options for treatments, further motivating large-scale studies of differences in NMD efficiency.

Despite reports of individual cases, a systematic study of NMD efficiency across human individuals and tissues is lacking, as well as the understanding of its mechanistic basis. Here, we address this gap by assessing NMD efficiency genome-wide. To this end, we deployed two statistical methodologies, measured globally for each individual and compared across individuals or tissues (see schematic in Fig. 1). The first method draws on expression of NMD endogenous target gene (ETG) transcripts, while the second method assesses allele-specific expression (ASE) of PTC variant-bearing genes. We report significant variation in NMD efficiency between individuals, associates with somatic and germline genetic variation, observed both in the normal tissues, as well as various cancers arising in these individuals. We further suggest significant differences between human tissues/cancers, with lower NMD efficiency in the nervous and reproductive system. Furthermore, we explored the impact of individual NMD efficiency on the tumor evolution processes and clinical outcomes of cancer patients.

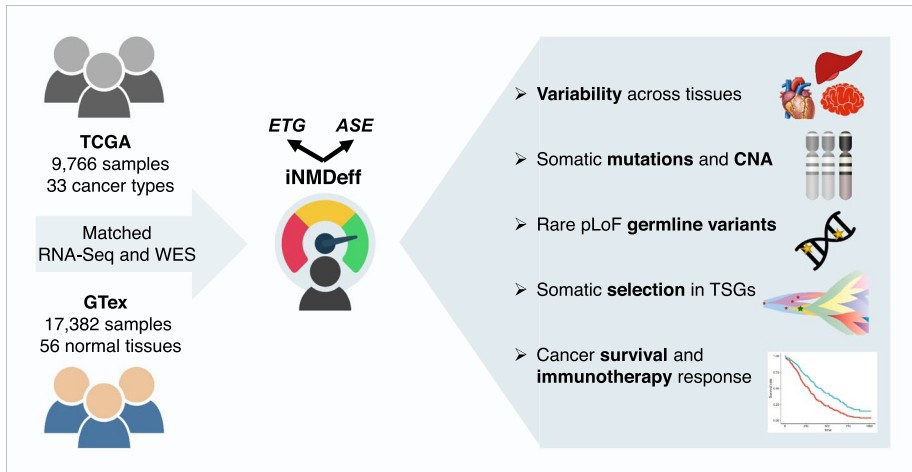

**Fig. 1** Variability and determinants of NMD efficiency across human tissues, tumors, and individuals. NMD: nonsense-mediated mRNA decay. ETG: endogenous NMD target gene method. ASE: allele-specific expression. CNA: copy number alterations. pLoF: putative loss-of-function. TCGA: The Cancer Genome Atlas project, GTex: Genotype-Tissue expression project. WES: whole-exome sequencing. iNMDeff: individual-level NMD efficiency. TSGs: tumor suppressor genes

## Results

### Individual-level quantification of NMD efficiency

We quantified the individual-level NMD efficiency (iNMDeff) by utilizing matched RNA-seq and whole-exome sequencing (WES) data from two cohorts: (i) tumor tissues in The Cancer Genome Atlas (TCGA), comprising 9,766 tumor samples from 33 different cancer types, and (ii) normal tissues from the Genotype-Tissue Expression (GTex) consortium, consisting of 979 individuals across 56 tissues, totaling 17,382 samples (with a median of 19 tissues per donor).

We devised two orthogonal methodologies to estimate iNMDeff in each tumor or normal sample (Additional File 1: Fig. S1A). Both are implemented using a negative binomial regression (see Methods) to model either transcript counts or allele expression counts, comparing two groups: NMD targets versus controls. Each method was subjected to rigorous filters to ensure that the set of transcripts/variants and individuals used were less confounded by other sources of variability (see Methods).

#### *Endogenous target gene (ETG) method*

We started with a set of genes that had been experimentally identified as endogenous NMD targets, collated from various studies that perturbed NMD activity in human cell lines [5, 6, 13, 14, 51] (see Methods). For each gene, we categorized its transcripts into NMD targets and control transcripts based on containing predicted NMD-triggering features (see Methods and Additional File 1: Fig. S1B). Here, we relied on two known features: (i) uORF at the 5' UTR, and (ii) at least one splice site (or EJC) within the 3' UTR. Next, for each NMD target gene, we selected two transcripts, one NMD target and the other NMD non-target, serving as an internal control. Our iNMDeff estimate is the regression coefficient of the indicator variable stating whether the pooled transcripts

within an individual are either NMD target (1) or control (0), from a negative binomial regression (see Methods); the iNMDeff can be interpreted as a negative log (base $e$) ratio of the expression levels of the NMD target transcripts divided by the expression levels of the control transcripts.

We calculated the iNMDeff for the different NMD gene sets from each study separately (see Methods): "NMD Karousis" [13], "NMD Colombo" [5], "NMD Tani" [6], and "NMD Courtney" [51], and we further used the transcripts tagged as "nonsense_mediated_decay" from the Ensembl gene annotation as an "NMD Ensembl" set. Additionally, genes found in at least 2 independent studies constituted a "NMD Consensus" gene set, and the union of genes across all studies was the "NMD All" gene set. For our control groups, we selected two sets of genes at random (excluding genes reported as NMD targets in any of the experimental studies mentioned above), selecting from genes with and without NMD-triggering features in their transcripts: "RandomGenes with NMD features," and "RandomGenes without NMD features." The former should behave similarly as the experimental NMD gene sets but is based only on computational prediction, while the latter is a negative control gene set.

### Allele-specific expression (ASE) method

We used exonic coding germline variants (population minor allele frequency, MAF, < 20%) to define three variant sets: (i) "NMD-triggering PTCs" resulting from nonsense variants and from frameshifting indels; (ii) "NMD-evading PTCs" also resulting from nonsense and indels; (iii) synonymous variants. The latter two were used as negative control sets. For each variant, the mutated allele counts observed in RNA-seq reads were used as NMD target levels and the wild-type allele counts at the same locus were used as control levels. We calculated the iNMDeff for each individual using the three aforementioned NMD variant sets separately. Similarly as in the ETG method, all variants within the individual were pooled to estimate the genome-wide iNMDeff, obtained from the regression coefficient of the indicator variable stating whether the variants are either mutated (1) or wild-type (0) (see Methods). The iNMDeff interpretation is that it is a negative log (base $e$) ratio of the raw RNA-seq allele counts of the mutant versus wild-type variants.

In summary, we computed the NMD efficiency for each individual—iNMDeff—multiple times, varying the NMD gene/variant set used, in TCGA (Additional File 2: Table S1) and in GTex (Additional File 2: Table S2) cohorts; including the negative controls, this amounted to 11 NMD gene sets for the ETG method, and 3 NMD variant sets for the ASE method. Henceforth, the ETG gene sets and the ASE variant sets will be collectively referred to as "NMD sets".

### Robustness of the individual-level NMD efficiency estimates

We focused on the ETG sets generated by combining genes from multiple studies—the stringent "NMD Consensus" and the permissive "NMD All" gene sets—while the ETG iNMDeff for the rest of the NMD gene sets are detailed in Additional File 1: Fig. S2A (TCGA) and Additional File 1: Fig. S2B-C (GTex). In TCGA and GTex, the iNMDeff estimates from the ETG method (Fig. 2A and Additional File 1: Fig. S2A-B) and from the ASE method (Fig. 2B and Additional File 1: Fig. S2C) show various trends that imply

they are reliable. For instance, the higher-confidence "NMD Consensus" ETG gene set displayed a readout of slightly higher efficiency (i.e. the log fold-difference in expression between the NMD target transcript and its non-NMD counterpart in the same gene) than the more permissive "NMD All" set ($p < 2e - 16$, Mann–Whitney $U$ test), which presumably contains weaker NMD target genes. The negative control gene set "RandomGenes without NMD features" shows negligible NMD efficiency ($p < 2e - 16$, when compared to "NMD Consensus"), with the median close to zero (Fig. 2A). See Additional File 3: Text S1 for more details.

To further support the concept of ETG method for estimating iNMDeff, we compared the iNMDeff estimates (from "NMD Consensus" set) with the expression of two well-known NMD target genes, *RP9P* and *GAS5* [5, 13]; of note, we excluded them from our methodology above to avoid circularity in this analysis. Encouragingly, individuals with high iNMDeff individuals with high *RP9P* and *GAS5* in tumor samples from TCGA (Additional File 1: Fig. S2D) and healthy tissue samples from GTex (Fig. 2C). Additionally, the gene expression of key NMD factors were analyzed. For NMD genes *UPF1*, *UPF2*, *UPF3A*, *UPF3B*, *SMG1*, *SMG5*, *SMG6*, *SMG7*, and *SMG9*, but not *SMG8*, a similar pattern to *RP9P* and *GAS5* was observed in both cohorts (Additional File 1: Fig. S2D-E). This is consistent with the negative feedback loop and cell-type-specific inherent in NMD factors, where inhibition of NMD by siRNA was reported to upregulate the mRNAs of all key NMD genes except for *UPF3A (SMG8*, and *SMG9* were not tested) [52]. The autoregulation is maintained through internal NMD-triggering features in the genes for the various NMD factors [52]. This correlation was less notable or even absent for EJC genes *MAGOH*, *RBM8A*, *EIF4A3*, and *CASC3* (Additional File 1: Fig. S2F-G).

To validate our ETG method for iNMDeff estimation, we sourced RNA-seq data from three cell lines (HeLa, HepG2, K562) from different studies [5, 53, 54] and processed it

(See figure on next page.)

**Fig. 2** Individual-level and tissue-level quantification of variable NMD efficiency. **A**, Estimation of individual-level NMD efficiency (iNMDeff) using the ETG method across 9,766 TCGA tumor samples, showcasing three NMD gene sets: NMD All, NMD Consensus, a random gene set with NMD-triggering features ("RandomGenes w/ NMD feat."); and as a negative control, a random gene set without NMD features ("RandomGenes w/o NMD feat."). **B**, iNMDeff estimations using the ASE method for two NMD variant sets: NMD-evading and NMD-triggering PTCs, alongside a non-NMD variant set as a negative control (Synonymous). ****p < 0.0001, by two-sided Mann–Whitney U test for A and B. **C**, Gene expression levels (TPM) of two well-known NMD targets (*RP9P* and *GAS5*) compared against the ETG iNMDeff (using the "NMD Consensus" gene set) sorted from lowest to highest along the X-axis in GTex. **D**, Cell line ETG NMD efficiency (cNMDeff) estimated in HeLa (n = 8), HepG2 (n = 2), and K562 (n = 2) cell lines using the "NMD Consensus" gene set (n = 130). cNMDeff was calculated using negative binomial regression, similar to our iNMDeff main analysis (see Methods). Comparison between *UPF1* knockdown (KD) and wild-type (WT) conditions showed consistent reduction in NMD efficiency across all cell lines. *P*-values were calculated using paired t-tests. Barplots and 95% confidence intervals represent the mean across cells. **E**, Variability in ASE iNMDeff between two example individuals from TCGA is depicted by the RNA variant allele frequencies (VAF) of coding germline PTCs, estimated as the proportion of mutant (MUT) allele RNA-seq counts over the total RNA-seq counts (MUT + wild-type (WT) counts) at the specific PTC loci. The individual with high iNMDeff (orange) has lower RNA VAF or due to NMD degradation of the MUT allele, and vice versa for the low iNMDeff individual (magenta). **F**, Spearman correlation between tissue-level NMD efficiency rankings based on median ETG iNMDeff and median ASE iNMDeff values per tissue (rank 1 denotes the tissue with highest median iNMDeff), for the GTex cohort. Tissues are grouped based on cell-of-origin: Nervous system-related tissues (Pan-nervous), Kidney-related tissues (Pan-kidney), Reproductive system tissues (Pan-reproductive), Gastrointestinal tissues (Pan-GI), and those originating from Squamous cells (Pan-squamous)

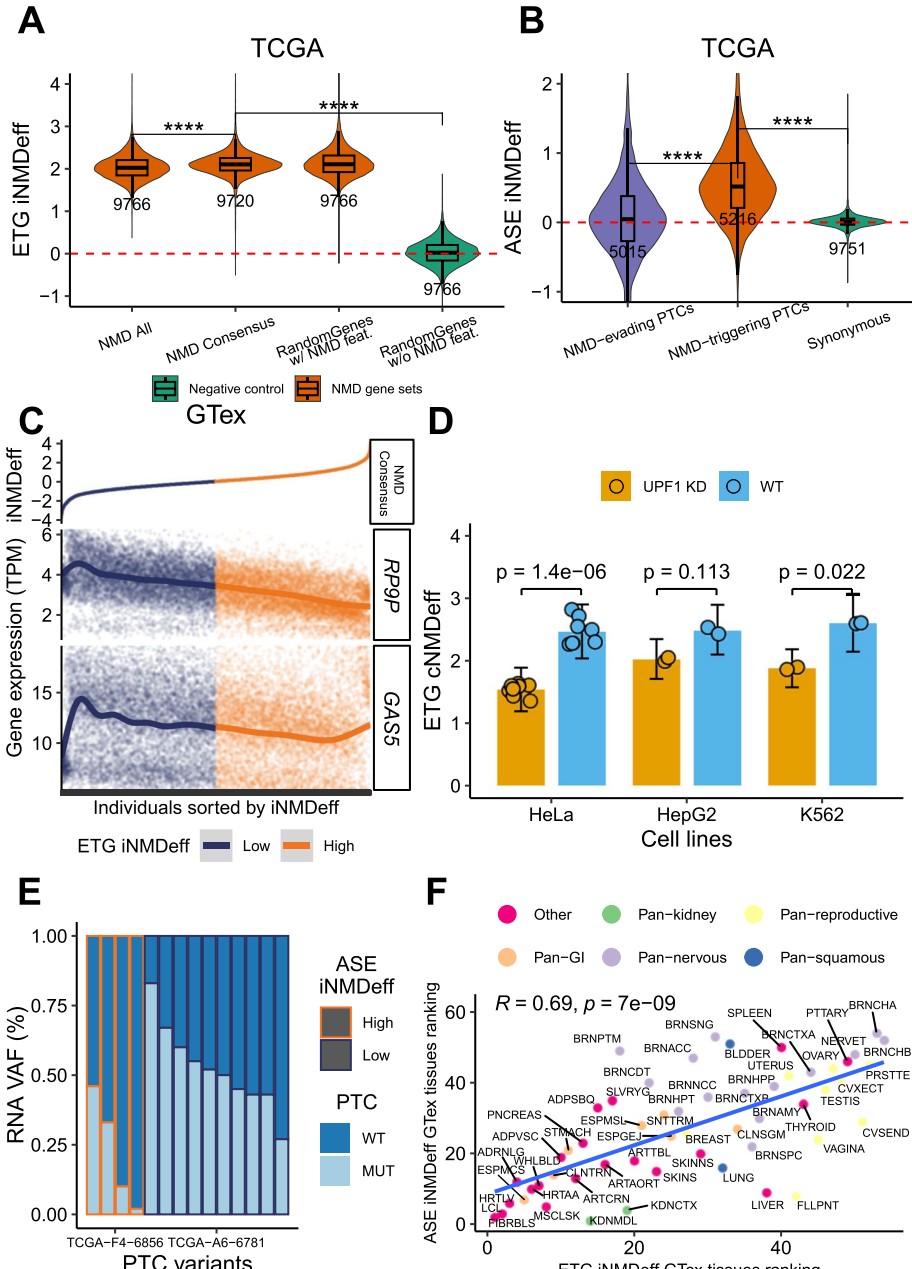

**Fig. 2** (See legend on previous page.)

using the same methodology as our main analysis (see Methods). Using our "NMD Consensus" set of NMD target genes, we applied negative binomial regression to estimate cell line NMD efficiency (cNMDeff), paralleling the iNMDeff in our main analysis (see Methods, Additional File 2: Table S3). We observed a remarkable separation of cNMDeff scores between *UPF1* knock-down (KD) and wild-type conditions (Fig. 2D). As predicted, *UPF1* KD significantly reduced NMD efficiency compared to wild-type cells in HeLa cells ($p = 1.4e-06$, $n = 8$ replicates), with consistent effects in K562 ($p = 2.2e-02$, $n = 2$) and HepG2 ($p = 0.11$, $n = 2$). Notably, every *UPF1* KD cell showed lower cNMDeff than any wild-type cell. As an additional analysis to confirm the robustness of the ETG

Consensus NMD gene set, we performed leave-one-out resampling analysis on our 130 NMD Consensus genes, consistently reproducing the *UPF1* KD versus wild-type differences (Additional File 1: Fig. S3A), demonstrating that our results are not driven by an outlying gene but rather stem from a consistent property of the NMD target gene set.

PRO-seq analysis of nascent RNA transcripts from three cell lines [55, 56] showed no significant differences in transcription initiation rates between NMD and non-NMD target transcripts, including when stratified by brain expression levels (Additional File 1: Fig. S3B), confirming that our ETG measurements reflect NMD activity rather than transcription rates (see Additional File 3: Text S1).

Next, as a test of reliability of the ASE iNMDeff estimates, we considered the differences in distribution of efficiency across the 3 sets of variants with different expected NMD efficiency (Fig. 2B). While ASE-derived individual level NMD efficiency is observable when calculated using "NMD-triggering PTCs", when these ASE estimates are derived from negative control variant sets, such as "NMD-evading PTCs" or "Synonymous", the NMD efficiency drops to almost negligible levels ($p < 2e-16$ for both variant sets, Mann–Whitney $U$ test, when comparing to "NMD-triggering PTCs," Fig. 2B).

To illustrate the ASE iNMDeff estimation, we present examples of raw allele-specific mRNA expression counts at PTC variant loci in two representative colon adenocarcinoma MSI (COAD_MSI) tumors from TCGA (Fig. 2E). After applying our ASE filtering criteria for germline PTC variants (see Methods), we retain four PTC variants with notably low RNA VAF or RNA-seq counts of their alternative alleles, relative to the reference allele, for the individual "TCGA-F4-6856," who was classified as having a high iNMDeff. This is consistent with the rapid degradation characteristic of an active NMD pathway. In contrast, the individual "TCGA-A6-6781," who exhibits a low iNMDeff, bears ten PTC variants, in 8 of which the mRNA counts of reference and alternative alleles are proportionally balanced, consistent with a heterozygous variant. This broadly consistent lack of ASE across 10 PTCs implies a global reduction in the NMD pathway activity in that individual (at least in the sampled tissue), affecting many PTC variants therein.

### Agreement between the two methods to estimate NMD efficiency

If both ETG and ASE methods are reflecting the true NMD activity of the individual, then a correlation between the two iNMDeff estimates would be anticipated. We proceeded to estimate pan-cancer and pan-tissue correlations between iNMDeff for ASE (using "NMD-triggering PTCs" variant set) and ETG (using "NMD Consensus" set) (Additional File 1: Fig. S4A-B), revealing a statistically significant correlation (Pearson $p < 2-16$ in both cases). The same trend was observed when correlations were calculated for each tissue or cancer stratified into subtypes, with positive correlations in 76 out of the 101 tested tissues/cancers (Additional File 1: Fig. S4C-D).

That ETG and ASE method agree well is seen upon stratifying the various samples by bins of iNMDeff. The top decile (samples with lowest ASE iNMDeff) of TCGA patients has a median scaled ETG iNMDeff of $-0.19$, while the bottom decile (samples with highest ASE iNMDeff) has a median scaled ETG iNMDeff of 0.20 (difference at $p = 4.12e-13$ by Mann–Whitney $U$ test, Additional File 1: Fig. S4E). Similarly, in GTex the top ASE decile has a median ETG iNMDeff of $-0.25$ while the bottom ASE decile has ETG iNMDeff $= 0.34$ ($p < 2e-16$, Additional File 1: Fig. S4F). Correlations between

the ETG and ASE estimates of iNMDeff across individuals are increasing as more stringent filtering criteria are applied to the ASE variants (from ~0.2 up to ~0.7 on GTex, see Additional File 1: Fig. S5A-B), presumably due to reduced noise due to low counts of PTC variants in some samples (see Additional File 3: Text S2). When pooling samples by tissue, we observe the correlation between ETG and ASE methods of R = 0.69 (GTex tissues, Fig. 2F) and R = 0.49 (TCGA tumor types, Additional File 1: Fig. S6A), supporting the concordance between the two NMD efficiency estimation methods, and suggesting systematic differences between tissues, which will be further tested below.

We also tested correlations between ASE iNMDeff and ETG iNMDeff and other biological covariates or technical variables, including considering separately our NMD gene/variant sets. The "NMD Consensus" ETG gene set correlates the most to the ASE iNMDeff (Additional File 1: Fig. S4A-B and S6B-C), and thus we used it as the default ETG method gene set (List of transcripts in Additional File 2: Table S4).

In summary, we developed two genomic methodologies for quantifying individual-level NMD efficiency—the ASE and ETG methods—which rely on RNA-seq signal in germline NMD-triggering PTCs variants, and in the "NMD Consensus" experimentally determined set of NMD target genes, for ASE and ETG methods, respectively. We demonstrated the robustness of these methods by various approaches, allowing us to use the ASE and the ETG method to investigate variation in iNMDeff across different human tissues and cancer types.

### Significant variability in NMD efficiency across human tissues and individuals
#### *Assessing inter-tissue variability of NMD efficiency*
Next, we applied our data set of iNMDeff estimates for 27,148 different tumoral and normal tissue samples to rigorously test the hypothesis that NMD efficiency varies across human tissues more than expected at chance. We grouped the iNMDeff estimates by cancer types in TCGA (Additional File 1: Fig. S7A), and by type of normal tissue in GTex (Additional File 1: Fig. S7B), and tested for differences between the tissues using a randomization test (see Methods for details and schematic from Fig. 3A). The test statistic we used, termed "Inter-Tissue iNMDeff Variability Deviation" (ITNVD), is a measure of how much the iNMDeff differences between tissue medians deviate from expectation if the iNMDeff values were randomly distributed across samples from different tissues. If there is no tissue-specific variability in iNMDeff, the ITNVD value should be close to 0.

The test was applied across both ASE and ETG estimated iNMDeff in the TCGA and GTex cohorts. We observed a significant, moderate variability deviation across both cancerous tissues in TCGA, and across normal tissues in GTex (Fig. 3B and Additional File 1: Fig. S7C). Encouragingly, the NMD gene/variant sets exhibited a much higher variability deviation than the negative control gene/variant sets, with a mean ITNVD of 0.27 versus 0.01 ($p = 3.5e{-}3$, $t$-test). Specifically, the ETG "NMD Consensus" set demonstrated the largest deviation (ITNVD = 0.62 in GTex and 0.26 in TCGA, both with $p = 5e{-}4$), and this was replicated in the ASE "NMD-triggering PTCs" variant set (ITNVD = 0.15 in GTex with $p = 3e{-}3$, and ITNVD = 0.07 in TCGA with $p = 5.3e{-}3$). In contrast, the variability deviation in the negative control sets was negligible: the ASE of "Synonymous" variant set showed a ITNVD of 0.01 in GTex and 0.03 in TCGA, and the ETG "RandomGenes without NMD-triggering features" set had a ITNVD of 0.02

in GTex and 0.03 in TCGA. These results suggest there is an important non-random iNMDeff variability associated with tissue and/or cell type identity, observed across both cancers and normal tissues.

A recent study suggested no differences in NMD efficiency between human tissues [57]. To reconcile this with our results we compared our ranking of tissues, ordered by the tissue median iNMDeff, with the ranking published by Teran et al. [57], who used ASE of PTCs from GTex data (Additional File 1: Fig. S8A). We found a strong correlation between their tissue rankings and our results, using either our ASE ($R=0.75$, $p=7e-10$) or our ETG estimates ($R=0.70$, $p=1.62e-8$). Our analysis demonstrates this differential NMD efficiency across tissues is significant.

### Significant differences between tissues in NMD efficiency

To identify which tissues and cancer types have the most and least efficient NMD and if their differences from other tissues are significant (Additional File 1: Fig. S7A-B), we compared the variation in iNMDeff within each specific tissue to the inter-tissue variation. We devised a statistic of Tissue-wise iNMDeff Deviation (TND, Additional File 2: Table S6-S7), with a baseline and a statistical significance derived from a randomization test (see Methods and schematic in Fig. 3A). The resulting distribution of deviations (Fig. 3C,D) suggests the tissue variability in NMD efficiency, where positive TND values indicate that a tissue exhibits higher iNMDeff than what would be expected by chance, and vice versa for negative values. Notably, 38 out of 54 tissues, and 19 out of 32 cancer types show a significant TND ($p < 0.05$ by randomization test) by the ETG method (for brevity, a selection is shown in Fig. 3C,D, and the complete List is in Additional File 1: Fig. S9A-B). The test for significant tissue-specificity of NMD efficiency replicates for 43% of these tissues in the ASE method as well. Next, we have categorized these into five broad primary groups for ease of interpretation: Nervous system-related tissues

(See figure on next page.)
**Fig. 3** Significant variability in NMD efficiency across human tumors, normal tissues, and individuals. **A** Schematic of the randomization tests, illustrating the methodologies for assessing NMD efficiency variability: Tissue iNMDeff Deviation (TND) and Inter-tissue iNMDeff Variability Deviation (ITNVD) (see Methods). **B** Displays ITNVD test scores for each NMD gene set in the ETG method (bottom panel) and NMD variant set in the ASE method (top panel). Scores for non-NMD negative control sets are also shown for comparison. Data are presented for both GTex (left) and TCGA (right) cohorts. Positive scores indicate variability in iNMDeff across tissues or cancer types, with stars (*) denoting statistical significance at $p < 0.05$, by randomization test. **C,D** show TND test scores for each cancer type (or subtype) within TCGA (**C**) and each normal tissue within GTex (**D**). Positive TND values indicate that a tissue exhibits higher iNMDeff than what would be expected by chance, and vice versa for negative values. Tissues are grouped based on cell-of-origin: Nervous system-related tissues (Pan-nervous), Kidney-related tissues (Pan-kidney), Reproductive system tissues (Pan-reproductive), Gastrointestinal tissues (Pan-GI), and those originating from Squamous cells (Pan-squamous). The groups of tissues were ordered based on the median TND scores, arranging them from the highest to the lowest median scores, top to bottom. Complete names of tissue acronyms can be found in Additional File 2: Table S5. **E** Scaled ETG and ASE iNMDeff estimates in low-grade glioma (LGG) and glioblastoma (GBM) brain tumors from TCGA, categorized by genetic or histological subtypes (first two panels, ordered by median ETG iNMDeff). It also displays the neuron cell type proportions derived from bulk RNA-seq deconvolution (third panel) and their pan-cancer Pearson correlation (R) with ETG iNMDeff (fourth panel). **F** Mirrors the analysis in **E** but focuses on GTex brain tissues, stratified by different subregions, showcasing the variability of NMD efficiency in normal brain tissue contexts and its correlation with neuron cell type proportions

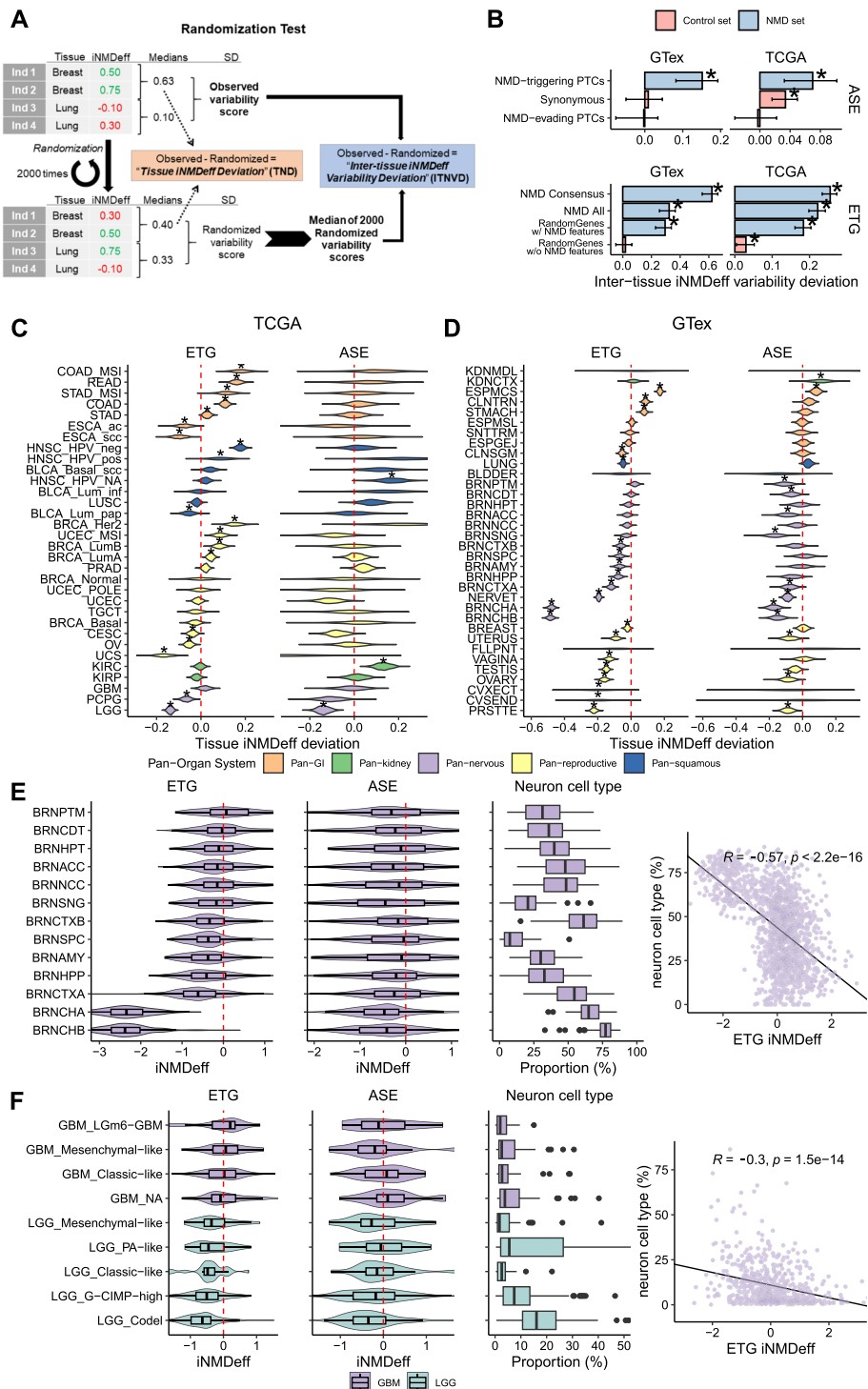

**Fig. 3** (See legend on previous page.)

(Pan-nervous), Kidney-related tissues (Pan-kidney), Reproductive system tissues (Pan-reproductive), Gastrointestinal tissues (Pan-GI), and those originating from Squamous cells (Pan-squamous) as denoted in a previous study [58].

The Pan-GI tissues displayed the highest iNMDeff in comparison to randomized values, i.e. positive TND, replicated across both the ETG and ASE NMD methods and in both TCGA and GTex cohorts. Specifically in TCGA (Fig. 3C), colon, stomach, and rectum adenocarcinomas (COAD, STAD, READ, respectively) exhibited significantly higher ETG iNMDeff than random (TND varying 0.03–0.16 for the 3 GI tissues, $p = 3.5e - 2$ to $1.2e - 3$). A similar pattern was observed in the GTex cohort (Fig. 3D) for normal colon (TND $= 0.09$, $p = 9.7e - 4$) and stomach (TND $= 0.08$, $p = 9.7e - 4$) tissues. Notably, MSI cancer subtypes, such as COAD_MSI and STAD_MSI, demonstrated higher iNMDeff compared to their non-MSI tumor counterparts (Additional File 1: Fig. S9C-D), placing them among the cancers with the highest ETG iNMDeff (TND $= 0.18$, $p = 1.2e - 3$ and TND $= 0.12$, $p = 2.3e - 3$, respectively). Esophagus mucosa (ESP-MCS) normal tissue displayed significantly higher iNMDeff in both ETG (TND $= 0.17$, $p = 9.7e - 4$) and ASE methods (TND $= 0.08$, $p = 2.7e - 3$); however, esophageal cancer showed a lower iNMDeff for both squamous cell carcinoma and adenocarcinoma (ESCA_scc and ESCA_ac, respectively; Fig. 3C) types, suggesting that some cell types might undergo a change in NMD efficiency during transformation from normal tissue into cancer.

In contrast to the Pan-GI, tissues classified as Pan-reproductive generally exhibited lower iNMDeff (Fig. 3C,D and Additional File 1: Fig. S9A-B) than expected based on randomization, i.e., negative TND. This trend was consistent across both TCGA and GTex cohorts and observed in both ETG and ASE methods. For example, in GTex's Pan-reproductive normal tissues: ovary, cervix ectocervix (CVXECT), and cervix endocervix (CVSEND) all showed lower iNMDeff in ETG (TND $= -0.16, -0.19$, and $-0.20$; $p = 9.7e - 4$, $8.7e - 3$, and $8.7e - 3$, respectively) and in ASE this replicated for ovary (TND $= -0.09$, $p = 2.2e - 2$). A similar trend was observed in TCGA, with ovarian serous cystadenocarcinoma (OV, ETG TND $= -0.05$, $p = 1.2e - 3$, ASE TND $= -0.02$, $p = 4.7e - 1$) and cervical squamous cell carcinoma and endocervical adenocarcinoma also exhibiting lower iNMDeff in both methods (CESC, ETG TND $= -0.04$, $p = 1.4e - 2$, ASE TND $= -0.08$, $p = 0.12$).

For breast normal tissue, a slightly lower trend towards lower iNMDeff was observed in ETG (non-significant in ASE), aligning with trends in TCGA for breast invasive carcinoma in the basal subtype and the normal-like subtype (Fig. 3C), but not in the other subtypes (BRCA_LumA, BRCA_LumB, BRCA_Her2). This suggests that cancer types with very diverse subtypes, such as breast cancer, may display differences in NMD efficiency between subtypes. For other normal Pan-reproductive tissues/cancers see Additional File 3: Text S3.

Analysis of Pan-squamous and Pan-kidney tissues overall did not yield significant and/or consistent results across the two NMD methods, in either TCGA tumors or GTex tissues (Fig. 3C-D). See Additional File 3: Text S3 for these and the rest of tissues/cancer types.

As an additional approach to assess if NMD efficiency varies across tissues, we created a linear model predicting iNMDeff, including tissue or cancer type, and other covariates (see Methods). In this model, the removal of the cancer type/tissue variable significantly reduced the explained variability ($R^2$), more so than removal of any of the other variables (Additional File 2: Table S8). This was evident in both ETG and ASE methods, where the

full model explained 17.1 and 58.7% of variability in TCGA and GTex cohorts, respectively, dropping to 6.3 and 13%, respectively, upon the tissue variable removal.

### Lower NMD efficiency in the tissues of the nervous system

Notably, the Pan-nervous group of tissues exhibited lower observed iNMDeff (Additional File 1: Fig. S7A-B) compared to randomized expectations (Fig. 3C,D), and this result remained robust when equalizing the number of samples across tissues (Additional File 1: Fig. S10). In the GTex cohort, almost all brain subregions showed a significant reduction in iNMDeff, many with large effect sizes, as observed in the TND statistic in one or both NMD methods. Specific examples replicating in both iNMDeff methods include the cerebellum (BRNCHA, ETG TND $= -0.48$, $p = 9.7e-4$; ASE TND $= -0.17$, $p = 2.8e-3$), the cerebellar hemisphere (BRNCHB, $-0.49$, $9.7e-4$; $-0.15$, $2.8e-3$), along with the cortex (BRNCTXA, $-0.12$, $9.7e-4$; $-0.08$, $1.4e-2$), and further the frontal cortex BA9, hippocampus, and the substantia nigra show significance in one of the NMD estimation methods (ASE or ETG) and a consistent trend in the other method. Additionally, nervous-related tissues such as the pituitary (PTTARY) and nerve/tibial (NERVET) displayed reduced iNMDeff in both methods (Fig. 3D). In TCGA data, two brain cancer types that arise from glial cells are available for comparison, and the same significantly reduced iNMDeff was evident in low grade glioma (LGG, $-0.14$, $p = 1.2e-3$, $-0.14$, $p = 8e-3$). This was further seen in the nervous system-associated pheochromocytoma and paraganglioma tumors (PCPG, $-0.06$, $p = 1.2e-3$, $-0.11$, $p = 0.12$), for both ETG and ASE methods.

Next, we hypothesized that different cell types within the nervous system might have different NMD efficiencies. To determine if the strong reduction in iNMDeff observed in normal and cancerous brain tissues is specifically linked to neuronal cells or to glial cell types, we utilized cellular composition deconvolutions from GTex tissues bulk RNA-seq data, as estimated by Donovan et al. [59]. Analyzing the neuron cell type proportions in each brain tissue type, ranked by the median ETG iNMDeff, revealed that tissues with lower ETG iNMDeff had a higher proportion of neurons (Fig. 3E). Notably, the two brain samples with the lowest iNMDeff, the cerebellum regions BRNCHA and BRNCHB, had the highest neuron proportions (medians of 65 and 77%, respectively). The correlation between neuron proportion and ETG iNMDeff across all brain subregions was negative (R $= -0.57$, Fig. 3E).

In TCGA tumor samples, we performed a deconvolution using the *UCDBase* method [60] and analyzed glioblastoma (GBM) and LGG tumor samples (see Methods). We stratified these cancer types into subtypes based on cell of origin or mutation type information[61] (Fig. 3F). The cancer subtype with the highest estimated neuron cell type proportion (median of 16%), LGG_Codel (characterized by the co-deletion of chr 1p/19q), exhibited the lowest iNMDeff (median ETG $-0.64$), while at the other end, GBM_LGm6 (characterized by the DNA methylation cluster LGm6) had a median neuron proportion of 2% and a median ETG iNMDeff of 0.21. The ETG iNMDeff correlation with estimated fractions of neurons across these cancer subtypes was also negative (R $= -0.30$, Fig. 3F).

As an additional analysis, we assessed the expression of the neuron-specific marker *RBFOX3* [62], which was negatively correlated with ETG iNMDeff (TCGA: R $= -0.31$,

$p = 4.4\mathrm{e}{-}15$; GTex: R$= -0.30$, $p < 2.2\mathrm{e}{-}16$), whereas the glia-specific marker *AQP4* [63] was positively correlated with iNMDeff (TCGA: R$=0.15$, $p=2.2\mathrm{e}{-}4$; GTex: R$=0.32$, $p<2.2\mathrm{e}{-}16$) supporting that neural cells in specific, rather than glia or other cell types infiltrating these tumor samples, are the cell types with lowered NMD activity in the central nervous system (Additional File 1: Fig. S11A-B). This provides an explanation why between the two major brain cancer types in TCGA, the LGG has lower iNMDeff compared to the GBM (median neuron proportion of 10% vs 3%, and median ETG iNMDeff of $-0.52$ vs 0.04).

In sum, our analysis underscores the significant variability of NMD efficiency across tissues, with high iNMDeff in digestive tract tissues, and low iNMDeff in the reproductive and nervous system tissues, the latter being associated with neuron content of the particular tissue.

### Extensive inter-individual variation of NMD efficiency

Having established there exist significant differences in NMD efficiency between human tissues, we next asked whether NMD efficiency is variable between different individuals, compared to a baseline variation between different variants within the same individual. For this, we derived the measure of NMD efficiency for a set of ASE PTCs (termed pNMDeff for PTC NMD efficiency, as detailed in Methods) in the TCGA separately for somatic and germline variants, and additionally for GTex germline variants. Next, we compared the intra-individual variation (different PTCs in the same individual) with a randomized baseline, and then the inter-individual variation (same PTC across individuals) with a control randomization (Additional File 1: Fig. S11C).

The randomization controls display higher variances than either inter-individual pNMDeff (excess variance$=0.83$–$1.01$ across the three datasets, $p=2.6\mathrm{e}{-}7$ to $1.3\mathrm{e}{-}83$) or intra-individual pNMDeff (excess variance$=0.41$–$0.78$, $p=1.2\mathrm{e}{-}1$ to $1.7\mathrm{e}{-}36$). The two tests for non-random variation suggest that each PTC has some inherent property determining its pNMDeff, and that this property can be shared across different PTCs within an individual but is different between individuals. The former test result is not unexpected given the various known rules of NMD efficiency depending on PTC placement [1, 8]; however, we note that here we selected those PTCs that should trigger NMD by the known rules, yet there is still systematic variation, therefore suggesting that additional NMD rules may exist. The latter test result means that the PTCs between individuals are more consistent (less variability) in pNMDeff than expected at chance, suggesting mechanisms governing individual-level NMD efficiency. Compared to the randomized controls, we observe a significant reduction in variance for both tests, reflecting the underlying rules that govern NMD efficiency. This finding underscores two key points: first, that there exists unexplored systematic variation in NMD efficiency between different PTCs even after accounting for known NMD rules, and second—more relevant to the focus of our study—that there is substantial variation in NMD efficiency between individuals.

As an additional statistic to support this, when examining the Spearman correlations of pNMDeff between PTCs (Additional File 1: Fig. S11D): the randomly selected pairs of samples for the same PTC (inter-individual) had a notably higher correlation than randomized controls, and similarly so for the randomly selected pairs of different PTCs

within individuals (intra-individual) ($p < =2.2e-16$ for both, Additional File 1: Fig. S11D). It bears mentioning that, notwithstanding this systematic variability in NMD efficiency both across PTCs and across individuals, much of the NMD efficiency across different contexts is preserved, compatible with the broadly conserved activity of the RNA surveillance via NMD. These findings motivated us to explore mechanisms that might underpin the inter-individual variability in NMD efficiency.

### Somatic pan-cancer CNA signatures are associated with NMD efficiency

A comprehensive analysis of genetic associations between somatic mutations and iNMDeff across TCGA tumors, analyzing over 6.4 million tests across various cancer and NMD-related genes (see List in Additional File 2: Table S9) and cancer types, revealed only two significant associations: a missense variant in the *TLX1* gene in lung adenocarcinoma (LUAD; linked with enhanced iNMDeff) and a truncating variant in the *CDH1* gene in breast invasive carcinoma (BRCA; linked with decreasing iNMDeff). See Additional File 3: Text S4, Additional File 1: Fig. S12 and Additional File 2: Table S10 for details.

We next proceeded to test associations of iNMDeff with somatic copy number alterations (CNAs). Given that CNAs often manifest as broader alterations affecting many neighboring genes up to an entire chromosome arm, we devised a custom method for somatic CNA association analysis. Instead of testing the gene-level CNA associations, which displayed inflation in our QQ plots (Additional File 1: Fig. S13A-B), possibly because of extensive genetic linkage between CNA of neighboring genes, our method considers recurrent large-scale, multi-gene CNAs as monolithic units.

To do so, we performed a sparse principal component analysis (sparse-PCA) on the pan-cancer gene-level CNA data (see Methods, Additional File 1: Fig. S14A), identifying 86 principal components, termed CNA principal component signatures (CNA-PCs, Additional File 2: Table S11). Notably, the leading (i.e. higher variance explained) 47 CNA-PCs represented recurrent larger-scale alterations, often spanning approximately arm-level changes, while the subsequent CNA-PCs pinpointed more localized recurrent CNA events (Additional File 1: Fig. S14B-C). Thereafter, we tested associations between the individual's weights of these CNA-PC signatures and our iNMDeff, at the pan-cancer and cancer type level (see Methods). Utilizing ASE method for discovery and ETG for validation, we identified 6 pan-cancer and 3 cancer-type specific (in UCEC and PCPG) CNA-PCs significantly associated with iNMDeff at FDR < 10% (List in Additional File 2: Table S12). We further retained cases where direction of effect was consistent between ETG and ASE iNMDeff methods, resulting in 3 robust pan-cancer associations with iNMDeff (Fig. 4A): CNA-PC3 (ASE FDR $= 1.3e-2$; ETG FDR $= 1.2e-37$), CNA-PC52 ($2.2e-2$; $3.1e-9$), and CNA-PC86 ($8.7e-2$; $8.6e-37$). Encouragingly, these recurrent CNA signatures exhibited consistent direction in their iNMDeff associations across various cancer types (Additional File 1: Fig. S15). Of note, the CNA-PCs 3 and 86 exhibited overlapping genome-wide patterns of CNA gains across chromosome segments (Pearson's R of 0.79).

### Chromosome 1q somatic copy number gain associates with reduced NMD efficiency

To further examine these three recurrent CNA signatures, we categorized individuals based on their scores for each CNA-PC. For CNA-PC3, individuals in group "High" predominantly exhibited an arm-level gain of chromosome 1q (gene dosage GISTIC scores ~ 1, Fig. 4B). In contrast, group "Mid," with intermediate CNA-PC3 scores, displayed a more localized gain around the 1q21-23.1 region (gene dosage GISTIC ~ 0.8, with ~ 0.4 towards the 1q end). The group "Low," with the lowest, ~ 0 scores in the sparse PCA, lacked notable CNA alterations at 1q (gene dosage change ~ 0). Interestingly, CNA-PC86 mirrored the pattern observed in CNA-PC3 but with a narrower focal gain at 1q21-23.1 (Additional File 1: Fig. S16A), thus both PC3 and PC86 recurrent CNA signatures reflect chr 1q gains, albeit at different scales. The individuals in groups "High" and "Mid" (by CNA-PC3 scores) who exhibit chr 1q gains, demonstrate a decrease in both ETG-estimated iNMDeff (Fig. 4C) and ASE (only "High" group, Additional File 1: Fig. S16B) and compared to those in group "Low," who lack any notable CNA alterations

(See figure on next page.)

**Fig. 4** Somatic pan-cancer CNA signatures are associated with NMD efficiency. **A** Pan-cancer CNA principal component signatures (CNA-PCs) associations with ASE iNMDeff across pan-cancer and individual cancer types, with the effect sizes shown as beta coefficients from linear models (see Methods). The vertical dashed red Line indicates a coefficient of zero, and the horizontal dashed red Line marks the significance threshold at 10% FDR for pan-cancer analysis. Three CNA-PCs (3, 52, and 86) showing significant and replicated pan-cancer associations are color-highlighted across cancer-types and labeled in the pan-cancer analysis. **B** CNA-PC3 reflects gene amplifications across chromosome 1, ordered by genome location, plotted along the *X*-axis, with amplifications assessed by averaging GISTIC CNA scores for each gene across TCGA samples (*Y*-axis). Samples are categorized into three bins based on scores from the pan-cancer CNA-PC3 signature: "High" showing complete chromosome 1q arm-level gain (gene dosage scores ~ 1), "Mid" with more localized gain in the 1q21.1–23.1 region (gene dosage ~ 0.8, with remaining chromosome at ~ 0.4), and "Low" showing no notable alterations in chromosome 1q (gene dosage scores ~ 0). **C** The left panel presents a scaled ETG iNMDeff of individuals classified by CNA-PC3 groups as "High," "Mid," and "Low." The right panel provides a validation in cell lines from the CCLE dataset (Achilles project), categorized by 1q CNA status: gain, neutral, and deletion; NMD was assessed with the proxy model for ETG cNMDeff based on gene expression data (see Methods). *P*-values are calculated through a two-sided Mann–Whitney *U* test. **D** Gene-level iNMDeff scores are calculated as the median iNMDeff across individuals with focal CNA amplifications for each gene on chromosome 1, arranged by genomic location. The top panel shows pan-cancer CNA frequencies per gene in TCGA. The middle panel displays gene-wise iNMDeff scores, with red intensity indicating co-amplification frequency with NMD factor gene *SMG5*. The bottom panel zooms into the 1q21.1–23.1 region, highlighting a reduction in iNMDeff when amplified, with some selected genes categorized as: Other, Candidates, NMD-related (NMD genes outside the 1q21.1–23.1 region) and Candidates NMD (NMD genes within the 1q21.1–23.1 region). **E** Candidate gene prioritization from chromosome 1q using two scoring criteria: Correlation between gene expression and ETG iNMDeff (Y-axis) or CNA amplification (X-axis). Candidate genes are expected in the bottom-right quadrant, indicating sufficient correlation between expression and iNMDeff (negative) and between expression and CNA amplification (positive). Vertical and horizontal dashed black Lines denote the thresholds for selecting candidates. Genes within 1q21.1–23.1 are shown in orange, with NMD genes (*SMG5*, *RBM8A*, *SF3B4*, *INTS3*) highlighted in red, and remaining chromosome 1q genes in black. Of 30 genes meeting thresholds, 20 reside within the 1q21.1–23.1 region (18 "Candidates" and 2 "Candidates NMD" genes: *SMG5* and *INTS3*), and 10 are from other 1q regions. Example candidate gene *PMF1* and known NMD-factor *SMG5* are displayed in the bottom panels, showing gene expression vs. ETG iNMDeff (bottom left) or CNA amplification (bottom right). **F** Simplified clustering heatmap of CRISPR KO co-dependency scores for 6 "Candidates," 2 "Candidates NMD" (*SMG5* and *RBM8A* plus *SF3B4* and *INTS3*), and 21 "NMD-related" genes from chromosome 1q (complete clustering found in Additional File 1: Fig. S20). Of note, *SF3B4* and *RBM8A*, despite not meeting ETG iNMDeff thresholds, were included as an additional "Candidates NMD" based on its related functions in NMD (spliceosome and EJC, respectively) and its location within 1q21.1–23.1. The analysis focuses on the 1q21.1–23.1 region, revealing a distinct cluster of candidate genes alongside NMD factor genes. Co-dependency scores have been normalized using the Robust PCA (RPCO) onion method [64], where generally, higher scores denote stronger gene functional associations

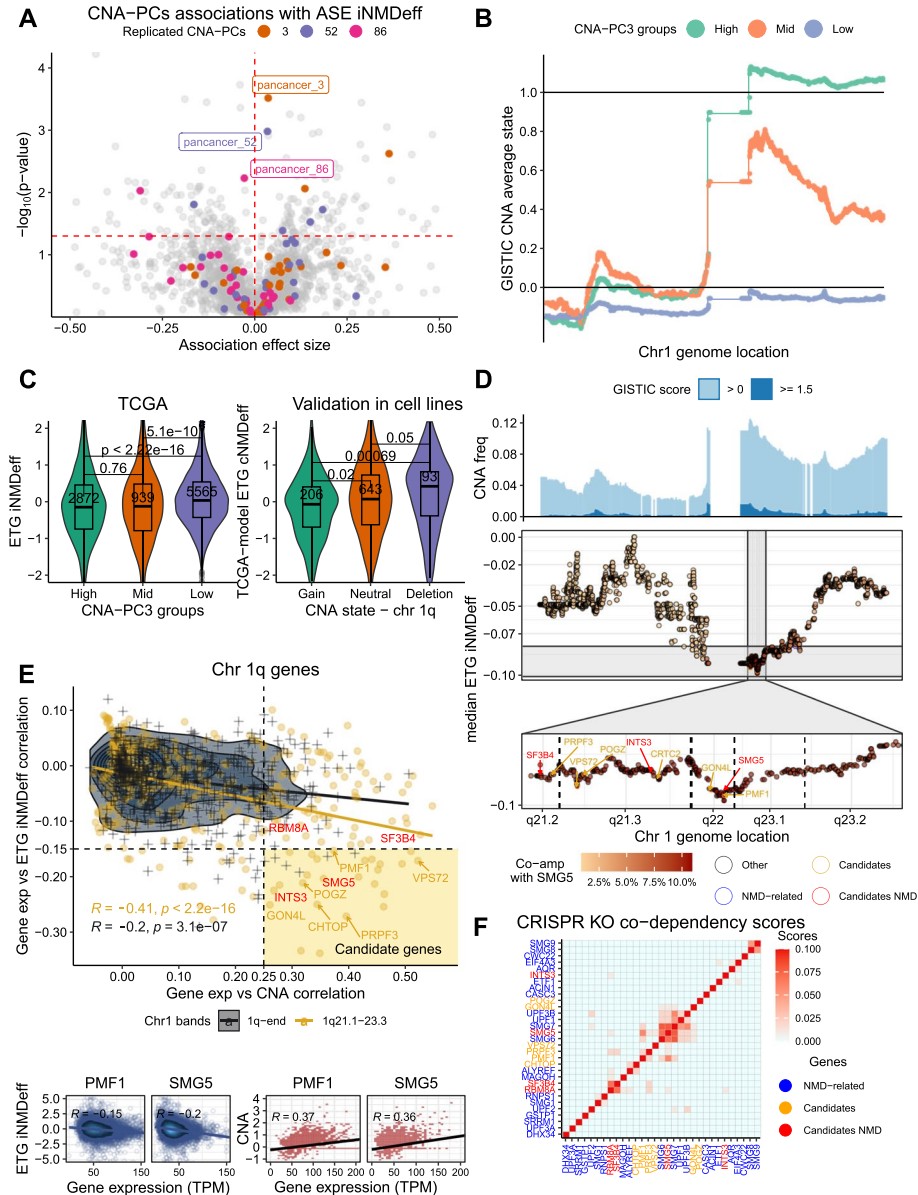

**Fig. 4** (See legend on previous page.)

(median ETG iNMDeff $= 2.07$ vs $2.12$, $p = 5.1e - 10$; median ASE iNMDeff $= 0.49$ vs $0.53$, $p = 4.7e - 2$, both two-sided Mann-Whitney tests comparing "High" against "Low"). A similar pattern was observed for individuals classified based on CNA-PC86 scores (Additional File 1: Fig. S16C).

To validate this 1q somatic CNA gain association with NMD efficiency in an additional data set, the GTex is not Suitable since it does not contain somatic alterations. Instead, we used the transcriptomic and CNA data of 1450 human cell lines from the Cancer Cell Line Encyclopedia (CCLE) [65]. We built proxy models of our ETG iNMDeff (see Methods) using global gene-level expression data to predict the NMD efficiency for every cell line (cNMDeff, Additional File 2: Table S13). The proxy models were trained in either TCGA or GTex cohorts (cross validation $R^2 = 0.4$ and $0.82$ for TCGA

and GTex, respectively, Additional File 1: Fig. S17A). We replicated our findings above (Fig. 4C and Additional File 1: Fig. S17B): cell Lines with 1q gains exhibited a decreased cNMDeff in comparison to diploid cells ($p = 0.02$ and non-significant, for TCGA-based and GTex-based proxy models of cNMDeff, respectively). This reduction was observed more clearly when comparing cells with 1q gains to those with 1q deletions ($p = 6.9e - 4$ and $p = 0.02$, for TCGA and GTex models respectively).

We next asked which cancer types were most influenced by the iNMDeff changes Linked with chromosome 1q CNA gains. To study this, we analyzed the distribution of iNMDeff in individuals across the Low/Mid/High groups, for the identified CNA-PC signatures, considering each cancer type separately (Additional File 1: Fig. S18A). BRCA, liver hepatocellular carcinoma (LIHC), cholangiocarcinoma (CHOL), and LUAD had over 50% of their samples falling into group "High" for CNA-PC3, indicating that 1q gains are common in breast, Liver, and lung cancers. This is also reflected in their iNMDeff. For instance, for 62% of LUAD samples that had a 1q gain, two-thirds (67%) of those also had a lower ETG iNMDeff than the median ETG iNMDeff across the LUAD samples with no gain. These 1q gains were proposed to be selected because they increase dosage of the *MDM4* oncogene [66], whose product downregulates the p53 protein post-translationally [67], hence phenocopying the loss of the tumor suppressor *TP53* [68]. However, there could be other oncogenes in chr 1q that are driving this gain.

When examining the more localized 1q21-23.1 amplification represented by CNA-PC86, 10–13% of the tumors from BRCA, LUAD, and LIHC were found in Group "High" (Additional File 1: Fig. S18B), and associated with reduced iNMDeff. Furthermore, other cancers, including uterine carcinosarcoma (UCS), lung squamous cell carcinoma (LUSC), OV, and skin cutaneous melanoma (SKCM), also exhibited high incidences of both these recurrent 1q gain CNA-PC3 and PC86 (Additional File 1: Fig. S18A-B). We note this narrower CNA segment does not include the *MDM4* oncogene, which is located at 1q32.1, thus other genes must be driving this recurrent genetic change in CNA-PC86, with possible candidates being known cancer genes *BCL9* (1q21.2), *ARNT* (1q21.3), *SETDB1* (1q21.3), or *SF3B4* (1q21.2).

### Identifying NMD-associated genes via focal CNA analysis

While CNA gains can affect numerous dosage-sensitive genes in one event, only a subset of the genes or a single gene within the CNA segment are expected to be directly implicated in the observed phenotype, here the NMD deficiency. Within the 1q arm reside three known NMD factors and two NMD-related genes identified in experimental and computational studies [69, 70]: *RBM8A* (located at q21.1), a core component of the EJC; *SMG5* (q22) and *SMG7* (q25.3), both core NMD factors; *SF3B4* (q21.2), a known splicing factor and oncogene; and *INTS3* (q21.3), involved in snRNA transcription. To prioritize candidate genes responsible for the NMD deficiency associated with 1q gains, we computed the median iNMDeff of individuals having a focal copy number amplification for each gene separately; considering only focal CNA events for this analysis helps distinguish between the Linked genes. Mapping these gene-wise scores along chr 1, we observed that tumor samples with CNA gains in genes proximal to the 1q21-23.1 region (Fig. 4D, top panel) have a pronounced reduction in ETG iNMDeff scores (Fig. 4D, bottom panel) and similarly in the ASE iNMDeff scores (Additional File 1: Fig.

S19). Notably, the NMD-related genes mentioned above, except for *SMG7*, are situated within this narrower 1q21-23.1 region.

To narrow down potential candidate NMD-modulating genes within this gene-dense region, we employed two scoring criteria across all genes from chr 1q. First, we calculated the correlation between gene expression—presumably the downstream effect of the CNA relevant to the phenotype—and ETG/ASE iNMDeff. Second, as a supporting test for prioritization, we assessed the correlation between gene expression and CNA. The correlation between these two scores (Fig. 4E and Additional File 1: Fig. S20A for ASE) is stronger within our target region (R = −0.41) than outside (R = −0.2), supporting that this segment harbors causal genes for reduced iNMDeff. We established thresholds for selecting candidates based on the two criteria (Fig. 4E), and among the 30 genes that passed the thresholds, 20 were from within the 1q21-23.1 region ("Candidates"), including *INTS3* and *SMG5* ("Candidates NMD" genes). Parsimoniously, the CNA of the core NMD factor *SMG5* might generate the effect on NMD efficiency; however, we considered the hypothesis that CNA of other genes within this segment might affect NMD efficiency.

To pinpoint the potential causal gene(s), we considered genetic interactions inferred from gene-level CRISPR screening data obtained from the Achilles Project [71], allowing to infer functional links between genes by studies of codependency profiles across cell lines. Here, we used the de-biased data (*onion* method, Robust PCA (RPCO)), with improved power for inferring gene function [64] (see Methods). Our hypothesis posits that if these candidate genes from 1q21-23.1 are indeed implicated in NMD efficiency, their CRISPR knockout (KO) fitness effects should correlate with established NMD factor genes like *UPF1*, *UPF2*, *UPF3B*, *SMG1*, and others; the codependency implies a functional link.

Therefore, we clustered and visualized these codependency CRISPR scores involving our set of candidate genes (18 "Candidates" and 4 "Candidates NMD"), 21 "NMD-related" genes, and 18 random negative "Control" genes on chr 1q but outside the 1q21.1−23.1 region (see Methods; note that *KIAA0907* was excluded from the analysis due to lack of CRISPR data). *SF3B4* and *RBM8A*, despite not meeting ETG iNMDeff thresholds, were included as an additional "Candidates NMD" based on their related functions to NMD (spliceosome and EJC, respectively) and its location within 1q21.1−23.1. This revealed a prominent cluster of functional interactions (Additional File 1: Fig. S20B) comprising NMD factor genes such as *SMG5*, *SMG6*, *SMG7*, *UPF1*, and *UPF3B*, suggesting that the CRISPR RPCO scores [64] are powerful for identifying NMD factors. A simplified visualization (Fig. 4F) focusing on 6 candidate genes co-clustering with NMD factors, and excluding controls, showed two genes from 1q21.1−23.1, *PMF1* and *GON4L*, showing notable CRISPR codependency scores with *SMG5* (0.060 and 0.006, for *PMF1* and *GON4L*, respectively), *SMG6* (0.020 and 0.006), and *SMG7* (0.019 and 0.012). Full set of genes is visualized in Additional File 1: Fig. S20B. For scale, the CRISPR codependency score between *SMG5* and *SMG7* is 0.077, and the majority of scores with control genes are 0. Furthermore, *VPS72* and *PRPF3* exhibited codependency with *SMG5* (0.080 and 0.011, respectively), and so might have relevance.

In a more detailed quantitative analysis, we examined all 274 genes within 1q21.1−23.1. For each, we computed the mean CRISPR codependency scores with 10 core

NMD factor genes and compared these with the mean CRISPR score for the same gene with the previous 18 random negative control genes on 1q. We identified 14 genes with FDR < 5% (one-sided Mann–Whitney *U* test). Notably and as expected, 3 of these genes are directly involved in NMD (*SMG5*, *RBM8A*, and *SF3B4*) (Additional File 1: Fig. S20C). The 4 genes identified above by a visual inspection of the clustering—*PRPF3* (FDR = 4.6%, one-side Mann–Whitney *U* test), *GON4L* (4.6%), *VPS72* (3.1%), and *PMF1* (2.5%)—also emerged as significant in this analysis. Additionally, 7 genes not considered as candidates due to not meeting our initial threshold criteria (correlations with gene expression and CNA) were significant here. Interestingly, 10 of these 14 genes are related to a RNA metabolic process (see Discussion). Overall, our analyses of gene overexpression and genetic interactions underscore that the 1q21.1–23.1 genomic region harbors multiple genes with potential to modulate NMD activity upon CNA gains, some of which have not been previously identified as NMD factors, with prime candidates being *PMF1* and *GON4L*.

Additionally, we analyzed the CNA-PC52 signature, also associated with reduced NMD efficiency and characterized by low-level gains in chromosome 2q31.1-2q36.3. This region, commonly altered in several cancer types (especially testicular germ cell tumors), contains four NMD-related genes (*CWC22*, *SF3B1*, *NOP58*, and *FARSB*). Further analysis supported the possible role of genes in this region in modulating NMD efficiency (see Additional File 3: Text S5 for details).

### Rare germline variant association analysis identifies genes associated with NMD efficiency

We hypothesized that an additional cause for inter-individual variation in NMD efficiency could be germline genetic variation across individuals, affecting NMD-relevant genes. To test this, we conducted a gene-based rare variant association study (RVAS, see schematic from Fig. 5A), relying on the SKAT-O test [72] (see Methods).

We defined three stringency thresholds for determining putative loss-of-function (pLoF) for the rare germline variants, i.e. those with a MAF < 0.1%. One threshold involved only NMD-triggering PTC variants, and the other two additionally included the predicted deleterious missense variants using CADD scores [73] at two different cutoffs: ≥ 25 (more stringent) and ≥ 15 (permissive).

We tested all genes in which at least 2 individuals carried a rare pLoF variant, for the three pLoF stringency levels, using the two iNMDeff estimation methods, ASE and ETG, and correcting by potential confounder variables (see Methods). Testing was performed in each cohort, TCGA and GTex, contributing cancer and normal tissue, respectively. Significant associations identified in one cohort (TCGA or GTex) were subsequently re-tested requiring replication in the other cohort (GTex or TCGA, respectively), relying on pairing 18 TCGA tumor types to corresponding GTex normal tissues (Additional File 2: Table S5). Replication did not require the use of the same iNMDeff method as in the discovery cohort, thus a significant TCGA association detected using the ASE iNMDeff method could be replicated in GTex using either ASE or ETG iNMDeff methods.

A total of 850,721 tests associating iNMDeff with rare germline variants were conducted in GTex and 221,613 in TCGA, after excluding dataset-tissue pairs to prevent inflation (lambda ≥ 1.5). The number of tests in GTex was higher due to multiple tissue samples analyzed per individual, allowing separate testing of the same gene across

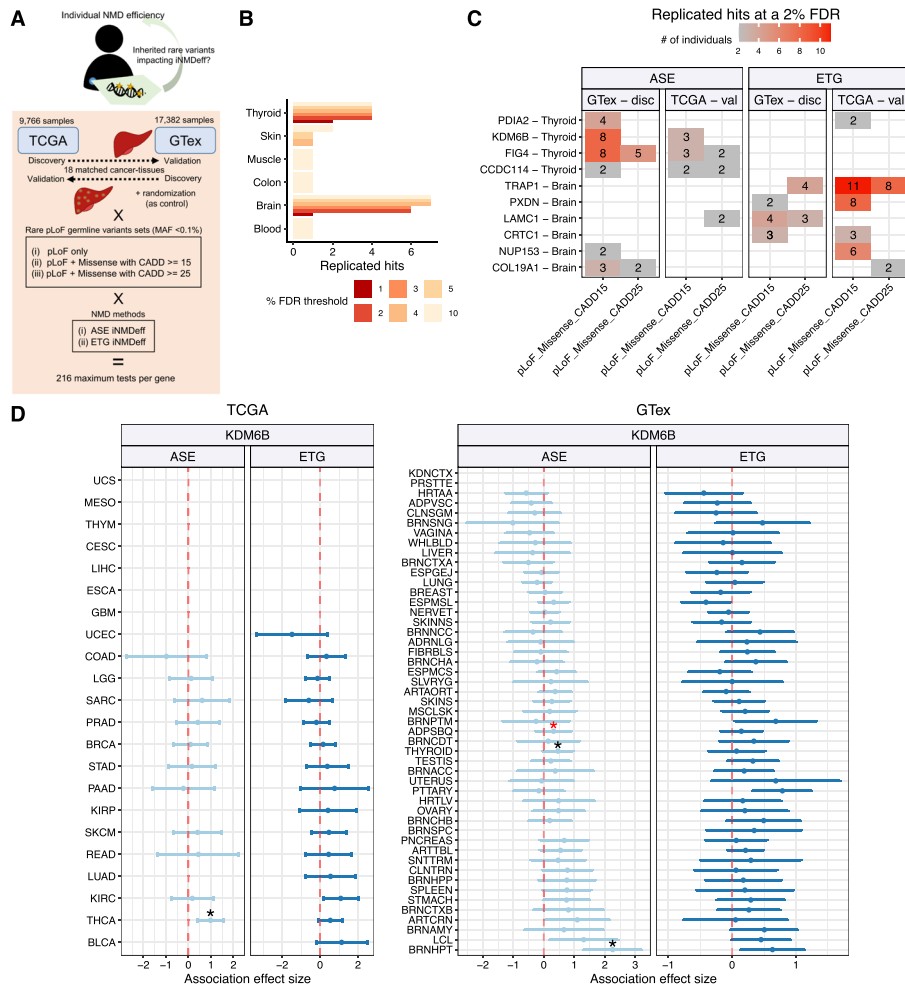

**Fig. 5** Rare deleterious germline variants are associated with NMD efficiency. **A** Overview of our rare variant association analysis (RVAS), illustrating the SKAT-O framework for identifying associations between rare germline putative loss-of-function (pLoF) variants and individual NMD efficiency. We used either TCGA or GTex as discovery cohorts with the other serving as validation, and vice versa. The analysis spans 18 matched cancer-normal tissues, three sets of pLoF variants, and employs two distinct iNMDeff methods for gene-level associations. iNMDeff values were randomized as a control measure. **B** Number of replicated gene-tissue pairs across various tissues against the range of FDR thresholds from 1 to 10%. **C** shows replicated gene-tissue pairs at a 2% FDR threshold, categorized by pLoF variant set, NMD method (ASE or ETG), and the cohort where the association was identified (either discovery or validation). The values indicate the total count of individuals harboring a rare pLoF variant within the successfully replicated genes. Replication did not require the use of the same iNMDeff method as in the discovery cohort. For instance, for *LAMC1* association, it was discovered in normal brain tissues (GTex) using ETG method and was further validated in brain tumors (TCGA) using the ASE method. **D** Gene burden test for the association between rare pLoFs within the *KDM6B* gene and iNMDeff. Effect sizes (beta coefficient) from linear model associations are shown for cancer types (TCGA, left panel) or normal tissues (GTex, right panel), segmented by NMD method, including the 95% confidence intervals. An asterisk highlights statistically significant associations of the original SKAT-O analysis at FDR < 5% (black) or FDR < 20% (red). Tissues with no values did not have samples with rare pLoFs in the gene

various tissues. Across the three variant sets, two NMD methods, and two cohorts, considering all directions of validation, we identified 48 tissue-specific significant associations (nominal FDR ≤ 2%) that were successfully replicated. This is significantly more compared to the 24 significant associations found by randomizing the iNMDeff

estimates ($p = 7e - 3$, Additional File 1: Fig. S23A), supporting the presence of non-random associations between iNMDeff and germline rare variants.

Out of these replicated associations, 27 were discovered in GTex and validated in TCGA. These hits span 10 unique individual gene-tissue type pairs clustered in 2 matched-tissues: thyroid (4) and brain (6) (Fig. 5B,C). Expanding the nominal FDR threshold to a more permissive 10% revealed additional replicated genes (List in Additional File 2: Table S14) across four more tissues: blood (1), colon (1), muscle (1), and skin (2), suggesting that future analyses with larger sample sizes may identify additional NMD-modulating genes and that NMD-modulating variants may manifest in various tissues (Fig. 5B).

We further calibrated these nominal FDRs by comparing against a distribution of FDRs obtained from running SKAT-O on a randomization of the iNMDeff estimates. This resulted in estimates of empirical FDR (Additional File 1: Fig. S23A), where the nominal FDR of 2% corresponded to a conservatively estimated empirical FDR threshold of ~30%, calculated as 3 replicated hits from randomized iNMDeff data, versus 10 hits obtained from our observed iNMDeff values. The 10 gene-tissue pair associations that met our criteria are detailed in Fig. 5C, categorized by pLoF stringency variant set and NMD estimation method. The same table for an FDR of 5% is also available (Additional File 1: Fig. S23B).

To assess the effect size of the associations for 10 replicated genes and determine their direction of effect on iNMDeff, we utilized a gene-burden test through general linear regression (see Methods) to complement the SKAT-O test results. We conducted this estimation separately for each NMD method in both TCGA and GTex, across all tissues (Additional File 1: Fig. S24A-B). Among the genes studied, the chromatin modifier *KDM6B* stood out in that it demonstrated a consistently significant positive effect size in the thyroid (0.46, SKAT-O FDR = 2%, in ASE), our GTex-discovery tissue, and in the TCGA-matched cancer type THCA (0.98, SKAT-O FDR = 0.5%, in ASE) (Fig. 5D and Additional File 1: Fig. S24A-B). The effect sizes in the ETG method were consistent with the ASE method (0.07 in GTex thyroid, 0.98 in TCGA THCA). The coefficient direction is positive, indicating that individuals with a pLoF variant in *KDM6B* have a higher iNMDeff than those without (Additional File 1: Fig. S25). Moreover, a positive effect size is also observed in the hypothalamus brain region (BRNHPT in GTex: 2.25 in ASE; 0.63 in ETG), and in the adipose subcutaneous tissue (ADPSBQ: 0.32 in ASE; 0.14 in ETG), with the ASE SKAT-O FDR = 0.55% and 17%, for brain and adipose, respectively. The other tissues and cancer types, while not significant with SKAT-O, usually exhibited a similar positive trend (Additional File 1: Fig. S24A-B), suggesting this *KDM6B* association with NMD efficiency may not be thyroid-specific but likely impacts other tissues as well.

From the 6 brain-identified replicated genes (Fig. 5C), the consistency of effect sizes (pLoF burden) across the two cohorts (GTex tissues and TCGA tumors) was observed only for *COL19A1* (see ASE in Additional File 1: Fig. S24 and S25). For *NUP153*, consistent effects were observed across various brain tissue samples within GTex but did not extend to TCGA brain tumors, possibly due to normal-cancer tissue differences and/or due to that glioma brain cancers originate from glia rather than neurons (Additional File 1: Fig. S24 and S25). Overall, our analysis provides evidence that deleterious germline

variants in diverse genes may alter NMD efficiency in individuals, with the histone demethylase *KDM6B* being a prime candidate for follow-up.

### NMD efficiency impacts somatic selection and cancer patient survival

Previous research [11, 45] reported that somatic nonsense mutations (producing PTCs) in tumor suppressor genes (TSGs) are positively selected specifically in NMD-triggering gene segments. We investigated whether the global NMD efficiency of an individual influences the selection of these mutations in tumors, using a dN/dS analysis across TCGA tumor samples (see Methods).

For TSGs, we first checked missense mutations, which normally do not trigger NMD. There was no significant difference in driver gene selection between high and low iNMDeff individual groups, defined either by the ETG method (Fig. 6A, right panel) or the ASE method (Additional File 1: Fig. S26A, right panel) for iNMDeff (missense mutations located in NMD-triggering regions ETG dN/dS = 1.19 vs 1.20, $p = 0.53$, and NMD-evading regions dN/dS = 1.03 vs 1.10, $p = 0.61$, one-sided Wilcoxon paired tests). However, in the case of nonsense (PTCs) mutations, we observed a difference with contrasting directions: genes with NMD-triggering PTC variants exhibited a significant trend towards higher positive selection in tumor samples with high iNMDeff, whereas tumors with low iNMDeff demonstrated less positive selection (dN/dS = 1.64 vs 1.33, $p = 4.7e-02$ for ETG iNMDeff, Fig. 6A, left panel), thereby suggesting that individual-level, global NMD efficiency can indeed shape selection on cancer genes (consistent trend in ASE iNMDeff: $p = 0.13$, Additional File 1: Fig. S26A, left panel). We also observed increased

(See figure on next page.)

**Fig. 6** Individual-level NMD efficiency modulates somatic selection in tumor suppressor genes and impacts progression-free survival (PFS) and response to immunotherapy. **A** The dN/dS ratios (via dNdScv method) of tumor suppressor genes (TSGs) for both NMD-triggering and NMD-evading nonsense (left panel) and missense (right panel) somatic mutations are compared between two groups of individuals with high and low iNMDeff, as determined by the median ETG iNMDeff. Statistical significance is assessed using one-sided Wilcoxon tests on paired data. PTCs were classified as NMD-triggering if they met all criteria: (1) > 250nt from Transcription Start Site (TSS), (2) in exons ≤ 500nt, and (3) not in the last exon or last 55nt of the penultimate exon. Conversely, PTCs were classified as NMD-evading if they met any criteria: (1) ≤ 250nt from TSS, (2) in exons ≥ 1000nt, or (3) in the last exon or last 55nt of the penultimate exon. **B** Proportion of CD8 + cytotoxic T-cell infiltration in TCGA cohort patients for melanoma (SKCM), pan-kidney cancers (KIRC and KIRP), and breast cancer (BRCA), treated with either immunotherapy or chemotherapy. Immune infiltration was inferred from RNA-seq data using the quanTIseq TIL10 signature from The Cancer Immunome Atlas (TCIA). Tumor samples were classified as high or low NMD efficiency based on top 30% vs bottom 30% ETG iNMDeff estimates, respectively. *P*-values were calculated using a one-sided Mann–Whitney U test. **C–E** Kaplan–Meier (KM) survival curves comparing PFS outcomes of the top 20% of individuals with the highest iNMDeff against the lowest 20% in TCGA SKCM. The analysis is divided into three patient categories: no treatment data available (**B**), those treated with immunotherapy at least once (**C**), and those treated exclusively with chemotherapy, excluding immunotherapy patients (**D**). Log-rank test *p*-values quantify the statistical significance of the survival differences. **F,H** Validation of the KM curves for PFS in immunotherapy treated patient cohorts from Liu et al. [76] (SKCM, **E**), Carrol et al. 2023 (EAC—esophageal adenocarcinoma, **F**), and Motzer et al. 2020 (RCC—renal cell carcinoma, **G**). **I** Performance of logistic regression models predicting immunotherapy response based on: (i) iNMDeff alone; (ii) TMB alone; (iii) a combination of iNMDeff, TMB, and their interaction term iNMDeff*TMB. iNMDeff was derived either from the ASE (left panel) or ETG (right panel) methods. Each model's predictive power is indicated by the area under the curve (AUC) on the *Y*-axis, with the dataset and sample size (*n*) specified on the *X*-axis. A black dashed Line at an AUC of 0.5 represents the threshold for random prediction accuracy and an asterisk denotes when the AUC is significantly different from it at a *p*-value < 0.05 via one-side DeLong test

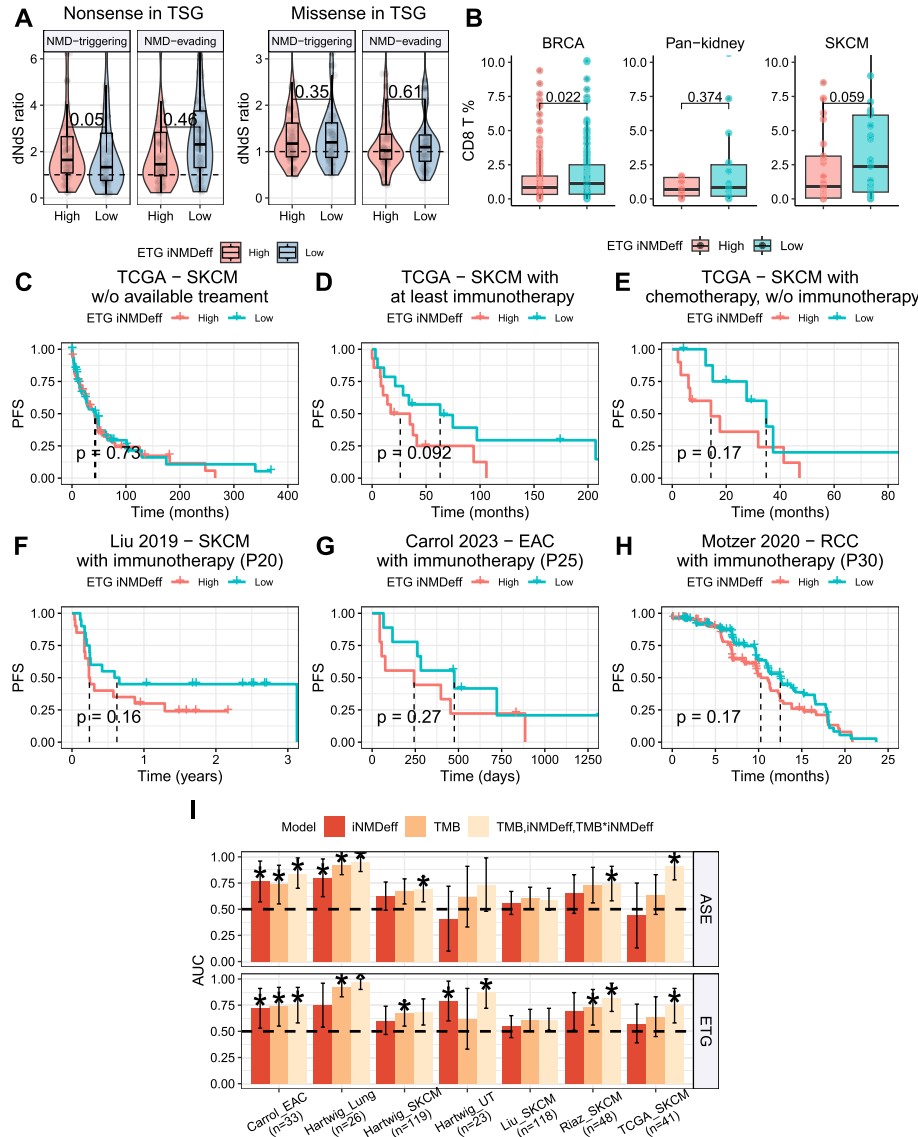

**Fig. 6** (See legend on previous page.)

positive selection for genes with NMD-evading PTC variants in tumors with low iNMD-eff (Fig. 6A), as expected from selective advantages through NMD-independent mechanisms, likely via complete protein loss-of-function in driver genes or dominant-negative effects of some PTCs [74, 75].

We identified strong iNMDeff-associated PTC selection in some individual genes particularly *SMAD4* and *TP53* and some degree of effect on various other genes (see Additional File 3: Text S6, Additional File 1: Fig. 26B-C and Additional File 2: Table S15 for details). Additionally, we noted tentative associations with higher mutation burden cancer types, including MSI, POLE-mutant, and lung cancers (details in Additional File 3: Text S6).

### NMD efficiency is associated with overall survival of cancer patients

In light of our previous findings that PTC-level NMD efficiency can modulate phenotype severity of genetic diseases [11], we explored, here focusing on cancer, whether individual-level NMD efficiency influences cancer survival outcomes. We considered Kaplan–Meier (KM) curves and Cox proportional hazards regression models with overall survival (OS) as our outcome, binarized into "High" and "Low" groups at varying thresholds, across 47 various TCGA cancer types stratified into subtypes, including also pan-cancer, and employing randomized data as a negative control (see Methods).

After applying FDR correction to all 960 tests (480 FDR adjusted meta-p combining ETG and ASE estimates of iNMDeff), we identified 17 in 960 significant tests at a nominal FDR of 5%, compared to 23 of the 9600 randomized tests, yielding an empirical FDR of 16% by randomization (results for individual percentiles given in Additional File 2: Table S16). The significant results spanned four cancer types: SKCM (median FDR meta-p $=0.78$%, median hazard ratio (HR) $=1.61$), ESCA_ac (FDR $=3.2$%, HR $=2.22$), PRAD (FDR $=2.8$%, HR $=0.11$), and LUAD (FDR $=0.75$%, HR $=3.49$). Of note, in ESCA_ac and LUAD, we were unable to calculate the Cox model for ASE at due to insufficient data; thus, the reported meta-p reflects only the ETG method. In PRAD, higher iNMDeff correlates with increased survival (Additional File 1: Fig. S27A-B). Roles of NMD in carcinogenesis are two-fold, where NMD can boost some tumor-promoting but also some tumor-suppressive mechanisms [30, 77]. We suggest that the balance tips towards the latter option for prostate cancer, where a more active NMD implies a less aggressive tumor. In contrast to the lowly mutated PRAD, in the often high mutation burden SKCM (Additional File 1: Fig. S27C-D), and also ESCA_ac (Additional File 1: Fig. S27E-F) and LUAD (Additional File 1: Fig. S27G-H), higher tumor iNMDeff is associated with reduced patient survival. This finding aligns with proposed approaches to potentiate immune response to tumors by NMD modulation [1, 46, 78, 79]; we note a certain number of SKCM samples in TCGA were from patients who received immunotherapy (73 out of 469; however only 5 LUAD samples in TCGA, and none from ESCA_ac and PRAD, were treated with immunotherapy). We posit that patients with lower iNMDeff might experience better treatment outcomes, as higher NMD efficiency should result in the silencing of frameshifted and/or truncated neoantigenic peptides, relevant to the efficacy of immunotherapy. Consistent with this notion, when assessing the data describing CD8 + cytotoxic T-cell infiltration (inferred from RNA-Seq in the TCGA cohort patients with treatment data), we noted positive trends towards higher CD8 + infiltration in those tumors having lower NMD efficiency. This was observed in cancer types commonly considered immunogenic—melanoma and kidney cancers (KIRC + KIRP)—and interestingly also in breast cancer (Fig. 6B). These findings were further validated in the independent Hartwig Medical Foundation cohort, where we observed significant increases in content (estimated via RNA-seq levels) of multiple CD8 + T-cell subsets in the lower iNMDeff tumors, across breast, kidney, lung, and skin cancers (Additional File 1: Fig. S28).

### NMD efficiency predicts progression-free survival in treated patients

As another test of the impact of NMD efficiency on patient outcomes, we stratified patients based on the type of treatment and focused on progression-free survival (PFS),

considering treatments separately and comparing the top 20% of individuals with the highest iNMDeff against the lowest 20%. For SKCM patients lacking available treatment data (Fig. 6C and Additional File 1: Fig. S29A for ASE using median iNMDeff), KM curves showed no significant difference (log-rank test) in PFS between the high and low iNMDeff groups (ETG $p = 0.73$, ASE $p = 0.77$).

However, a difference emerged among patients who received immunotherapy (along with a maximum of one chemotherapy treatment; Fig. 6D and Additional File 1: Fig. S29B for ASE), noting reduced PFS in the high iNMDeff group compared to the low iNMDeff group (ETG $p = 9.2e-2$ and ASE non-significant). The same trend was observed when discretizing at different thresholds (percentiles: 20 to 30, HR = 1.77–2.02, FDR = 32%). The borderline significance here might be attributed to the small number of SKCM patients treated with immunotherapy ($n = 73$). The TCGA pan-cancer analysis (includes all immunotherapy treated patients, $n = 184$) revealed a trend at the 20–25th percentiles (HR = 1.31–1.88, FDR = 19%) (Additional File 2: Table S17).

Next, we asked if this effect of NMD efficiency on survival is particular to immunotherapy, or it extends also to other types of therapies. Among SKCM patients treated with chemotherapy, but not immunotherapy (Fig. 6E, Additional File 1: Fig. S29C), a similar but stronger trend was observed (ETG $p = 0.17$ and ASE $p = 0.16$; ETG HR = 2.81, $p = 5.1e-2$). Efficient NMD is more strongly associated with poorer tumor response to chemotherapies compared to immunotherapies in melanoma.

We next asked if other cancer types apart from SKCM show associations between iNMDeff and response to chemotherapy in terms of PFS. Again, from within the patients receiving chemotherapy, we compared the high iNMDeff against the low iNMDeff group. Indeed, we identified four additional cancer types at an FDR-adjusted meta-$p < 10\%$ (results for individual percentiles given in Additional File 2: Table S18): pancreatic adenocarcinoma (PAAD, median HR = 2.16, median FDR = 3.8%), LGG (HR = 1.57, FDR = 4.5%), bladder cancer (BLCA_Basal_scc subtype, ETG: HR = 3.99, FDR = 3.4%), and ESCA_scc (ETG: HR = 2.5, FDR = 0.047%). No other cancer types were found with an opposite-direction trend for the same FDR threshold. The positive HR, highlighting the worse PFS in chemotherapy-treated patients whose tumors have higher NMD efficiency, extends to other cancer types. At a more relaxed FDR threshold of 25%, caution must be taken in interpreting hits as we identified additional cancer types, some with an opposite direction (i.e. better PFS, Additional File 2: Table S18).

### Association of iNMDeff with immunotherapy response in diverse cancer types

To validate the association of NMD efficiency with immunotherapy response, we analyzed independent datasets that had both RNA-seq and WGS/WES available: (i) Liu et al. [76], comprising 144 melanoma patients treated with anti-PD-1 immune checkpoint inhibitors (ICI); ii) Carroll et al. [80], involving 35 esophageal adenocarcinoma (EAC) patients who underwent first-line immunochemotherapy followed by chemotherapy; (iii) Motzer et al. [81], consisting of 886 patients with advanced renal cell carcinoma (RCC) treated with anti-PD-L1 or anti-PD-1. We applied our abovementioned iNMDeff proxy model (originally developed to predict cell-line NMD efficiency using global gene-level expression, see Methods) to estimate iNMDeff for each patient in these cohorts,

yielding iNMDeff for 120 patients in Liu-SKCM dataset, 33 in Carroll-EAC dataset, and 353 in Motzer-RCC dataset.

The KM curves for PFS (Fig. 6F–H and Additional File 1: Fig. S28D-F) demonstrated a consistent trend across the three immunotherapy studies for iNMDeff. In specific, this trend was evident in melanoma (Fig. 6F and Additional File 1: Fig. S29D; ETG $p = 0.16$ at 20th percentile, ASE $p = 0.21$, at 40th percentile), esophagus (Fig. 6G and Additional File 1: Fig. S29E; discretized at percentile 25th, ETG $p = 0.27$, ASE $p = 0.7$), and renal cancer (Fig. 6H and Additional File 1: Fig. S29F; ETG $p = 0.17$ at the 30th percentile, ASE $p = 0.15$ at the percentile 45th). The combined meta-analysis $p$-value (by Fisher's method) from the 3 studies would be $p = 0.13$ and $p = 0.27$, for ETG and ASE, respectively. When analyzing the data using the Cox proportional hazards model, including covariates such as sex, age, and TMB (see Methods) (Additional File 2: Table S19), the HR from all three studies trended in the same direction for both iNMDeff estimation methods in Liu-SKCM (ETG: HR = 1.68; ASE: HR = 1.37) and Motzer-RCC (1.29 and 1.23), and for the ETG method in Carroll-EAC (1.50; not supported in ASE).

Tumor mutation burden (TMB) is a well-established predictor of immunotherapy response, likely due to its association with neo-antigen content. In a predictive model, we combined TMB with iNMDeff, aiming to classify immunotherapy responses into responders versus non-responders (see Methods). This model was applied across the previous datasets TCGA-SKCM, Carrol-EAC, and Liu-SKCM, excluding Motzer-RCC due to missing TMB and treatment response data. Additionally, we included new datasets of immunotherapy-treated patients from Hartwig Medical Foundation[82] (urinary tract, $n = 23$, lung, $n = 26$ and melanoma, $n = 119$) and Riaz et al. (melanoma, $n = 48$) [83], totaling seven datasets for this analysis.

The TMB model's predictive accuracy was significantly enhanced by including iNMDeff, showing a 7.4% mean increase (for both ASE and ETG) in correctly classified patients compared to a TMB-only model, averaging across the datasets (Fig. 6I and Additional File 2: Table. S20). The area under the precision-recall curve (AUC) increased notably, from 0.704 (TMB-only) to 0.778 (TMB + iNMDeff), averaged across the datasets. Notably, this improvement manifested consistently but at variable strength across the different datasets: 1–9% in Riaz-SKCM, 11–27% in TCGA-SKCM, 1–2% in Hartwig-SKCM, 11–25% in Hartwig-Urinary, 3–5% in Hartwig-Lung and 1–10% in Carrol-EAC. Liu-SKCM did not exhibit improvement. Regarding other performance metrics, averaged across datasets: recall (sensitivity) increased from 0.89 to a range of 0.85–0.91 (for iNMDeff by ETG and ASE, respectively), and precision improved from 0.72 (TMB-only) to a range of 0.74–0.78 by including iNMDeff.

In conclusion, individual-level NMD efficiency can impact cancer patient survival: there was a trend of reduced PFS in immunotherapy-treated patients with high NMD efficiency, observed consistently across 3 cancer types. Furthermore, including iNMDeff improves predictive models of immunotherapy response drawing on TMB. For context, we note there is a strong prior that a globally more efficient NMD in a patient would hinder immunotherapy, based on recent work associating NMD inactivity upon individual indel mutations in a tumor and immunotherapy response [11, 44].

## Discussion

In our study, we quantified individual-level NMD efficiency—iNMDeff—in large-scale RNA-seq and germline WES data from normal tissues and cancers, totalling $\sim$ 27,000 samples. We considered both normal and mutated transcripts as tools to assess the variation in NMD efficiency across individuals and tissues (Fig. 2).

We found a significant variability of iNMDeff across normal tissues and cancer types (Fig. 3). This suggests tissue-specific regulation of the NMD pathway, which is broadly conserved between cancers and healthy tissues. Gastrointestinal tissues showed higher iNMDeff, and within them the hypermutated MSI colon and stomach cancers were among the ones with highest NMD efficiency. This is in agreement with a reported overexpression of NMD factors in MSI primary colon tumors and that chemical inhibition of NMD caused a reduction in MSI tumor growth in mouse [40]. Supporting this, studies have shown higher NMD pathway scores in MSI-H versus MSS tumors [84], while *UPF1* overexpression in CRC has been linked to metastasis and inhibition of apoptosis [85] and chemoresistance to oxaliplatin [86], with implications for patient survival. Together, this suggests that colon tumors, especially MSI (speculatively, due to higher frameshift mutation burden), necessitate a higher NMD efficiency to survive, supporting that targeting the oncogenic activity of NMD could be a viable, personalized treatment strategy for some cancers [46]. Whether SNV/indel burden or other genomic features predict responses of an individual tumor to NMD inhibitors remains to be determined in future work. We acknowledge the need for additional validation of increased NMD in digestive tissue tumors through experimental models or analysis in additional cancer datasets.

In contrast to the digestive tract, we observed that reproductive and nervous system tissues exhibited lower iNMDeff than other tissues, particularly so for neurons therein (as estimated from differential analysis of brain regions and glioma subtypes). A previous study in mice monitored NMD activity through the expression levels of a single gene, *MEN1* [23], across seven tissues including brain, testis, ovary, and heart. However, our comprehensive analysis across many human tissues suggests that considering multiple genes and mutations therein as markers is helpful to reach robust conclusions about tissue-specific NMD activity.

A recent study of ASE in GTex healthy tissues reported only modest NMD differences between tissues; however, it did rank the brain samples at the low end of NMD efficiency [57], aligning with our results that were seen also in the ETG method, and also in TCGA with ASE and ETG estimates. This might stem from the brain's unique NMD regulation during development [87], for instance through the known role of miR-128's targeting of *UPF1* to inhibit NMD, thus promoting neural differentiation, a mechanism persisting into adulthood [88]. In addition, other NMD-regulatory miRNA circuits have been reported in the brain [89, 90]. Together, ours and these previous findings support a reduced mRNA surveillance by NMD in mature neurons compared to other cell types. Extending this idea, developmental roles of NMD in other tissues such as blood [91], muscle [92], liver [93], embryogenesis [94], and spermatozoa [95], might also contribute to the variability of NMD efficiency we observe across adult tissues. Lastly, we did observe differences in NMD efficiency between some cancer types or subtypes and their matched normal tissues, suggesting a potential shift in NMD efficiency during transformation. For instance, while breast cancer tumors of the normal-like and basal

subtypes, and their corresponding normal tissues align in NMD efficiency, the other breast cancer subtypes HER2, Luminal A, and Luminal B display higher iNMDeff; this is consistent with reports of higher *UPF1* expression in these subtypes [39]. This variation also underscores there may exist differences between cancer subtypes in NMD.

In addition to variation between tissues, variation in NMD efficiency between individuals is significant. Some of this is explained by association with somatic genetic variants in genes *TLX1* and *CDH1* (Additional File 1: Fig. S12); the latter is a known risk gene in hereditary diffuse gastric cancer and hereditary breast cancer [96–99], and it is tempting to speculate that NMD has a mediating role in this [100]. We also extended the somatic variant analysis to CNAs, by a bespoke sparse PCA-based method to control for genetic linkage in CNA association studies, here applied to NMD efficiency variability (Fig. 4). Cancer-type-specific significant findings were limited, which can be attributed to smaller sample sizes in individual cancer type analyses, thus our analyses focused on shared rather than cancer-specific mechanisms. Three pan-cancer CNA-PCs reflecting large-scale gains were robustly associated with reduced NMD activity, most strongly CNA-PC3 and 86 located in regions 1q and 1q21-23.1. The 1q gain is found in ∼25% of cancers and may be selected through increasing dosage of the *MDM4* oncogene phenocopying *TP53* mutation [66], or of *MCL-1* antiapoptotic factor [101] or via upregulation of Notch genes [102]. This chromosome arm however also contains genes encoding NMD factors *SMG5*, *SMG7*, the EJC gene *RBM8A*, and NMD-related genes *INTS3* and *SF3B4* (a known splicing factor oncogene), which we identified as associating with NMD efficiency upon CNA gain; additionally, we identified 6 other genes involved in various RNA metabolic processes with *PMF1* (transcriptional cofactor [103]*)*, *GON4L* (a nuclear negative transcription factor [104]), *PRPF3* (spliceosome component [105]), and *VPS72* (chromatin remodeler [106]) being particularly notable.

In addition to 1q gains, further the chromosome 2q gains (found in CNA-PC52) that we find associated with NMD efficiency are interesting to follow-up, since this arm encompasses candidate genes *CWC22* and *SF3B1* implicated in the splicing process, whose dysregulation may plausibly affect NMD. It was shown that mutations in the spliceosome genes *SF3B1* and *U2AF1* render cancer cells more sensitive to NMD inhibition in a synthetic lethality fashion [30, 107, 108] adding confidence that CNA of this same gene might disbalance NMD. A recent study also showed that depleting individual spliceosome components in cell lines increased NMD-targeted mRNA levels, suggesting their importance for NMD function (especially *AQR*, *SF3B1*, *SF3B4*, and *CDC40*) [109]. They proposed two mechanisms: direct interference through impaired EJC assembly and compromised EJC deposition on exonic RNA within the spliceosome, or indirect effects where increased novel NMD substrates overwhelm the pathway—indeed, they found a direct correlation between the amount of novel NMD substrates detected and the degree of NMD inhibition observed. Our study reveals a more complex picture in cancer, where chromosomal gains containing various splicing factors (1q gain: *SF3B4, PRPF3*, 2q gain: *CWC22, SF3B1*) and other NMD-related genes are associated with reduced NMD efficiency. Unlike single-gene depletion studies, these chromosomal gains affect multiple genes simultaneously, potentially creating combinatorial and paradoxical effects on NMD efficiency.

In line with this, a previous study identified significant enrichment of germline copy number variants of NMD genes in neurodevelopmental disorder patients [110], including *UPF2*, and also EJC genes including the chr 1q-located *RMB8A*. Interestingly, both copy-number gains and also losses of the same genes, for example, *UPF2* and *RMB8A*, were linked to the phenotype, implying that NMD and EJC gene dosage disbalances impact NMD function and disease pathology.

As for somatic variation, it was suggested that tumor evolution favors mutations and CNA co-occurring in NMD factor genes [20]. Interpreting the effects of NMD factor genetic alterations on NMD efficiency and downstream phenotypes (e.g. cancer patient survival) requires future work on modeling an regulatory network impacting NMD factors, including negative feedback, and its dynamics [52]. This feedback mechanism, which varies by cell type and developmental stage, would serve to mitigate disruptions in NMD factor genes; our data suggests that this mechanism does not fully counterbalance the effects of chromosome gains or co-occurrence mutations that affect NMD factors and/or NMD-related genes such as spliceosome genes.

In addition to somatic variants in tumors, also the rare deleterious germline variants can affect NMD efficiency in at least the thyroid and brain. We identified 10 genes with diverse functions, potentially affecting NMD efficiency indirectly (Fig. 5). For instance, *CRTC1*, primarily active in the brain subregions, acts as a transcriptional activator [111]. *NUP153* is involved in mRNA nuclear export and neural progenitor cell regulation [112]. *KDM6B*, a histone demethylase, is pivotal in gene expression regulation by removing the Polycomb repressive histone marks [113, 114], and also linked to neurodevelopmental disorders [115]. We also identified genes whose mechanistic links to NMD are not obvious; replication in additional cohorts would be the sensible next step. A recent study identified genetic variants that influence NMD-targeted transcripts and their NMD decay efficiency across healthy GTex tissues [116], with more than 50% of NMD-QTLs identified in the brain in particular. Interestingly, *SF3B4*, one of our identified candidates from somatic associations, emerged as the gene containing the highest number of germline NMD-QTLs.

Next, our analysis suggests various associations of differential NMD efficiency to tumor evolution (Fig. 6A): shaping selective pressures on tumor suppressor genes, with prominent effects on *SMAD4*, *TP53*, and *APC*. In a cancer-type specific analysis, we observed a similar trend in hypermutated tumors like stomach MSI, colon MSI, POLE-mutant uterus, and also lung cancers, where these TSGs were more frequently mutated. In these individuals/tumors which better utilize NMD mechanisms to ablate the activity of these tumor suppressor genes, applying NMD inhibition therapeutically—alone or in combination with readthrough agents [117–119]—may be promising in restoring tumor suppression activity to treat cancer. As a side observation, genes with NMD-evading PTC variants do also exhibit positive selection, but, importantly, there is not a significant difference between tumor samples with high and low iNMDeff (Fig. 6A). This emphasizes that the effect of inter-individual NMDeff variation on the somatic selection affects only driver genes with NMD-triggering PTC variants. The positive selection of NMD-evading nonsense mutations is explained because these truncating mutations can be advantageous despite evading NMD, either through complete loss-of-function effects

even without mRNA degradation, or through dominant-negative mechanisms of truncations [74, 75].

In support of the idea that manipulating NMD may treat some cancers, we observed that the naturally variable NMD efficiency is associated with cancer survival phenotypes (in accord with our recent population genomic study arguing that NMD modulates phenotype severity of various heritable genetic diseases [11]). We found that high NMD efficiency is associated with improved or worse outcomes depending on cancer type (Additional File 1: Fig. S27), consistent with a previously proposed complex, multifaceted roles of NMD influencing in cancer [30, 77]. Our analysis particularly highlights the effects of the global NMD efficiency on responses to immunotherapy and chemotherapy (Fig. 6C–H). Previous studies suggested that NMD-evading status of individual mutations [11, 44] does predict response to immunotherapy; here we present evidence of an additional role of the global, individual-level NMD efficiency in this, which we observed across various patient cohorts and cancer types. A plausible mechanism is rooted in NMD's role in silencing the presentation of neoantigens. Indeed, this is consistent with our observation that predictive models containing TMB and iNMDeff have higher accuracy in predicting responses (Fig. 6I). We do note that the iNMDeff associations with efficiency of conventional chemotherapy, which may or may not be mediated by immune mechanisms, are stronger than immunotherapy. Speculatively, NMD may confer a generally increased tumor cell fitness in the context of occurrence of deleterious, proteins that arose via somatic mutation (as shown before for one example, the HSP110DE9 chaperone mutant [40]), conferring additional ability to withstand various challenges to cancer fitness. In line with this, transient pre-treatment with NMDI-1 inhibitor prior to administering doxorubicin demonstrated increased cell death compared to doxorubicin alone [120]. This suggests that reduced NMD efficiency may predispose tumor cells to an enhanced cytotoxic response to chemotherapy, consistent with generally better survival outcomes for individuals bearing tumors with inherently low NMD activity.

We acknowledge that our ETG and ASE metrics represent indirect measures of NMD efficiency rather than direct assessments of the NMD degradation process. Since RNA-seq provides static snapshots of transcript abundance rather than dynamic measurements of transcription and degradation rates, our calculated ratios may be influenced by tissue-specific differences in transcriptional activity of NMD-targeted genes. However, PRO-seq analysis of nascent RNA transcription from three cell lines showed no significant differences in transcription initiation rates between NMD and non-NMD target transcripts, including when stratified by brain expression levels (Additional File 1: Fig. S3B), reassuring that our ETG measurements indeed reflect NMD activity rather than transcription rates (see Additional File 3: Text S1).

In summary, our study reveals extensive variation in NMD efficiency across individuals, tissues, and tumors, resulting in part from specific genetic alterations, and underscores the potential of NMD efficiency estimates as a predictive biomarker for cancer treatment using immunotherapy and/or NMD inhibition. Future studies incorporating functional assays and direct measurements of mRNA decay kinetics would be valuable to validate these findings and assess the broader applicability of our approach as a potential biomarker.

## Conclusions

This comprehensive analysis of individual-level NMD efficiency across ~ 27,000 samples reveals substantial tissue-specificity and inter-individual variation in NMD activity. Digestive tissues, particularly MSI cancers, exhibit elevated NMD efficiency, while reproductive and nervous system tissues show reduced activity. This variation is partially driven by genetic alterations, including copy number gains affecting multiple NMD and splicing factors, and rare inherited germline variants. Variable NMD efficiency shapes tumor evolution by influencing selective pressures on tumor suppressor genes and predicts patient responses to both immunotherapy and chemotherapy. These findings position individual-level NMD efficiency as a promising biomarker for personalized treatment strategies, particularly for identifying cancer patients who may benefit from NMD inhibition therapy. Future validation through direct measurements and clinical studies will be essential to realize the therapeutic potential of these findings.

## Methods

### Sequencing data

We downloaded matched tumor and normal whole-exome sequencing (WES) *bam* files, along with tumor RNA-seq *fastq* files, from The Cancer Genome Atlas (TCGA) consortium [121] via the Genomic Data Commons (GDC) Data Portal (https://portal.gdc.cancer.gov/[122]. This dataset encompassed 9,766 matched samples across 33 distinct cancer types. Copy number alteration (CNA) data, including both arm-level and focal-level, was obtained from the Broad Institute's GDAC Firehose portal (https://gdac.broadinstitute.org/[123]. From Genotype-Tissue expression (GTex) project [124], we directly obtained WES *VCF* files and tissue-specific transcript-level RNA-seq counts data, accessed through dbGaP (via AnVIL, https://anvilproject.org/), v8 (dated 05–06-2017). This dataset included 979 individuals and spanned 56 different tissues, amounting to a total of 17,382 samples, with an average of 19 tissue types per individual. We also downloaded the allele-specific expression (ASE) counts data for 838 individuals. RNA-seq and CNA data of 1,450 human cell lines from the Cancer Cell Line Encyclopedia (CCLE)[65] through DepMap portal (https://depmap.org/portal/download/all/) was also downloaded.

### Clinical and other data sources

TCGA cancer cluster subtypes, determined based on mRNA non-matrix factorization (NMF), were sourced from the Broad Institute's GDAC Firehose portal (https://gdac.broadinstitute.org/)[123]. In addition, we acquired estimates of leukocyte fraction [112] and tumor purity [113] for all samples within the TCGA dataset. Patient clinical data and metadata were obtained directly from the GDC portal (https://portal.gdc.cancer.gov/)[122]. For progression-free survival (PFS) information, we relied on data available through cBioPortal (https://www.cbioportal.org/)[125]. Immune infiltration proportion of CD8 + cytotoxic T-cells, as estimated via "*quanTIseq TIL10*" signature, was downloaded from The Cancer Immunome Atlas (https://tcia.at/home)[126].

### TCGA cancer type stratification into subtypes

The classification of TCGA cancer molecular subtypes was meticulously gathered using functions from the *TCGABiolinks* R package, namely *PanCancerAtlas_subtypes* (only for BRCA) and *TCGAquery_subtype*, or directly from the referenced articles in certain cases. The classifications are as follows:

- Breast invasive carcinoma (BRCA): Classified according to the PAM50 mRNA profiling into five subtypes: Basal, Normal-like, Luminal A (LumA), Luminal B (LumB), and HER2-enriched [127].
- Sarcoma (SARC): Based on histological subtypes [128], with specific subtypes such as soft tissue leiomyosarcoma (STLMS) and uterine leiomyosarcoma (ULMS) being aggregated into the "Muscle" category, and Dedifferentiated liposarcoma (DDLPS) classified under "Fat."
- Bladder urothelial carcinoma (BLCA): Utilized mRNA clustering to divide samples into Basal squamous cell (Basal_scc), Luminal papillary (Lum_pap), and Luminal infiltrated (Lum_inf) subtypes [129].
- Head and neck squamous cell carcinoma (HNSC): Categorized based on RNA-seq data into HPV positive or negative groups [130]. Samples with no category were denoted as "NA".
- Colon adenocarcinoma (COAD), stomach adenocarcinoma (STAD), and uterine corpus endometrial carcinoma (UCEC): These cancers were classified into Microsatellite Instability (MSI) or Microsatellite Stable (MSS) [131–133]. For UCEC, an additional distinction was made for samples with SBS10a/b mutational signatures extracted from WES SigProfiler (somatic mutations in *POLE* gene), predominantly within the MSS subgroup [134].
- Esophageal carcinoma (ESCA): Classified into esophageal squamous-cell carcinoma (scc) and adenocarcinoma (ac) [135].
- Glioblastoma multiforme (GBM): Subdivided based on the Glioma CpG island methylator phenotype into High (G-CIMP-high) and Low (G-CIMP-low), alongside methylation cluster 6 (LGm6-GBM), Classic-like, and Mesenchymal-like [61].
- Lower grade glioma (LGG): Classified into G-CIMP-high, G-CIMP-low, Classic-like, Mesenchymal-like, PA-like, and 1p/19q codeletion (Codel) [61].

### Alignment and quantification of RNA-seq and ASE

In adherence to GTex guidelines (https://github.com/broadinstitute/gtex-pipeline/tree/master/rnaseq)[136], we processed TCGA RNA-seq data by aligning the reads to the human genome *GRCh38.d1.vd1*, utilizing *STAR* v2.5.3a [137]. Subsequent transcript-level quantification was conducted using *RSEM* v1.3.0 [138]. For allele-specific expression (ASE) analysis, we retained RNA-seq reads with a minimum mapping quality of 255. To correct for allele-mapping bias, we employed *WASP* for adjustment [139]. Additionally, an unbiased approach for the removal of duplicate reads was implemented using a custom *WASP* script. This script randomly discards duplicate reads, ensuring that the removal process is independent of the read scores.

**Variant calling and annotations**

Variant calling in TCGA was done using *Strelka2* v2.9.10 [140]. For somatic WES, normal and tumor bam files were compared utilizing the *configureStrelkaSomaticWorkflow.py* script, employing the "—exome", "—tumorBam," and "—normalBam" flags. In the case of germline WES, the analysis was conducted using only normal bam files, applying the *configureStrelkaGermlineWorkflow.py* script with "–exome" and "–bam" flags. For RNA-seq data, aimed at obtaining ASE counts, we again used normal bam files, this time with the *configureStrelkaGermlineWorkflow.py* script and the "–exome", "–rna," and "–forcedGT" flags. The latter flag forces genotype calling to conform to previously identified genotypes from WES. Single-nucleotide variants (SNVs) and insertions-deletions (INDELs) were called separately in all instances and subsequently merged. We annotated all resulting *VCF* files using ANNOVAR [141] (2020–06-07 version), referencing GENCODE v26 (ENSEMBL v88), and incorporated the *gnomad211_exome* database to acquire population minor allele frequencies (MAF) from gnomAD [142]. In the case of GTex, as previously mentioned, the WES *VCFs*, RNA-seq counts data, and ASE counts were directly downloaded. The only additional step undertaken was a liftover to convert genome coordinates from GRCh37 to GRCh38. Regions with poor alignment between the genome versions were systematically identified and removed to ensure liftover accuracy [143]. Subsequently, the variants were annotated in the same manner as in TCGA, as described earlier.

**Obtaining NMD target gene sets from various studies**

We created different gene sets of NMD targets from a range of studies employing both experimental and computational methods, to use in the estimation of ETG iNMDeff:

(1) Tani H. et al. [6] identified *UPF1* NMD targets categorized into three groups: (A) 248 indirect targets with over twofold upregulation but stable decay rates; (B) 709 direct targets with unaltered expression levels but over twofold increase in decay rates; and (C) 76 bona-fide direct *UPF1* targets with both over twofold upregulation and extended decay rates. We excluded the genes from group B in subsequent analyses due to their lower confidence as NMD targets.

(2) Colombo M. et al. [5] provided *UPF1*, *SMG6*, and *SMG7* specific NMD target genes. For *UPF1* targets, we selected those with a "meta_pvalue" < 0.05, resulting in 2,725 genes. For *SMG6* targets, we kept those with "UPF1_FDR" < 0.05 and "meta_SMG6" < 0.05, yielding 1,780 genes. Likewise, for *SMG7*, we required "UPF1_FDR" < 0.05 and "meta_SMG7" < 0.05, leading to a List of 1,047 genes.

(3) Karousis E. D. et al. [13] utilized nanopore sequencing and compiled a gene set pairing NMD target transcripts with controls, encompassing 1,911 genes.

(4) Courtney et al. [51] identified a set of 2,793 NMD target genes.

(5) Schmidth S.A et al. [14] Listed 233 *SMG6* specific target genes.

(6) The ENSEMBL gene annotation file tags some genes as "nonsense_mediated_decay."

With all this, we created distinct NMD gene sets for each study, with the exception of Schmidt. These sets are named: "NMD Tani," "NMD Colombo," "NMD Karousis,"

"NMD Courtney," and "NMD Ensembl." Additionally, we constructed specific gene sets for "SMG6" and "SMG7" NMD targets. For "SMG6," we derived the set from the intersection of *SMG6* NMD targets identified in both Colombo's and Schmidt's studies. For "SMG7," we exclusively utilized the *SMG7* NMD target genes from Colombo's study. Additionally, we overlapped genes found in at least 2 studies to compile a "NMD Consensus" gene set, and the complete list of genes across all studies as "NMD All" gene set. Lastly, we derived negative control gene sets, by selecting genes with and without NMD-triggering features (see the following section) in their transcripts and excluding the NMD target gene sets from above: "RandomGenes with NMD features," and "RandomGenes without NMD features."

### Predicting NMD-triggering features for the selection of endogenous NMD targets and controls (ETG method)

For each gene within our NMD gene sets from the ETG method, we classified its transcripts into NMD targets and controls based on computationally predicted NMD-triggering features (see schematic on Additional File 1: Fig. S1B), as per ENSEMBL definition. This classification relied on two primary features identified in the literature: (i) the presence of an upstream open-reading frame (uORF) in the 5' untranslated region (UTR) [15, 16] that does not overlap the CDS, and (ii) the existence of at least one splice site (or exon-junction complex, EJC) in the 3' UTR, located more than 50 nt downstream of the termination codon [12, 17]. We also considered the GC content of the 3' UTR, known to influence transcript degradation by NMD [18].

To delineate NMD target-control pairs for each gene, our specific criteria is shown in Additional File 1: Fig. S1B. In summary, the criteria for NMD target transcripts were as follows:

(i)   Presence of both a start codon and a stop codon.
(ii)  A median transcript expression ($\log(TPM)$) across pan-cancer (TCGA) of $\geq 1$.
(iii) Presence of either a 3'UTR splice site or $\geq 2$ uORFs (both having a minimum length of 30 nt).
(iv)  If multiple transcripts meet the above criteria, the one with the highest 3' UTR GC content is selected.

For control transcripts, the requirements were:

(i)   Presence of both a start codon and a stop codon.
(ii)  A median transcript expression ($\log(TPM)$) across pan-cancer (TCGA) of $\geq 3$.
(iii) Absence of any NMD-triggering features.

Lastly, for a transcript to be classified as an NMD target, and a transcript to be its paired control, the ratio of their expressions must be $\leq 0.9$. This threshold ensures that the selected NMD target transcripts exhibit lower expression levels compared to their corresponding controls, aligning with the expected impact of NMD-triggering features on transcript stability.

To enhance the accuracy of this selection and mitigate the inclusion of potential false positives, we utilized data from cell lines with *UPF1* knockdown (KD) via an inhibitor, as reported in four different studies: HeLa cells [5, 53], HepG2 [54], K562 [54]. For each

selected transcript pair, we stipulated that the expression ratio in the three wild-type cell lines must average $\leq 0.9$, reflecting the anticipation of high NMD efficiency in the control cell lines. To acquire the necessary RNA-seq count data, we conducted our own alignment and quantification using the same tools and versions previously outlined for the TCGA and GTex cohorts (as detailed in the section "Alignment and quantification of RNA-seq and ASE").

It is important to note that not all selected transcript pairs were utilized in our ETG iNMDeff, due to the filterings performed afterwards (see section "Filterings for the ETG method").

### Using NMD rules for predicting NMD-triggering vs NMD-evading PTCs (ASE method)

For the ASE method, we determined whether a premature termination codon (PTC), resulting from nonsense or indel mutations, is likely to undergo NMD based on its location within the coding sequence (CDS) of the transcript. For indels, we predicted the potential generation of a downstream PTC and identified its exact genomic location with a custom script. To categorize these PTCs as either NMD-triggering or NMD-evading, we relied on established NMD rules [8], except the long-exon rule:

 (i)   The 55nt-rule: PTCs positioned $\leq 55$ nt downstream from the last base of the penultimate exon are considered NMD-evading.
 (ii)  The last exon rule: PTCs located within the last exon are deemed NMD-evading.
 (iii) The start-proximal rule: PTCs situated within the first ~200 nucleotides of the transcript are classified as NMD-evading.

In scenarios where a transcript had a splice site in the 3'UTR, a PTC that would typically be classified as NMD-evading under the 55nt or last exon rule was instead considered NMD-triggering. This reclassification is due to the presence of an additional EJC located downstream of the PTC.

Using this classification, we generated three distinct NMD variant sets for each individual: "NMD-triggering PTCs," "NMD-evading PTCs," and "Synonymous." These sets were specifically designed to accurately estimate the NMD efficiency for each individual, along with its two negative control sets.

### Filterings for the ETG method

To ensure an accurate estimation of ETG iNMDeff, all filtering processes were conducted on a per-individual basis and on all NMD gene sets separately. The following criteria were applied in both TCGA and GTex cohorts:

- We retained transcripts exhibiting a $\log_2$(raw counts) $\geq 1$ in at least 50% of individuals within the specific cancer type of the individual being analyzed. If both transcripts in any NMD target-control pair failed to meet this threshold, the entire pair was excluded.
- Non-coding transcripts were excluded from the analysis.
- We discarded transcripts that overlapped with any type of somatic (only in TCGA) or germline truncating mutations (such as nonsense, start loss, nonstop mutations,

inframe insertions and deletions, frameshift insertions and deletions, and splice site mutations) as well as those affected by CNAs (only on TCGA).

- A minimum of 2 pairs of NMD target-control transcripts was required for the estimation of an individual's NMD efficiency.
- For all NMD gene sets, except from the "NMD Consensus" set, we randomly selected up to 50 pairs of NMD target-control transcripts. This approach was adopted to streamline the computational process.

**Filterings for the ASE method**

Similar to the ETG method, in the ASE approach, all filterings were meticulously carried out on a per-individual basis and for each NMD variant set ("NMD-triggering PTCs," "NMD-evading PTCs," and "Synonymous"). The following criteria were applied in both TCGA and GTex cohorts:

- We retained only those germline heterozygous variants that met the "PASS" threshold in the *VCF* file.
- A minimum total read coverage (sum of WT and MUT alleles count) requirement was set at 5 for nonsense and synonymous mutations, and at 2 for indels.
- Variants with a MAF > 20% were excluded to avoid confounding effects in the analysis. Common variants are known to exhibit lower NMD efficiency, likely due to evolutionary selection [8, 11, 27, 57, 144].
- Variants located in non-coding transcripts or in transcripts consisting of a single exon were excluded.
- Variants in 331 genes identified as undergoing positive selection were removed [145].
- For variants in genes experiencing negative selection, we employed the loss-of-function observed/expected upper bound fraction (LOEUF) estimate [146]. Genes in the lowest first percentile based on this score, representing the 10% most constrained genes, were excluded from the analysis.
- We excluded variants co-occurring with any type of somatic truncating mutation (such as nonsense, start loss, nonstop mutations, inframe insertions and deletions, frameshift insertions and deletions, and splice site mutations) in TCGA. This also included variants affected by CNAs in TCGA.
- Specifically for the "NMD-triggering PTCs" variant set, further exclusions were made:

    ○ NMD-evading PTCs (and vice versa).
    ○ Frameshift variants that do not predict a downstream PTC.

- For the "Synonymous" variant set:

    ○ We excluded variants overlapping any gene from our "NMD All" gene set to avoid confounding the analysis.

- In GTex only, if the variant only appeared in one tissue, then it was considered a somatic variant, and not germline, thus discarded.

- A minimum of 3 germline variant PTCs was required for the estimation of an individual's NMD efficiency.
- Additionally, to optimize computational efficiency, we capped the number of variant pairs used in the analysis randomly sampling a maximum of 100.

### Quantification of individual NMD efficiency (iNMDeff) by a negative binomial regression

To estimate individual NMD efficiency (iNMDeff), we employed Bayesian generalized linear models, fitting a negative binomial distribution using "*Stan*" [147]. This modeling was implemented via the'stan_glm'function from the *rstanarm* R package [148], with 'family = neg_binomial_2' specified as the parameter.

For the ETG method, the model is applied, pooling all filtered transcripts together within a sample, for each of the 11 NMD gene sets (includes the negative control) separately, as follows:

$$\text{RawTranscriptExp} \sim \text{NegBin}(\mu, \theta)$$

$$\log(\mu) = \beta_0 + \beta_1 \cdot \text{NMDtarget} + \beta_2 \cdot \text{geneID} + \beta_3 \cdot \text{transcriptLength}$$

where:

- RawTranscriptExp: raw RNA-seq read counts for each transcript (one row per transcript).
- *NMDtarget*: binary indicator: (1) transcript is predicted to be an NMD target; (0) control of the matched pair.
- geneID: ENSEMBL gene identifier. Included as categorical variable to adjust for between-gene variability.
- transcriptLength: Total length of the transcript (sum of exon lengths, in bp). This adjustment is necessary to address the potential bias of more frequent read clustering in larger transcripts.

By comparing each NMD target transcript against its paired control from the same gene, we establish an internal control. This approach effectively accounts for potential confounders affecting trans-gene expression levels. For instance, CNAs or transcription factors might alter the expression of one transcript without affecting the other. Such discrepancies are particularly pertinent if comparing transcripts across different genes. Although we already exclude genes overlapping with CNAs, this internal control further ensures the robustness of our analysis against such confounding factors.

For the ASE method, the model is applied, pooling all filtered germline PTCs together within a sample, for each of the 3 NMD variant sets (includes the negative control) separately, as follows:

$$\text{RawAlleleExp} \sim \text{NegBin}(\mu, \theta)$$

$$\log(\mu) = \beta_0 + \beta_1 \cdot \text{NMDtarget} + \beta_2 \cdot \text{geneID}$$

where:

- *RawAlleleExp*: allele-specific expression raw counts of the germline PTC (one row per allele).
- *NMDtarget*: Binary indicator: (1) mutated/alternative allele, thus, predicted to be NMD target; (0) wild-type or reference allele, thus, our control, within the selected pair.
- *geneID*: ENSEMBL gene identifier, included to adjust for between-gene variability.

In both NMD methods, the coefficient assigned to the *NMDtarget* variable serves as our estimate of iNMDeff for a specific NMD variant or gene set, as well as its corresponding negative control. We reversed the direction of the raw coefficient values, so that now higher coefficients indicate greater iNMDeff, and lower coefficients indicate reduced efficiency. The final interpretation is that it is a negative log (base e) ratio of the raw expression levels of the NMD target transcripts (ETG) or MUT alleles (ASE) divided by the control transcripts (ETG) or WT alleles (ASE). For a more intuitive interpretation, one could exponentiate the coefficient to derive the ratio between NMD targets and controls. In this context, ratios above 1 would suggest lower NMD efficiency, while ratios below 1 would indicate higher NMD efficiency. It is important to note that for our analysis, we utilized the original log coefficients rather than these exponentiated ratios.

Interestingly, both iNMDeff methods suggest a possible differential NMD activity across tissues or individuals, and the differences in their NMD estimates might reflect differently active NMD subpathways or other types of biological or technical variation. The ETG method, based on transcript-level expression, offers a high data point count per individual, crucial for estimating iNMDeff accurately, but faces challenges like confounding factors such as CNAs and varying expression levels among transcripts of a gene, which we attempted to address with rigorous filtering and pairing NMD-target transcripts with appropriate controls. Conversely, the ASE method, while robust to confounders due to its internally controlled quantification of the two alleles within the same transcript, is limited to more highly expressed genes and thus provides fewer data points, leading to potential noise issues (see Additional File 3: Text S2 for discussion). We used both methods as independent approaches, ensuring a more reliable analysis of NMD efficiency across individuals.

### Validation and robustness of ETG iNMDeff estimates
#### *Using cell lines with UPF1 knockdown*
We validated our ETG method by analyzing NMD efficiency in *UPF1* knockdown (KD) cell lines, as *UPF1* is a central NMD factor, compared to their isogenic wild-type controls. RNA-seq data was obtained from three cell lines: HeLa (GSE152435, $n=5$; GSE86148, $n=3$) [5, 53], HepG2 (GSE88148, $n=2$; GSE88466, $n=2$) [54], and K562 (GSE88140, $n=2$; GSE88266, $n=2$) [54], sourced from different studies processing the raw data using the same methodology as our main analysis. This included short-read mapping to the human genome using *STAR* and transcript-level quantification using *RSEM*, with versions and parameters identical to TCGA/GTex processing (as detailed in the section 'Alignment and quantification of RNA-seq and ASE').

For NMD efficiency estimation, we applied our NMD Consensus gene set, comprising 130 target-control gene pairs (260 total transcripts, see Additional File 2: Table S4), and used negative binomial regression to calculate cell line NMD efficiency (cNMDeff). This approach paralleled our iNMDeff estimation methodology (see section "Quantification of individual NMD efficiency (iNMDeff) by a negative binomial regression").

By using external datasets from controlled experiments, we ensure that confounders (e.g., tissue, subtype, environment, and genetic background) do not play a role. As an additional analysis to confirm the robustness of the ETG Consensus NMD gene set, we iteratively removed one gene at a time from our NMD Consensus set and repeated the analysis with the 129 remaining genes. Therefore, this process was repeated 130 times (once for each gene in the set).

### Differential transcription rates as a confounding factor using PRO-seq

To investigate potential confounding effects of transcription rates on NMD efficiency estimates, we analyzed PRO-seq data from three cell lines: SH-SY5Y (neuroblastoma, GSE214243, $n=2$) [55], A549 (lung, GSM5169137, $n=1$) [56], and U2OS (bone, GSM5169140, $n=1$) [56]. PRO-seq, which captures nascent RNA transcripts, provides nucleotide-resolution measurements of active RNA polymerase II transcription across the genome.

We quantified promoter activity by measuring PRO-seq signals near the transcription start site (TSS) and 5'UTR (as per ENSEMBL v88). For A549 and U2OS, we used processed dREG scores [149], while standard PRO-seq scores were used for SH-SY5Y. The analysis focused on our ETG Consensus gene set, comparing PRO-seq signals between NMD-target transcripts and their matched non-NMD transcripts from the same genes.

The peak analysis workflow began with overlapping PRO-seq peaks to transcripts in our ETG Consensus set. Multiple peaks frequently overlapped the same transcript or both NMD-target and control transcripts due to their proximity or shared TSS. We selected peaks overlapping TSS or within 5'UTR (or if the end peak was at 1000nt distance), retaining only those within 1000nt of the TSS. Among these peaks, we selected the one with the closest distance to the TSS for each transcript. To determine TSS-peak distances, we identified peak summits representing maximum coverage positions. For A549 and U2OS, we used pre-calculated Height Summit positions, while for SH-SY5Y, we used peak midpoints due to their narrower peak width ($\sim$12nt compared to 499–530nt in A549 and U2OS). Peaks with the minimum score for SH-SY5Y were removed. We excluded peaks with identical summit positions between NMD and non-NMD transcripts of the same gene. Finally, scores were log-transformed and scaled to enable comparison across cell lines.

To specifically address brain-associated differential NMD efficiency, we stratified transcripts into brain-upregulated and non-brain-upregulated categories based on transcript expression from GTex and TCGA datasets. For classification, mean gene TPM values were compared between merged normal brain subregions and all other tissues (GTex), or between LGG/GBM brain tumors and other cancers (TCGA). Brain-upregulation was determined using Mann–Whitney $U$ test (alternative$=$greater, $p<0.05$).

## Randomization tests for iNMDeff variability

### Inter-Tissue iNMDeff variability deviation (ITNVD) test

This test consists in shuffling the data 2000 iterations, to see if the iNMDeff differences observed between tissues could have happened by chance (see also Fig. 3A). For each randomization, we calculated the variability (standard deviation, SD) of the tissue-specific median iNMDeff values, to obtain a single variability score per iteration. We compared the median of the 2000 calculated variability scores to the observed variability in the original, unshuffled data, and termed the difference between both values as the "Inter-tissue iNMDeff Variability Deviation (ITNVD)," which serves as a measure of effect size. If there is no variability, the value should be close to 0. We then calculated a *p*-value to assess the statistical significance of this effect size.

$$p - value = \frac{sum(RandVarScores > ObsVarScores) + 1}{TotalNumIterations}$$

### Tissue iNMDeff deviation (TND) test

This test compares the observed tissue-specific median iNMDeff against a randomized median iNMDeff (note that these medians were essential in deriving the ITNVD score discussed above), as illustrated in Fig. 3A. The resulting difference between them is termed "Tissue iNMDeff Deviation (TND)." We calculated this deviation score 2000 times, once for each randomization of the data, to create a comprehensive distribution of these deviations. Essentially, a positive value indicates that the tissue has a higher iNMDeff than expected by chance, and vice versa. As before, we then calculated a *p*-value to assess the statistical significance of this effect size.

$$p - value = \frac{sum(RandMedianINMDeffScores > ObsMedianINMDeffScore) + 1}{TotalNumIterations}$$

## Tissue variability of iNMDeff using linear models

As an alternative approach to assess the variability of iNMDeff across different tissues and cancer subtypes, we employed a linear modeling approach. This model aimed to predict iNMDeff values, based on a set of explanatory variables. These included demographic factors (age, sex), technical aspects (RNA-seq sample library size), and others (number of NMD targets as measured in ETG or number of PTCs as measured in ASE). For cancer samples from the TCGA, we further adjusted for other variables such as tumor purity, CNA burden, race, tumor mutation burden (TMB), categories of *POLE* mutations, vital status, and all 86 pan-cancer CNA-PCs. In contrast, for the GTex, the model also accounted for the "Death Group" classification.

In constructing the model for each NMD method (ETG and ASE) and cohort (TCGA and GTex) separately, we adopted a rigorous approach. After establishing the initial model incorporating all variables, we systematically eliminated each variable in turn. This stepwise removal allowed us to evaluate the impact of each variable on the model's overall explanatory power, specifically measuring the reduction in the total variance explained ($R^2$) by the model (Additional File 2: Table S8).

**Cellular deconvolution for TCGA and Hartwig tumors**

We applied the UniCell: Deconvolve Base (*UCDBase*) method [60] to perform bulk RNA-seq deconvolution across the entire TCGA cohort. For this particular analysis, pan-cancer RNA-seq was directly downloaded from XENA browser [150], as the RNA-seq data underwent normalization to mitigate batch effects, followed by a $log_2$ transformation of normalized values ($log_2(NormValue + 1)$) prior to deconvolution. This process resulted in a comprehensive matrix encompassing 10,460 samples, which spanned 708 distinct cell types across various hierarchical levels. Our analysis specifically targeted samples from GBM and LGG to assess cell type composition within these tumor types, specifically for: "astrocyte_of_the_cerebral_cortex", "Bergmann_glial_cell", "brain_pericyte", "endothelial_cell", "neuron", "oligodendrocyte", and "oligodendrocyte_precursor_cell". A similar analysis was done for the Hartwig cohort, where we obtained the following immune cell proportions of CD8 + T-cells: "cd8_positive_alpha_beta_cytotoxic_t_cell", "cd8_positive_alpha_beta_t_cell", "activated_cd8_positive_alpha_beta_t_cell", "cd8_positive_alpha_beta_memory_t_cell", "cd8_positive_alpha_beta_cytokine_secreting_effector_t_cell".

**Quantification of ASE PTC-NMD efficiency (pNMDeff) for inter-individual NMD efficiency variation**

To assess the inter-individual variability of NMD efficiency, we quantified the NMD efficiency for each PTC arising from nonsense mutations or indels, to which we refer as pNMDeff, following the approach previously described by our group[8], applying ASE data instead of transcript-level counts. This methodology was utilized for evaluating germline ASE PTCs across both TCGA and GTex cohorts, and for somatic PTCs within TCGA. The analysis entailed comparing the expression (raw counts) of mutated (MUT) versus wild-type (WT) alleles at the PTC site:

$$pNMDeff = -log2(\frac{MUT\ RNA\ counts}{WT\ RNA\ counts})$$

A pNMDeff score of 0 indicates no mRNA degradation, a score of 1 indicates that the MUT RNA-seq allele counts are half of the WT allele, while a complete heterozygous mRNA degradation approaches infinity.

PTCs or transcripts were systematically excluded from our analysis based on several stringent conditions:

- Exclusion of indels without a downstream PTC prediction.
- Exclusion for additional germline or somatic indel, nonsense, or splice site disrupting variants (not applicable to somatic PTCs in TCGA), or overlap with somatic CNAs.
- Exclusion of PTC-bearing transcripts comprising a single exon without additional 3' UTR splice sites.
- Exclusion based on the gene's LOEUF score position in the first decile, indicating negative selection, or identification of the gene under positive selection.
- Exclusion of homozygous PTC variants.

- Exclusion of PTCs with a DNA variant allele frequency > 20% (for somatic variants in TCGA).
- Exclusion of transcripts with median pan-tissue or pan-cancer expression level (TPM) < 5, or significant expression variation among samples (coefficient of variation > 0.5).

Utilizing pNMDeff data from germline ASE in TCGA and GTex, and somatic ASE in TCGA, we assessed inter- and intra-individual variability of NMD efficiency, focusing on NMD-triggering PTCs as per canonical NMD rules[8]. We applied two metrics: one based on variance and another on correlations.

*Intra-individual variability*: we sampled two PTCs per individual, calculating the Spearman correlation between their pNMDeff scores across 1000 iterations.

*Inter-individual variability*: we compared two sampled individuals for the same common PTC, again calculating Spearman correlation between their pNMDeff scores, also across 1000 iterations.

Variance-based metrics involved calculating the variance of pNMDeff scores for either at least 3 PTCs within individuals or across individuals sharing the same PTC.

Each analysis was conducted separately for each cohort and was supplemented by a control analysis involving randomized pNMDeff values to establish a baseline for comparison.

### NMD-related gene classification

We integrated data from 112 NMD-related genes to investigate gene-level associations with somatic mutations and to evaluate CRISPR codependency scores for identifying causal genes. This selection of NMD genes was informed by two pivotal studies with experimental validations. From Baird et al., [69], we selected the top 76 genes identified via siRNA screening, focusing on those with a median seed-corrected *Z*-score > 1.5. Alexandrov et al. [70] contributed an additional 24 top genes. Our selection also encompasses core NMD factors identified through extensive literature review [3, 7, 151], including *SMG1*, *SMG5*, *SMG6*, *SMG7*, *SMG8*, *SMG9*, *UPF1*, *UPF2*, *UPF3A*, and *UPF3B*, crucial for the NMD mechanism. We included core EJC factors such as *CASC3*, *EIF4A3*, *MAGOH*, and *RBM8A*, essential for RNA splicing and surveillance, alongside EJC-associated genes *ACIN1*, *ALYREF*, *AQR*, *CWC22*, *RNPS1*, *SRRM1*, which interact with or might be part of the EJC. The DECID complex, involving *DHX34*, *ETF1*, *GSTP1*, plays a role in distinguishing defective mRNAs, and genes within the NMD subpathway in the endoplasmic reticulum, *GNL2*, *NBAS*, *SEC13*, were also included in our analysis.

### Association analysis of somatic mutations with variable iNMDeff

The association between iNMDeff and somatic mutations was investigated using general linear models ("*glm*" function from the R package "*stats*"), applied to each gene within 33 cancer types and in a pan-cancer dataset. Covariates for individual cancer types included sample tumor purity, the first 86 principal components of CNA-PCs, and cancer subtype based on mRNA NMF clustering. In the pan-cancer analysis, additional covariates such as CNA burden, RNA-seq sample library size, TMB, age, and sex were incorporated to

address the resulted inflation (Additional File 1: Fig. S13). We tested all the genes but our focus was on (i) a set of 727 recognized cancer genes from the CGC [152]; (ii) 112 genes related to the NMD pathway (see above); (iii) the remaining 18,780 genes were considered as random genes for our analysis. Each gene was subjected to three separate analyses involving the following sets of genetic variants: (i) truncating mutations (including nonsense, indels, and splicing variants) and missense mutations, both tested within the same model unless one mutation type was absent, (ii) synonymous mutations, and (iii) somatic CNAs, where the variable indicated gene amplification (GISTIC CNA score $> 0$) or deletion (GISTIC CNA score $< 0$). In scenarios (i) and (ii), the variable was set to 1 if any mutation was present in the gene, otherwise 0.

### Somatic pan-cancer CNA signatures by sparse-PCA

We performed a sparse principal component analysis (sparse-PCA) on the 24,777 gene CNA arm-level data in 10,654 TCGA samples, using "*sparsepca*" R package [153]. Of note, we duplicated the rows by splitting genes into amplifications/neutral (keeping GISTIC CNA scores $\geq 0$, negative scores were transformed to 0) and deletions/neutral (keeping GISTIC CNA scores $\leq 0$, positive scores were transformed to 0), effectively obtaining a matrix of $49,554 \times 10,654$ dimensions.

In essence, sparse-PCA differs from standard PCA by performing implicit feature selection, focusing only on the most relevant genes that contribute significantly to data variation. This approach is particularly effective for high-dimensional data, as it efficiently manages sparsity, isolates key variables, reduces noise, and improves the interpretability of the results. In sparse-PCA, two parameters are pivotal: "*alpha*" and "*k*." The "*alpha*" parameter governs the sparsity level of the principal components. Higher alpha values induce increased sparsity, thereby reducing the number of genes contributing significantly (non-zero weights) to each principal component (PC). In essence, "*alpha*" serves as a tool for regularization. On the other hand, "*k*" determines the number of sparse PCs to be calculated. The actual number of principal components with non-zero weights is what we refer to as effective PCs. To empirically optimize these parameters, we employed an autocorrelation scoring approach, focusing on the spatial relationship between neighboring genes. Autocorrelation in this context involves comparing each resultant PC, arranged according to genomic location, against a version of itself that is lagged by one gene position. High autocorrelation is anticipated when numerous genes, especially those in proximity, exhibit similar or identical weights due to being affected by the same arm-level Change. Therefore, we expect the correlation between each original sparse-PC and its re-ordered version to approach a value near 1. This was quantitatively assessed by calculating the median autocorrelation of the lowest 1% percentile of effective PCs across various "*alpha*" and "*k*" values (as depicted in Additional File 1: Fig. S14A). Our findings indicated that an "*alpha*" of $3e-04$ and "*k*" of 100 yielded 86 effective pan-cancer PCs (with the remaining 14 PCs having gene weights of zero), while maintaining a median autocorrelation for the first 1% of PCs close to 1.

Each component in the CNA-PC might represent either a genomic gain (Additional File 1: Fig. S14B) or deletion (Additional File 1: Fig. S14C), necessitating careful interpretation of the scores and their signs on a case-by-case basis. For example, a negative score in the CNA-PC does not automatically imply a deletion; it could be associated with a

gain, such as that of chromosome 2p. Therefore, the specific context and characteristics of each CNA-PC must be thoroughly examined to accurately determine whether the observed score reflects a gain or a deletion in the genome.

### Association analysis of somatic pan-cancer CNA signatures with variable iNMDeff

The association between individual weights of the pan-cancer CNA-PCs and our estimates of iNMDeff for both NMD methods was analyzed in 33 individual cancer types, and pan-cancer, from the TCGA. This analysis was conducted using general linear models with the "*glm*" function from the R base package "*stats*." We incorporated several covariates into our models to control for potential confounding factors. These covariates included sample tumor purity, cancer subtype based on mRNA NMF clustering, RNA-seq sample library size, age, and when applicable, sex. Effect size was determined by the beta coefficient of the CNA-PC variable. In the context of our model, a positive coefficient indicates that higher values of the specific CNA signature are correlated with increased NMD efficiency in individuals. It is important to note that these CNA signatures could represent either genomic gains or deletions, hence the direction of the association should be interpreted accordingly.

In our analysis, ASE iNMDeff served as the initial discovery phase, leading to the identification of significant cancer or pan-cancer-specific CNA-PC associations which were then validated using ETG iNMDeff, with FDR adjustments made only for these associations. This approach identified nine total CNA-PCs significantly associated with iNMDeff at an FDR < 10%. Of these, only three pan-cancer CNA-PCs (3, 52, and 86) exhibited consistent directions in their association with NMD efficiency, indicated by the same direction of effect sizes between ASE and ETG (Additional File 1: Fig. S15). In contrast, the remaining six CNA-PCs, which included three cancer-type specific (one in PCPG and two in UCEC) and three pan-cancer CNA-PCs, showed discrepancies in the direction of NMD effect and were therefore not considered in further analyses.

### Prediction of iNMDeff based on gene expression for validations

We developed LASSO or ridge regression models to predict iNMDeff based on gene-level expression data (proxy models). The model was constructed separately for our two NMD methods, ETG and ASE, utilizing gene expression data from either TCGA or GTex. Prior to model development, genes with low expression ($log_2(TPM) \geq 0.5$) in fewer than 50% of pan-tissue samples) and those with low variance were excluded. The dataset was split into a training set (70% of the data) and a test set (30%). Additional preprocessing steps were taken before model construction to account for differences between cohorts, i.e., tumor (TCGA) or tissue samples (GTex) and the other dataset where we wanted to predict iNMDeff. This included correcting the RNA-seq counts matrix through quantile normalization and *comBat* [154], following the methodology outlined in Salvadores et al. [155].

These models were utilized in two distinct validation scenarios: (i) validating chr1q gain associated with reduced iNMDeff in the cell lines from the CCLE and (ii) validating the association of iNMDeff with PFS in external datasets from various studies. For scenario (i), we predicted the cell line NMD efficiency (cNMDeff) for 942 cells. For scenario

(ii), we estimated the iNMDeff for 120 patients in Liu et al. melanoma dataset, 33 in Carroll et al. esophagus dataset, 353 in Motzer et al. kidney dataset, 251 in Hartwig cohort, and 48 in Riaz et al. melanoma dataset.

### Gene codependency CRISPR KO scores with NMD activity in cell line screens

To delineate NMD-associated genes, we leveraged genetic interaction data derived from gene-level CRISPR screening, as provided by the Achilles Project [71]. The initial dataset characterized negative scores as indicators of cellular growth inhibition or lethality post-gene knockout (KO). To enhance the analytical robustness and infer gene functionality more accurately, we opted for a normalized, de-biased dataset employing the onion method and Robust PCA (RPCO), renowned for its efficacy in deducing gene roles [64]. This adjustment yields a more intricate interpretation of the scores.

From the "Outputs generated by normalization pipelines—Normalized networks (ONION)" Section [156]., we accessed the symmetrical matrix file "*snf_run_rpca_7_5_5. Rdata*", encompassing data for 18,119 genes (https://zenodo.org/records/7671685#.Y_gi9nbMK5c). Utilizing this matrix, we performed heatmap clustering (using "*ggcorrplot*" function from the "*ggcorrplot*" R package) to visually represent the relationships among candidate genes implicated in chromosomal gains, alongside core NMD genes, and a set of randomly selected control genes for comparative analysis. For clarity in visualization, scores above 0.1 were capped at this value. Specifically, for chromosome 1q gains associated with CNA-PC3, we analyzed 6 out of 30 candidate genes (Fig. 4F), discarding those that did not cluster effectively for visualization (Additional File 1: Fig. S20B). The complete set included 18 "Candidate" genes (those genes residing within 1q21.1–23.1 region), 4 "Candidates NMD" (NMD-related genes residing within 1q21.1–23.1 region: *SMG5*, *SF3B4*, *RBM8A*, *INTS3*), 21 NMD-related genes (NMD-related genes outside 1q21.1–23.1 region)—comprising 9 essential NMD factors, 3 EJC components, and 6 EJC-related genes, alongside 3 DECID complex members—and 18 control genes situated on chromosome 1q but outside the 1q21.1–23.1 region, excluding adjacent regions (1q23.2-1q31.3) for rigor. Note that *KIAA0907* was excluded from the analysis due to lack of CRISPR data. *SF3B4* and *RBM8A*, despite not meeting ETG iNMDeff thresholds, were included as an additional "Candidates NMD" based on its related functions in NMD (spliceosome and EJC, respectively) and its location within 1q21.1–23.1. Similarly, for chromosome 2q gains linked to CNA-PC52, all obtained 14 "Candidate" genes and 4 "Candidate NMD" genes (*CWC22*, *SF3B1*, *NOP58*, *FARSB*) were evaluated alongside the set of 21 NMD-related genes and 22 control genes outside the specific 2q31.1-2q36.3 region. *FARSB*, despite not meeting ETG iNMDeff thresholds, was included as an additional "Candidates NMD" gene based on its related functions in NMD and its location within 2q31.1-2q36.3.

It is important to note that the KO of some NMD factors, especially *UPF1*, may be lethal to cells, which could explain the lower similarity scores observed among NMD genes in CRISPR screens, which fully ablate gene activity.

Furthermore, we quantitatively assessed the NMD association by calculating the mean CRISPR codependency scores between each candidate gene and the 10 core NMD factor genes (*UPF1*, *UPF2*, *UPF3A*, *UPF3B*, *SMG1*, *SMG5*, *SMG6*, *SMG7*, *SMG8*, *SMG9*). This was juxtaposed with the mean CRISPR scores obtained between the same candidate

gene and the set of random negative control genes, specific to the chromosomal regions under investigation (either 1q or 2q). Specifically, for the chromosome 1q region, linked to CNA-PC3, our assessment encompassed all 274 genes located within the 1q21.1–23.1 boundary and were compared against 18 randomly selected control genes situated outside this specified region. Similarly, for the chromosome 2q region, associated with CNA-PC52, our analysis included all 303 genes found within the 2q31.1-2q36.3 region and were compared to 383 random control genes positioned outside the targeted region.

### Rare variant association studies (RVAS) via SKAT-O

We conducted a gene-based rare variant association study (RVAS) following the method from a previous study in our group [157], utilizing the SKAT-O combined test [72] (see schematic from Fig. 5A). Unlike burden testing, where variants are aggregated before regression against a phenotype, SKAT individually regresses variants within a gene against the phenotype and tests the variance of the distribution of individual variant score statistics. The SKAT-O test statistic is a weighted combination of the burden test (QB) and the SKAT variance test (QS) statistics, formulated as $Q\rho = \rho \times QB + (1 - \rho)QS$. In this equation, $Q\rho$ represents the weighted mean statistic, and the parameter $\rho$ determines the weighting of each test component. Notably, a gene burden test demonstrates greater power when all rare loss-of-function (pLoF) variants in a gene are causal, whereas SKAT exhibits higher power when some rare pLoF variants are not causal or when rare pLoF variants are causal but exert effects in opposing directions [72].

We defined three stringency thresholds for determining putative pLoF for the rare germline variants, i.e., those with a population allele frequency (MAF) of < 0.1%:

 (i)  pLoF: NMD-triggering PTC variants.
 (ii) pLoF_Missense_CADD15: NMD-triggering PTC variants + Missense with CADD ≥ 15.
 (iii) pLoF_Missense_CADD25: NMD-triggering PTC variants + Missense with CADD ≥ 25.

The predicted deleterious missense variants were calculated using CADD scores [69] at two different cutoffs: ≥ 25 (more stringent) and ≥ 15 (permissive).

Association testing was conducted separately for each cancer (33) or normal tissue type (56) and collectively for pan-cancer or pan-tissue analysis. We only considered individuals with European ancestry. A gene's effect on iNMDeff was tested only if a rare pLoF variant in that gene was found in at least two individuals, across 18 matched cancer-normal tissues (Additional File 2: Table S5). This resulted in varying numbers of genes being tested per cancer or tissue type. Significant associations are allowed to validate between GTex and TCGA cohorts across the different pLoF variant sets and NMD efficiency estimation method (ETG or ASE). In other words, as long as there is a significant association in matched tissues in both GTex and TCGA, regardless of the dataset or NMD method used, the association of the gene with NMD efficiency is considered replicated. The level of significance was determined after applying FDR correction using Benjamini–Hochberg within each tissue type separately in both cohorts (only correcting using the significant genes in the validation cohort). As a negative control,

we randomized the iNMDeff values and re-tested the associations in the same way. To assess the significance difference between the observed proportion of significant genes (48/1,072,334) and the randomized proportion (24/1,035,154) we calculated the *Z*-score and its associated *p*-value.

Additionally, we established an empirical FDR benchmark by contrasting the proportion of significant findings against those from randomized data. We calculated an empirical FDR of 30% at a nominal threshold of 5%, following this formula:

$$Empirical\ FDR\ threshold = \frac{\frac{number\ of\ observed\ significant\ hits}{total\ number\ of\ observed\ tests}}{\frac{number\ of\ randomized\ significant\ hits}{total\ number\ of\ randomized\ tests}}$$

In more detail, we performed SKAT-O testing using the R package "*SKAT v2.0.1*." The initial step involved regressing covariates against iNMDeff using the function "*SKAT_Null_Model*." When applicable, we controlled for age at diagnosis, sex, cancer (in TCGA) or tissue (in GTex) type, ancestry (first 6 PCs), RNA-seq sample library size, and tumor purity (in TCGA), as covariates. Categorical variables were encoded as dummy variables using "*fastDummies v1.6.3*." Missing age data were imputed using the median value from the respective cohort. Post null model initialization, SKAT-O was executed with the "*SKAT*" function, setting the method to "*SKATO*." This process involved running SKAT-O with 10 different *p*-values (ranging from 0 to 1) and identifying the *p*-value yielding the lowest *p*-value.

We only tested the additive type of inheritance model, encoding individual variants as follows: 0 for no rare pLoF, 1 for rare pLoF, 2 for rare pLoF with somatic loss of heterozygosity (LOH, in TCGA) or biallelic rare pLoF.

### Gene-burden testing to estimate effect sizes for the significant RVAS genes

In addition to SKAT-O, we conducted gene-based burden testing to aggregate variants within the same gene, applying identical models as previously described. This was necessary because SKAT-O does not report effect sizes. The association testing was executed using linear regression with the *'glm'* function from that *'stats'* base R package. The model was structured as *iNMDeff* $\sim$ *Gene + covariates*. Here, the 'Gene' variable was treated as a binary categorical variable, representing the presence or absence of aggregated gene-level variants from the additive model. Covariates were incorporated as appropriate, including age at diagnosis, sex, cancer (in TCGA) or tissue type (in GTex), ancestry (first 6 PCs), RNA-seq sample library size, and tumor purity (in TCGA). Primarily, burden testing was employed to ascertain the effect sizes, i.e., the beta coefficients, of the 10 significant and replicated hits identified in our RVAS analysis. A positive effect size indicates that pLoF variants within the gene are associated with a higher iNMDeff and vice versa. *P*-values were adjusted by FDR within every gene and tissue separately, for every cohort.

### Redefinition of cancer driver gene list for estimating selection

Our initial selection of cancer-related genes originated from the 727 genes listed in the Cancer Gene Census (CGC) [152]. Within this list, we reclassified genes labeled as

"Oncogene, Fusion" as Oncogenes (OGs), and similarly reclassified genes for tumor suppressor genes (TSGs). However, we excluded genes labeled as "oncogene, TSG" or "oncogene, TSG, fusion" due to the ambiguity in their classification. Additionally, we narrowed our focus to OGs with dominant inheritance patterns and excluded those specific to leukemia. To further refine our list, we cross-referenced it with two distinct gene sets: (1) The gene set identified by Solimini et al. [158], who employed computational predictions and experimental validations through shRNA techniques. We intersected 1667 "STOP" genes with our 240 TSGs and 1249 "GO" genes with our 187 OGs. (2) Mutpanning gene set [159], where we focused on 41 genes with a cancer incidence higher than 3 and a significant $q$-value < 0.01, intersecting with our 240 TSGs. For our 187 OGs, we included all 217 genes with any cancer incidence and a significant $q$-value < 0.01. By combining the genes from these two intersections, we formed a more refined and specific gene set, which ultimately comprised 50 TSGs and 52 OGs. This integrative approach ensures a comprehensive and evidence-based selection of 102 key cancer genes.

### Estimating selection in cancer driver genes via dN/dS

The dN/dS method is a state-of-the-art method to estimate selection, computing the dN/dS ratios for different types of mutations—missense, nonsense, and frameshift—at the gene level. A dN/dS ratio above 1 for a gene indicates positive selection, while a ratio below 1 signifies negative selection. dN/dS was calculated using the "*dndscv*" function from the R package "*dndscv*" using the substitution model "*submod_192r_3w* " [160].

We implemented this approach on all nonsense (stratifying NMD-triggering and NMD-evading PTCs) and missense mutations (serving as controls) found within the CGC set of genes in the pan-cancer TCGA cohort. We also applied this analysis for 33 cancer types separately, to identify cancer-specific genes. Individuals were grouped in iNMDeff high vs low, based on the median. In other words, our analysis was divided into various categories: (i) comparison between NMD-triggering and NMD-evading PTCs; (ii) comparison of individuals with high versus low iNMDeff; (iii) analysis of the two types of somatic mutations, namely nonsense/PTCs and missense; (iv) the two categories of cancer genes: TSG and OG. Consequently, each gene underwent dN/dS ratio calculation eight times, reflecting the above different contexts including mutation type, NMD region type, cancer gene type, and iNMDeff group sample type.

To enhance the accuracy of our classification between NMD-triggering and NMD-evading PTCs, we incorporated a new NMD-rule [8] acknowledging that PTCs in longer exons might escape NMD. PTCs were classified as NMD-triggering if they met all criteria: (1) > 250nt from TSS, (2) in exons ≤ 500nt, and (3) not in the last exon or last 55nt of the penultimate exon. Conversely, PTCs were classified as NMD-evading if they met any criteria: (1) ≤ 250nt from TSS, (2) in exons ≥ 1000nt, or (3) in the last exon or last 55nt of the penultimate exon. For the weaker long-exon rule of NMD evasion, we created a "buffer zone" by excluding mutations in exons of lengths between 500 and 1000nt, reducing potential misclassification.

### Survival analysis and Cox regressions

To investigate the relationship between individual NMD efficiency and cancer survival, we employed survival analysis and Cox proportional hazards regression models

for each of the 47 cancer types stratified into subtypes and for the two NMD methods. These models were used to analyze two key survival outcomes: overall survival (OS) and progression-free survival (PFS). The structure of the model was as follows: $survival\_outcome \sim iNMDeff\_group + covariates$. In this model, "*iNMDeff_group*" was utilized as a binary categorical variable to distinguish between groups with high and low iNMDeff. This classification was based on a quantitative threshold determined by a specific percentile. In TCGA, we incorporated several covariates when applicable, which included sex, cancer type, RNA-seq sample library size, tumor purity, age (categorized into quartiles and treated as a discrete variable), and TMB. In the external validation datasets, and when applying Cox regressions using PFS as the outcome, the selection of covariates was adjusted based on data availability. For Carrol et al. and Motzer et al. datasets, we only could use age and sex. However, in the Liu et al. dataset, we were able to include sex and tumor purity as covariates. For Carrol et al. and Liu et al., but not Motzer et al., we additionally included TMB.

The Cox regression analysis was systematically repeated across a spectrum of percentiles (every 5th percentile up to the 50th), enabling an intricate evaluation of NMD efficiency's impact on survival. For example, classifying individuals at the 20th percentile is comparing the top 20% of individuals with the highest iNMDeff against the bottom 20% with the lowest iNMDeff. Meta-analysis *p*-values were calculated between the two NMD methods when possible, applying FDR correction per cancer type afterwards, within the range of percentiles and both NMD methods. For outlier associations, we set exclusion criteria for cases where confidence intervals exceeded 20, or the exponential hazard ratio coefficients were beyond the bounds of 15 and 0.001, or resulted in "Inf." Additionally, associations were disqualified if regression analysis warnings indicated a failure to converge, specifically if the model "Ran out of iterations and did not converge" or a "Non-convergence warning detected" was issued. We performed a randomization of iNMDeff values and re-execution of Cox regressions to establish an empirical FDR benchmark by contrasting the proportion of significant findings against those from randomized data. With this, we calculated an empirical FDR of 16% at a nominal threshold of 5%, following this formula:

$$Empirical\ FDR\ threshold = \frac{\frac{number\ of\ observed\ significant\ hits}{total\ number\ of\ observed\ tests}}{\frac{number\ of\ randomized\ significant\ hits}{total\ number\ of\ randomized\ tests}}$$

For our survival analysis, we utilized the *'surv'* and *'survfit'* functions from the *'survival'* R package. Kaplan–Meier curves were constructed using the *'ggsurvplot'* function from the *'survminer'* R package. Additionally, Cox proportional hazards regression models were executed employing the *'coxph'* function from the *'survival'* R package.

For progression-free survival (PFS) analysis and Cox regression, we included all samples with no available treatment data as negative controls ($n = 6,189$). The number of samples for immunotherapy and chemotherapy varied depending on the cancer type and the percentile of iNMDeff tested, ranging from 1 to 73 samples for immunotherapy (total: 184 samples) and from 1 to 275 samples for chemotherapy (total: 2,965 samples). For the chemotherapy group, we excluded drugs that were more likely classified

as immunotherapy agents. These included pembrolizumab, nivolumab, ipilimumab, tremelimumab, cemiplimab, atezolizumab, avelumab, durvalumab, dostarlimab, and retifanlimab.

**Immunotherapy-treated patients from external datasets and filterings**

To validate our PFS or immunotherapy response findings, we utilized external patient datasets from different studies, each focusing exclusively on individuals undergoing different immunotherapy treatments:

(1) Liu et al. [76]: 144 melanoma (SKCM) patients treated with anti-PD1 immune checkpoint inhibitors were examined. We normalized their RNA-seq gene-level TPM data using $log_2(TPM + 1)$, excluding patients who received more than one previous therapy.

(2) Carroll et al. [80]: This research involved 35 patients with inoperable esophageal adenocarcinoma (EAC) who received first-line immunochemotherapy, initiating with ICI for four weeks followed by chemotherapy. RNA-seq gene-level raw counts were obtained, transformed to TPM using "*convertCounts*" function from "*DGEobj. utils*" R package, and normalized $log_2(TPM + 1)$. Only pre-treatment ("PreTx") RNA-seq samples were considered.

(3) Motzer et al. [81]: This dataset comprised 886 patients with advanced renal cell carcinoma (RCC), treated with the ICI avelumab plus axitinib or only sunitinib. The RNA-seq gene-level $log_2(TPM)$ counts were obtained for these patients. It is important to note that in the original data, values below 0.01 were set to 0.01, thus eliminating the need for pseudocount adjustments in our analysis. Samples receiving only sunitinib were omitted as it is a targeted therapy and not an immunotherapy.

(4) Hartwig Medical Foundation [82]: this study included RNA-seq data from 3020 metastatic samples across 22 cancer types. We focused on 251 samples treated with immunotherapy (variable "consolidatedTreatmentType" = "immunotherapy"), normalizing their gene-level RNA-seq counts using $log_2(TPM + 1)$.

(5) Riaz et al. [83]. Tumors from 68 patients with advanced melanoma (SKCM), who progressed on ipilimumab or were ipilimumab-naive, before and after nivolumab initiation. We accessed the RNA-seq gene-level raw counts (UCSC hg19) and transformed their TXIDs into Ensembl Gene IDs using "*biomaRt*." After converting the raw counts to TPM and normalizing via $log_2(TPM + 1)$, we retained only the samples under treatment ("On"), excluding pre-treatment ("Pre") cases due to their scarcity.

(6) TCGA: 469 melanoma (SKCM) samples, 73 were identified as having received at least one immunotherapy treatment.

The iNMDeff for these datasets was predicted using our proxy models based on gene-level expression RNA-seq data, with TCGA or GTex cohorts serving as training sets. Averages of predictions from both cohorts were taken as the final iNMDeff for each individual, calculated separately for ETG and ASE methods. This approach yielded two iNMDeff values per individual. The final sample sizes, after filtering, were 120 for Liu-SKCM,

33 for Carroll-EAC, 353 for Motzer-SKCM, 251 for Hartwig, 48 for Riaz-SKCM, and for TCGA-SKCM, we utilized directly our estimated iNMDeff values (no proxy model used here) for 36 and 73 ASE and ETG melanoma patients, respectively.

### Prediction model of immunotherapy response

To predict immunotherapy response, we designed a simple logistic regression model incorporating TMB and our predicted iNMDeff to classify immunotherapy outcomes as "Responders" (Complete Response, CR, and Partial Response, PR) or "Non-Responders" (Stable Disease, SD, and Progressive Disease, PD). This analysis spanned various datasets, excluding Motzer-SKCM due to unavailability of TMB and treatment response data, and included TCGA-SKCM, Carrol-EAC, Liu-SKCM, and Hartwig (categorized by urinary tract, lung, and melanoma), as well as Riaz-SKCM. The approach to TMB calculation and response categorization was dataset-specific and manually curated:

- TCGA SKCM ($n = 41$): Patients with at least one immunotherapy treatment were identified, with the best response recorded as the outcome. Patients with "NA" in response to treatment were excluded. Clinical PD or SD were tagged as "Non-Responders", whereas PR or CR as "Responders". TMB was computed as the total count of somatic coding mutations per individual, normalized by the exome size (36 Mb).
- Liu-SKCM ($n = 118$): Used the "BR" variable, excluding "MR" responses, and classified the remaining as per standard criteria. The "Total_muts" variable served as TMB.
- Carrol-EAC ($n = 33$): Applied "Response_binary" directly for classification into "Responders" and "Non-Responders." Only pre-treatment (PreTx) iNMDeff-estimated samples were considered, with "PreTx.Tumor_Mutational_Load" as TMB.
- Riaz-SKCM ($n = 48$): Directly used the "Response" variable, including only patients treated "On" treatment, utilizing "Mutation.Load" for TMB.
- Hartwig ($n = 251$): "firstResponse" was filtered to exclude "ND" and "MR," with "CR" and "PR" classified as Responders; "PD," "Non-CR/Non-PD," "Clinical progression," "SD" as "Non-Responders," stratified by cancer type for adequately sized samples (discarded if $n < 20$): Hartwig-UT (urinary tract, $n = 23$), Hartwig-lung ($n = 26$), and Hartwig-SKCM (melanoma, $n = 119$). "TOTAL_SNV" was taken as TMB.

Covariate adjustments were made based on dataset characteristics, incorporating age and sex generally, with Liu-SKCM excluding age but including cancer subtype due to significant TMB differences across melanoma subtypes, and Riaz-SKCM only including cancer subtype.

The "*glm*" function from R's stats package, set with "binomial" family parameter, was employed for the logistic regression. Model efficacy was gauged using "AUC" and "roc" from the "pROC" R package, contrasting AUC values against a randomized baseline (~ 0.5 AUC) via "roc.test" for significance evaluation. Moreover, sensitivity (recall), specificity, and precision were calculated using the "*caret*" R package, providing a comprehensive evaluation of the model's performance across various metrics.

## Supplementary Information

---

Additional File 1: Supplementary Figures S1-S29

Additional File 2: Supplementary Tables S1-S20

Additional File 3: Supplementary Text S1-S6

Additional File 4: Peer review history

---

### Acknowledgements
We are grateful to Dr. Javier Lanillos for processing the TCGA RNA-Seq data in brain tumors and Hartwig RNA-Seq data using UniCell Deconvolve, and for pointing us to immunotherapy studies of interest. Further, we thank all members of the Genome Data Science lab for discussions. This publication and the underlying research are partly facilitated by Hartwig Medical Foundation and the Center for Personalized Cancer Treatment (CPCT) which have generated, analyzed, and made available data for this research (request number DR-260). The results published here are in whole or part based on data generated by the TCGA Research Network (https://www.cancer.gov/tcga).

### Review history
The review history is available as Additional file 4.

### Peer review information

### Authors' contributions
GP-M performed all data collection and curation, wrote code, and performed bioinformatics analysis and visualization of results. FS conceived and supervised the study. Both authors jointly designed the analyses, interpreted the results, and wrote the manuscript.

### Funding
P.-M. was funded by an AGAUR FI fellowship, and F.S. was funded by the ICREA Research Professor program. Work in the lab of F.S. was supported by an ERC StG "HYPER-INSIGHT" (757700), ERC CoG "STRUCTOMATIC" (101088342), Horizon2020 project "DECIDER" (965193), Horizon Europe project "LUCIA" (101096473), Spanish government project "REPAIRSCAPE," CaixaResearch project "POTENT-IMMUNO" (HR22-00402), a Novo Nordisk Fonden Start Package grant, the Danish Cancer Society grant "AI-DRIVERS", the SGR funding of the Catalan government, the Severo Ochoa Centers of Excellence award of the Spanish government to the hosting institution.

### Data availability
Data used in this study includes WES ***bam*** files and RNA-seq ***fastq*** files from TCGA (downloaded via GDC Data Portal, https://portal.gdc.cancer.gov/ [121]. From Genotype-Tissue expression (GTex) project [124], we directly obtained WES Data used in this study includes WES ***bam*** files and RNA-seq ***fastq*** files from TCGA (downloaded via GDC Data Portal, https://portal.gdc.cancer.gov/ [121]. From Genotype-Tissue expression (GTex) project [124], we directly obtained WES ***VCF*** files, tissue-specific transcript-level RNA-seq counts data and allele-specific expression (ASE), all accessed through dbGaP (via AnVIL, https://anvilproject.org/), v8 (dated 05–06-2017). The GENCODE annotation file used was v26 [161]. Copy number alteration (CNA) data, including both arm-level and focal-level information, was obtained from the Broad Institute's GDAC Firehose portal (https://gdac.broadinstitute.org/ [123] TCGA cancer cluster subtypes, determined based on mRNA non-matrix factorization (NMF), were sourced from the Broad Institute's GDAC Firehose portal (https://gdac.broadinstitute.org/ [123]. In addition, we acquired estimates of leukocyte fraction [112] and tumor purity [113] for all samples within the TCGA dataset. Patient clinical data was obtained through GDC portal and progression-free survival (PFS) information through cBioPortal (https://www.cbioportal.org/. [122, 125] Immune infiltration proportion of CD8 + cytotoxic T-cells, as estimated via "***quanTIseq TIL10***" signature, was downloaded from The Cancer Immunome Atlas (https://tcia.at/home [126]. Cancer molecular subtypes were sourced from different studies [61, 127–133, 135], see Methods section "TCGA cancer type stratification into subtypes" for more details. We obtained NMD targets gene sets from a range of studies [5, 6, 13, 14, 51], see Methods section "Obtaining NMD target gene sets from various studies" for more details. CRISPR KO data for 18,119 genes were obtained from the "Outputs generated by normalization pipelines—Normalized networks (ONION)" Section [156], via accessing the symmetrical matrix file "***snf_run_rpca_7_5_5.Rdata***" (https://zenodo.org/records/7671685#.Y_gi9nbMK5c). RNA-seq and CNA data for 1450 cell lines were sourced from CCLE [65] via DepMap portal (https://depmap.org/portal/download/all/). External datasets of immunotherapy-treated patients were obtained from different studies: Liu-SKCM [76], Carrol-EAC [80], Motzer-RCC [81], Riaz-SKCM [83], Hartwig [82] (via request DR-260). PRO-seq data was obtained from three studies: SH-SY5Y (GSE214243) [55], A549 (GSM5169137) [56], and U2OS (GSM5169140) [56]. RNA-seq data of ***UPF1*** KD and WT cell lines were obtained from several studies: HeLa (GSE152435 and GSE86148) [5, 53], HepG2 (GSE88148 and GSE88466) [54], and K562 (GSE88140 and GSE88266) [54]. Data generated in the analyses presented in this study can be found on Zenodo at https://doi.org/10.5281/zenodo.15974216161. The code used in the study can be found at https://github.com/gpalou4/iNMDeff [162] and is released under the MIT License. A copy of the code is also deposited on Zenodo [163].

## Declarations

### Ethics approval and consent to participate
This work was performed using publicly available datasets and thus specific ethical approval was not sought.

### Competing interests
The authors declare that they have no competing interests.

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

## 