## [Additional File 4: Peer review history · Genome Biology]

Review history

First round of review

Reviewer 1

In this interesting resource article Palou-Marquez and colleagues focus on the variable efficiency of the NMD pathway across human tissues, individuals and tumor samples. Some of these aspects had been previously covered, but never to the extent that this study covers.

The authors use two experimental approaches to assess NMD efficiency. First, they measure allele-specific expression (ASE) of PTC-bearing transcripts, which is complemented by an assessment of the expression of endogenous NMD target transcripts (ETG). From these initial studies, the authors conclude that both methods are robust for quantifying individual-level NMD efficiency.

This is a timely area of research and not much has been done in the field to quantitatively assess NMD efficiency and its variation across tissues, individuals or in relation to pathology. With this in view, this study reports interesting novel findings that show in a genome-wide manner that there is extensive NMD variability. Of particular interest, lower NMD efficiency has been found in the nervous system. Another interesting finding, with possible clinical implications is the extensive inter-individual variation of NMD efficiency.

Overall, this is a very good study carefully planned and executed, with a large number of experimental data and computational analysis. In summary, this is an interesting study that confirms and expands previous studies and uses a well-developed computational approach to assess the efficiency of the NMD pathway and its variability across different conditions and in relation to human disease. I think this is a good fit for Genome Biology.

Specific comments

- Some of the conclusions related to cancer are highly speculative. For instance, the fact that NMD seems to be more potent in gastrointestinal tissues, does not fully support the proposed idea that colon tumors do rely on efficient NMD, which was shown in just one paper, which is not particularly strong

Reviewer 2

In this study, Palou-Márquez and Supek develop and benchmark two computational methodologies - based on ETG (endogenous target gene) and ASE (allele-specific expression) - to quantify the efficiency of nonsense-mediated mRNA decay (NMD) using exome and RNA-seq data from the TCGA and GTEx cohorts. Their methods appear robust, with a careful description of both the methodology and the strengths and weaknesses of each approach detailed in the Methods section.

Using these methodologies, they explore NMD efficiency variability across various tissues and tumors in human individuals. The rigor of the study is evident, as they take care to validate their findings. For example, a weak but significant association between copy number alteration (CNA) gains and reduced NMD efficiency is validated using an independent dataset from the Cancer Cell Line Encyclopedia.

The study concludes by investigating the potential of global NMD efficiency measures as biomarkers for predicting immunotherapy outcomes in certain cancer types, showing some promise.

Disclaimer: While I am familiar with the NMD mechanism, I have only a general understanding of the statistical techniques used in this study. Therefore, my evaluation is more focused on the readability and biological aspects than the statistical rigor of the methods.

Overall, I am positive about the work, especially with regard to the robustness of their methods and the conclusion that there is substantial variability between tissues and individuals. However, I remain somewhat skeptical about the practical utility of these methods as biomarkers when applied to individual patients. That being said, this does not detract from the merit of the study, and future research may well demonstrate its broader applicability.

Major Specific Comments:

Figure Presentation and Citation Order: It can be confusing that figures and figure panels are not cited in sequential order. For example:

- Supplemental Figure S3 is cited before Supplemental Figure S2.
- Figure 2D-E is cited before Figure 2C.
- Figure 3F is cited before 3E.
- Figure 4E is cited before 4C-D. It would be clearer if figures were cited in order.

Interpretation of Results - NMD Efficiency and Chemotherapy vs. Immunotherapy: In the section discussing chemotherapy and immunotherapy (p.28, l.16-19), it seems the data from Figure S21B-C indicates that chemotherapy outcomes may be more affected by NMD efficiency than immunotherapy. The current interpretation seems somewhat biased toward immunotherapy. A more balanced interpretation would be helpful.

Positive Selection on NMD-Evading Mutations: The analysis of NMD-triggering mutations seems a bit biased. It would be helpful to provide an explanation for why there is similar (Figure 6A) or even stronger (Supplemental Figure S19A) positive selection on NMD-evading mutations.

Methodology - Discovery vs Validation: In Figure 5C and Supplemental Figure S16B, there are individuals in the validation dataset with no corresponding individuals in the discovery dataset (e.g., LAMC1 in the ASE plot). Are all discovered genes/tissues from ASE and ETG subjected to validation by either method?

Significance of Candidates - Randomized Procedure: On page 22, the study identifies 48 significant candidates, while the randomized version produces 24 significant candidates. Is this difference useful or significant? Further discussion of the implications of this finding would be beneficial.

Method Section - NMD Efficiency Metric: The formula for pNMD_{eff} (p.41, l.7) could benefit from clarification. I understand that a score of 0 means no mRNA degradation, but why does a score of 1 represent complete heterozygous mRNA degradation? A score of 1 would imply that the mutant RNA-seq allele counts are half of the wild-type RNA-seq allele counts. Please clarify this point.

Minor Specific Comments:

Figure 1: Explain the abbreviations used in the figure within the legend.

Figure S1B: It seems there is a typo; ">=2 uORF 30 bp?" should likely read ">=2 uORF 30 bp?"

Page 8, Line 17: A reference to Figure 2B is missing.

Figure 2C - Variants: The presented variants (4 and 10 for the two individuals) should be clarified. Do these represent all detected PTC variants in the samples?

Page 10, Line 6: A reference to Figure S4D is missing.

Figure S6A-B: It would be helpful to explain what the yellow and green-ish bars represent in the Pearson correlation plots. Additionally, is there any trend observable in these correlations?

Page 14, Lines 42-44: The sentence about reduced variance in NMD efficiency is difficult to understand. Consider rephrasing it for clarity.

Page 14, Lines 46-50: This sentence may be missing a reference to Supplemental Figure S6C.

Page 24, Line 12: "In specific" should be changed to "Specifically."

Page 30, Lines 31-35 - Sentence Structure: The sentence discussing MEN1 gene and NMD activity should be split into two sentences for clarity.

Page 31, Line 15: The phrase "...also contains NMD factors SMG5, SMG7..." should be modified to "...also contains genes encoding NMD factors SMG5, SMG7..."

Figure S15B+D: The y-axis label might represent the negative log₁₀ of the FDR values. Please verify and clarify.

Methods Section - NMD-Related Genes: A table summarizing all NMD-related genes used in the study should be included.

Page 46, Line 59: The word "redefinment" should be changed to "redefinition."

Reviewer 3

In this paper, the authors analyzed thousands of tumor and healthy tissue samples to examine variations in nonsense-mediated mRNA decay (NMD) efficiency across different tissue types and between individuals. Their findings suggest that NMD efficiency is lower in the nervous and reproductive systems compared to the digestive system, with potential genetic influences from copy number changes in core NMD genes and germline mutations in genes like KDM6B. However, there are significant concerns about the accuracy and reliability of their NMD efficiency measurements. These concerns raise questions about the robustness of the authors' conclusions.

The major concerns include:

(1) Their definition of NMD efficiency may not really capture the strength of NMD targeting on mRNA degradation. There is no validation/proof to show their calculated measurements really represent NMD efficiency. Their proposed NMD efficiency is roughly defined as the average expression negative log ratio of NMD targeted transcripts vs. control transcripts for both the approach using endogenous target gene and the one using allele-specific expression. Note that expression measurements from RNA-seq are just static snapshots, which do not capture the full dynamic processes of transcription, splicing, and degradation. As a result, it is problematic to claim that NMD efficiency is lower in the nervous system. For example, if transcription in nervous tissues generates more NMD-targeted transcripts, they could appear in high quantities even if NMD efficiency is actually high, which would artificially produce a lower NMD efficiency score under this definition.

(2) The agreement between the two methods the authors used to measure NMD efficiency is not satisfactory: only about 75% of the tissues/cancers have positive correlation. And the correlation based on pan-cancer or pan-tissue is low, at 0.14 and 0.19, respectively (as shown in Supp. Fig. S3). This low level of correlation raises concerns about the reliability of their NMD efficiency estimates.

(3) In GTEx, the sample sizes of brain tissues are generally smaller than those of other tissues. This discrepancy could confound the authors' observation that NMD efficiency is lower in the nervous system.

(4) Why the replicated association between the CNA-PCs and NMD efficiency at the pan-cancer level is usually not observed at the cancer type level?

(5) Some figures and legends in the paper are unclear or seem inconsistent with the data. Such inconsistency could lead to misunderstandings. For example, the legend for Fig. 4A mentions six CNA-PCs, though only three of these are actually replicated.

(6) In Fig. 4E, there is no statistically significant difference in NMD efficiency between the high and mild groups for CNA-PC3, which further highlights the unreliability of their NMD efficiency measurements.

(7) In Fig. 4F, there are only six, not 30 candidate genes as mentioned in the figure legend and main texts.

(8) The manuscript is quite lengthy, which can detract from the key findings. To improve clarity, the authors could consider moving less critical sections and methods to supplementary materials, focusing on the most significant findings in the main text. This would make the paper more accessible and ensure that the main conclusions are clearly highlighted.

Authors' response to reviewers

Reviewer #1:

In this interesting resource article Palou-Marquez and colleagues focus on the variable efficiency of the NMD pathway across human tissues, individuals and tumor samples. Some of these aspects had been previously covered, but never to the extent that this study covers.

The authors use two experimental approaches to assess NMD efficiency. First, they measure allele-specific expression (ASE) of PTC-bearing transcripts, which is complemented by an assessment of the expression of endogenous NMD target transcripts (ETG). From these initial studies, the authors conclude that both methods are robust for quantifying individual-level NMD efficiency.

This is a timely area of research and not much has been done in the field to quantitatively assess NMD efficiency and its variation across tissues, individuals or in relation to pathology. With this in view, this study reports interesting novel findings that show in a genome-wide manner that there is extensive NMD variability. Of particular interest, lower NMD efficiency has been found in the nervous system. Another interesting finding, with possible clinical implications is the extensive inter-individual variation of NMD efficiency.

Overall, this is a very good study carefully planned and executed, with a large number of experimental data and computational analysis. In summary, this is an interesting study that confirms and expands previous studies and uses a well-developed computational approach to assess the efficiency of the NMD pathway and its variability across different conditions and in relation to human disease. I think this is a good fit for *Genome Biology*.

We would like to thank the reviewer for the positive comments regarding interest in the findings and for highlighting that it is a well-crafted research article.

Specific comments:

- Some of the conclusions related to cancer are highly speculative. For instance, the fact that NMD seems to be more potent in gastrointestinal tissues, does not fully support the proposed idea that colon tumors do rely on efficient NMD, which was shown in just one paper, which is not particularly strong.

We believe the reviewer is referring to the hypothesis that colon tumors (and more substantially the Microsatellite Instability - MSI - tumors, based on doi: 10.1038/s41389-018-0079-x) do rely on efficient NMD efficiency, and that this is not completely validated in our study. Indeed, we agree that this issue needs clarification. In addition to the study the reviewer referenced (we believe they were referring to 10.1038/s41389-018-0079-x) and our own analysis (see Supp. Fig. S7A and Supp. Fig. S9C), we highlight several others that support this hypothesis that efficient NMD is observed in colon cancers. For example:

1. "*Pan-cancer proteogenomics connects oncogenic drivers to functional states*" DOI: 10.1016/j.cell.2023.07.014 → MSI-high tumors showed significantly higher NMD enrichment pathway scores compared to microsatellite-stable (MSS) tumors.
2. "*Up-frameshift Protein 1 Promotes Tumor Progression by Regulating Apoptosis and Epithelial–Mesenchymal Transition of Colorectal Cancer*" DOI: 10.1177/15330338211064438 – *UPF1* expression was upregulated in colorectal cancer (CRC) tissues and cell lines. High *UPF1* expression correlated

with advanced stage, lymph node metastasis, and shorter survival. Furthermore, *UPF1* promoted metastasis by inducing epithelial-to-mesenchymal transition (EMT) and inhibiting apoptosis.

3. "*UPF1 promotes chemoresistance to oxaliplatin through regulation of TOP2A activity and maintenance of stemness in colorectal cancer*" DOI: <https://doi.org/10.1038/s41419-021-03798-2>—*UPF1* was overexpressed in CRC, predicted poor overall survival (OS) and recurrence-free survival (RFS). *UPF1* contributed to chemoresistance to oxaliplatin by interacting with *TOP2A* and maintaining stemness.

These prior studies support our observation regarding high NMD efficiency in tumors of the colon (in some cases, observed in the MSI cancers). We would also point out that claims about higher NMD in GI tract are perhaps not central in the current study. Rather, the GI serves as a reference point for tissue specificity of NMD efficiency, and our focus is the (robust) claim that NMD efficiency is lower in the central nervous system, likely specifically in neurons.

Further validation of the digestive tissues increased NMD activity remains for future work, for example via experimental models or analysis in other cancer datasets (unfortunately, currently large-scale cancer RNA-Seq datasets, apart from TCGA, are not commonly available) would be valuable for confirming the GI observation.

We have commented on this in the revised Discussion (new sentences in blue):

"We found a significant variability of iNMDeff across normal tissues and cancer types. This suggests tissue-specific regulation of the NMD pathway, which is broadly conserved between cancers and healthy tissues. Gastrointestinal tissues showed higher iNMDeff, and within them the hypermutated MSI colon and stomach cancers were among the ones with highest NMD efficiency. This is in agreement with a reported overexpression of NMD factors in MSI primary colon tumors, and that chemical inhibition of NMD caused a reduction in MSI tumor growth⁴⁰. Supporting this, studies have shown higher NMD pathway scores in MSI- H versus MSS tumors⁸², while UPF1 overexpression in CRC has been linked to metastasis and inhibition of apoptosis⁸⁴ and chemoresistance to oxaliplatin⁸³, with implications for patient survival. Together, this suggests that colon tumors, especially MSI (speculatively, due to higher frameshift mutation burden), necessitate a higher NMD efficiency to survive, supporting that targeting the oncogenic activity of NMD could be a viable, personalized treatment strategy for some cancers⁴⁶. Whether SNV/indel burden or other genomic features predict responses of an individual tumor to NMD inhibitors remains to be determined in future work. We acknowledge the need for additional validation of increased NMD in digestive tissue tumors through experimental models or analysis in additional cancer datasets."

Reviewer #2:

In this study, Palou-Márquez and Supek develop and benchmark two computational methodologies - based on ETG (endogenous target gene) and ASE (allele-specific expression) - to quantify the efficiency of nonsense-mediated mRNA decay (NMD) using exome and RNA-seq data from the TCGA and GTEx cohorts. Their methods appear robust, with a careful description of both the methodology and the strengths and weaknesses of each approach detailed in the Methods section.

Using these methodologies, they explore NMD efficiency variability across various tissues and tumors in human individuals. The rigor of the study is evident, as they take care to validate their

findings. For example, a weak but significant association between copy number alteration (CNA) gains and reduced NMD efficiency is validated using an independent dataset from the Cancer Cell Line Encyclopedia.

The study concludes by investigating the potential of global NMD efficiency measures as biomarkers for predicting immunotherapy outcomes in certain cancer types, showing some promise.

Disclaimer: While I am familiar with the NMD mechanism, I have only a general understanding of the statistical techniques used in this study. Therefore, my evaluation is more focused on the readability and biological aspects than the statistical rigor of the methods.

Overall, I am positive about the work, especially with regard to the robustness of their methods and the conclusion that there is substantial variability between tissues and individuals. However, I remain somewhat skeptical about the practical utility of these methods as biomarkers when applied to individual patients. That being said, this does not detract from the merit of the study, and future research may well demonstrate its broader applicability.

We thank the reviewer for their positive evaluation of our study and for highlighting the robustness of our methodologies as well as the potential of NMD efficiency measures as biomarkers. Please see our responses to the reviewer's thoughtful comments below.

Major Specific Comments:

Figure Presentation and Citation Order: It can be confusing that figures and figure panels are not cited in sequential order. For example:

Thank you for pointing out these mistakes, we apologize for the oversight. They have been corrected and properly cited in the manuscript.

- Supplemental Figure S3 is cited before Supplemental Figure S2.

Corrected

- Figure 2D-E is cited before Figure 2C.

We have changed Figure 2 and moved some panels to Supplementary materials. Now it is correctly cited.

- Figure 3F is cited before 3E.

We have swapped panels.

- Figure 4E is cited before 4C-D. It would be clearer if figures were cited in order.

We have changed the panels.

Interpretation of Results - NMD Efficiency and Chemotherapy vs. Immunotherapy: In the section discussing chemotherapy and immunotherapy (p.28, l.16-19), it seems the data from Figure S21B-

C indicates that chemotherapy outcomes may be more affected by NMD efficiency than immunotherapy. The current interpretation seems somewhat biased toward immunotherapy. A more balanced interpretation would be helpful.

Indeed, we focused our interpretation towards immunotherapy, given that it coincides with previous research linking NMD with immunotherapy (Lindeboom, R. et. al. 2019, Litchfield, K. et. al. 2020). Basically, there was a strong prior on that NMD efficiency -- here, uniquely, studied as a naturally variable trait across tumors/individuals -- would connect with immunotherapy.

To address this concern:

Firstly, we added additional data to support the association of NMD efficiency with immune reactivity. Here, in lieu of immunotherapy response data, we assessed the more abundant data describing CD8+ cytotoxic T-cell infiltration (inferred from RNA-Seq in TCGA cohort using quanTIseq TIL10 signature, considering samples with available treatment data i.e. the ones already shown in previous panels Fig. 6C, D). We noted positive trends towards higher CD8+ infiltration in those tumors having lower NMD efficiency, compared to tumors with higher NMD efficiency (Fig. 6B). This was seen in cancer types commonly considered immunogenic -- melanoma and kidney cancers -- and interestingly this association between CD8+ and iNMDeff was seen also in breast cancers.

We have additionally performed the same type of analysis on the Hartwig Medical Foundation cohort, for the samples which had RNA-Seq available (where it was processed using UniCell Deconvolve): in this cohort we did also observe highly significant positive association of various CD8 cell subtypes content, with lower iNMDeff in the tumor (new Supplementary Fig. S28).

We mentioned the result in the main Result text (new text in blue) “We *infer* that patients with lower iNMDeff might experience better treatment outcomes, as higher NMD efficiency results in the *silencing* of truncated peptides that *might* act as neoantigens, essential for the efficacy of immunotherapy. *Consistent with this notion, when assessing the data describing CD8+ cytotoxic T-cell infiltration (inferred from RNA-Seq in the TCGA cohort patients with treatment data), we noted positive trends towards higher CD8+ infiltration in those tumors having lower NMD efficiency. This was observed in cancer types commonly considered immunogenic – melanoma and kidney cancer – and interestingly also in breast cancer (Fig. 6B). These findings were further validated in the independent Hartwig cohort, where we observed significant increases in content (estimated via RNA-Seq) of multiple CD8+ T-cell subsets in the lower iNMDeff tumors, across breast, kidney, lung and skin cancers (Supp. Fig. S28).*”.

Secondly, we made clarifying edits to the text: we do agree more emphasis is due on the less-expected (but not completely unanticipated, see DOI: 10.1038/ncomms7632) importance of individual-level NMD efficiency on chemotherapy outcomes, and we have corrected the text to balance this interpretation.

We changed this sentence in the results section: “*Thus, we cannot exclude that efficient NMD might be associated with poor response of melanoma not only to immunotherapy, but also to chemotherapies*” to: “*Efficient NMD is more strongly associated with poorer tumor response to chemotherapies compared to immunotherapies*”

We have added this sentence in the revised Discussion (new part in blue text): “*We do note that the iNMDeff associations with efficiency of conventional chemotherapy, which may or may not be mediated by immune*

mechanisms, are stronger than immunotherapy. Speculatively, NMD may confer a generally increased tumor cell fitness in the context of occurrence of deleterious proteins that arose via somatic mutation (as shown before for one example, the HSP110DE9 chaperone mutant⁴⁰), conferring additional ability to withstand various challenges to cancer cell fitness. In line with this, transient pre-treatment with NMDI-1 prior to administering doxorubicin demonstrated increased cell death compared to doxorubicin alone¹²⁰. This suggests that naturally reduced NMD efficiency may predispose tumor cells to an enhanced cytotoxic response to chemotherapy, consistent with better survival outcomes for individuals bearing tumors with inherently low NMD activity."

Positive Selection on NMD-Evading Mutations: The analysis of NMD-triggering mutations seems a bit biased. It would be helpful to provide an explanation for why there is similar (Figure 6A) or even stronger (Supplemental Figure S19A) positive selection on NMD-evading mutations.

We agree that the reviewer mentions -- that nonsense mutations are positively selected in NMD-evading gene regions, similarly as mutations in NMD-triggering gene regions -- merits further discussion.

Before we discuss, let us mention that this observation (difference between NMD-triggering and NMD-evading regions, Fig 6A compare left and right sub-panel) does not affect the central result of this analysis: that there is an effect of inter-individual NMDeff variation on the somatic selection in driver genes. Namely, lower iNMDeff associated with significantly lower dN/dS (comparison between red and blue boxplot) in the NMD-triggering regions (left-subpanel), but this difference based on iNMDeff is reassuringly not observed with mutations in the NMD-evading region, which serve as one negative control, nor in missense mutations, serving as another negative control. This association of selection with iNMDeff stands, regardless of positive selection on NMD-evading regions (which is interesting as a side observation, as the reviewer mentions).

Why NMD-evading nonsense mutations are positively selected: as a caveat, we cannot fully exclude some misclassification of NMD-evading mutations, where some of them are actually NMD-triggering to some extent. For the long-exon NMD evasion rule, it is known that it is a weaker rule with only partial NMD evasion (Lindeboom, R. et. al. 2019). Aiming to improve accuracy, we modified the standard <400nt versus >=400nt threshold of the long-exon rule, to use more stringent criteria: >=1000nt for NMD-evading and <= 500nt for NMD-triggering mutations. We created a "buffer zone" by excluding mutations in exons of lengths between 500-1000nt, reducing potential misclassification. Upon updating the thresholds, we observed a clearer difference between High and Low groups ETG iNMDeff for NMD-triggering nonsense mutations (ETG p-value from 0.2 to 0.05, see plots below; and ASE p = 0.47 to 0.13). In other words, there is a robustly higher selection signal of NMD-triggering nonsense mutations in tumor suppressor genes in the group of samples

with higher NMD efficiency, compared to the group of samples with lower NMD efficiency. Revised Fig 6A panel below:

Fig 6. Individual-level NMD efficiency modulates somatic selection in tumor suppressor genes and impacts progression-free survival (PFS) and response to immunotherapy. A, The dN/dS ratios of tumor suppressor genes (TSGs) for both NMD-triggering and NMD-evading nonsense (left panel) and missense (right panel) somatic mutations are compared between two groups of individuals with high and low iNMDeff, as determined by the median ETG iNMDeff. Statistical significance is assessed using one-sided Wilcoxon tests on paired data. PTCs were classified as NMD-triggering if they met all criteria: (1) >250nt from TSS, (2) in exons ≤500nt, and (3) not in the last exon or last 55nt of the penultimate exon. Conversely, PTCs were classified as NMD-evading if they met any criteria: (1) ≤250nt from TSS, (2) in exons ≥1000nt, or (3) in the last exon or last 55nt of the penultimate exon. **B,** [...]

This has been better clarified in Methods (“*Estimation of selection in cancer genes via dNdS*”) and it is additionally mentioned in the figure legends (Fig. 6A and Supp. Fig 26). See below full text:

Methods:

“To enhance the accuracy of our classification between NMD-triggering and NMD-evading PTCs, we incorporated a new NMD-rule⁸ acknowledging that PTCs in longer exons might escape NMD. Accordingly, PTCs located in exons with a length ≥ 1000 nt were categorized as NMD-evading. Conversely, PTCs considered NMD-triggering were restricted to exons with a maximum length of 500 nt.

→

“To enhance the accuracy of our classification between NMD-triggering and NMD-evading PTCs, we incorporated a new NMD-rule⁸ acknowledging that PTCs in longer exons might escape NMD. PTCs were classified as NMD-triggering if they met all criteria: (1) >250nt from TSS, (2) in exons ≤ 500 nt, and (3) not in the last exon or last 55nt of the penultimate exon. Conversely, PTCs were classified as NMD-evading if they met any criteria: (1) ≤ 250 nt from TSS, (2) in exons ≥ 1000 nt, or (3) in the last exon or last 55nt of the penultimate exon. For the weaker long-exon rule of NMD evasion, we created a “buffer zone” by excluding mutations in exons of lengths between 500-1000nt, reducing potential misclassification.”

Nonetheless, we still observed a high positive selection in the NMD-evading nonsense mutations for the group of samples with low iNMDeff as well as the group of samples with high iNMDeff, presumably because

many of the truncating mutations are deleterious to protein function regardless of NMD silencing at mRNA level. An explanation for the observed trend is that, simply, these truncating mutations lead to complete loss-of-function of the protein even without mRNA being degraded; indeed many truncations would be expected to do that. Some of the truncations might even have dominant-negative effects, where the truncated version of the protein blocks function of the wild-type allele (Liu, Y. & Bodmer, W. F. 2006; Briones, A. C. et. al. 2024).

We emphasize again, as written above, the significant, and meaningful difference between tumor samples with high and low NMD efficiency in selection on nonsense mutations (compare red and green box plot in Fig. 6A), and that this difference is observed only in NMD-triggering regions but not in NMD evading. We have clarified this in the Discussion section, mentioning the possibility of misclassifying some mutations as NMD-triggering or NMD-evading (e.g. via the weak “long-exon” rule), as well as that NMD-evading nonsense mutations often do have sufficient loss-of-function to be positively selected (NMD silencing is not always necessary for tumor suppressor gene LoF), see below:

Discussion (new text added):

“As a side observation, genes with NMD-evading PTC variants do also exhibit positive selection, but, importantly, there is not a significant difference between tumor samples with high and low iNMDeff (Fig. 6A). This emphasizes that the effect of inter-individual NMDeff variation on the somatic selection affects only driver genes with NMD-triggering PTC variants. The positive selection of NMD-evading nonsense mutations could be explained by the fact that these truncating mutations can be advantageous despite evading NMD, either through complete loss-of-function effects even without mRNA degradation, or through dominant-negative mechanisms^{118,119}.”

Methodology - Discovery vs Validation: In Figure 5C and Supplemental Figure S16B, there are individuals in the validation dataset with no corresponding individuals in the discovery dataset (e.g., LAMC1 in the ASE plot). Are all discovered genes/tissues from ASE and ETG subjected to validation by either method?

Thanks for pointing this out, as our testing setup may not have been the most clearly described. All genes/tissues were tested using ASE/ETG in one cohort (TCGA vs GTex) and validated in the other cohort, when enough rare germline variants in the gene or ASE iNMDeff estimates for the sample were available (reminder that with ASE, ~50% of samples have no iNMDeff score compared to ETG, due to the stringent filtering criteria in ASE method). This is done using both ASE and ETG methods, and also performed vice versa changing the cohort's direction (which is discovery which is validation), for a total of 8 combinations. However, we consider an association validated for a result identified by and iNMDeff method (e.g. ASE), whether it is the same method as the discovery (here, ASE) or the other method (here, ETG), meaning we allow for “cross-method” validation of hits as long as the hit is seen in both TCGA and GTex. In this particular case, for LAMC1 association, it was discovered in brain GTex using ETG method, and was further validated in brain tumors TCGA using ASE method. See Supp. Fig. S23B (relevant data from that figure is given in table below):

iNMDeff method used in the discovery cohort	iNMDeff method used in the replication cohort	Significant hits
---	---	------------------

ASE	ASE	KDM6B, FIG4, CCDC114
ASE	ETG	PDIA2,PLB1,COL19A1
ETG	ASE	LAMC1, NUP153
ETG	ETG	PYGM, PXDN, TRAP1, CRTC1

We have added the following sentence to the “Rare variant association analysis identifies 10 genes associated with NMD efficiency” Results section to clarify:

“Notably, replication did not require the use of the same iNMDeff method as in the discovery cohort. For instance, a significant association detected using the ASE iNMDeff method in TCGA could be validated in GTex using either ASE or ETG or both iNMDeff methods.”

Significance of Candidates - Randomized Procedure: On page 22, the study identifies 48 significant candidates, while the randomized version produces 24 significant candidates. Is this difference useful or significant? Further discussion of the implications of this finding would be beneficial.

Thank you for your comment. This difference is indeed useful, as the randomization test serves as a baseline to estimate the expected number of significant genes under no association between rare pLoF germline variants in various genes and iNMDeff. The significant candidates from the randomized test represent false positives, while the observed excess over this baseline quantifies the fraction of true associations in the test (1-FDR).

To assess the significance of this difference, we compared the observed proportion of significant genes ($p_{\text{obs}} = 48/1072334$) with the randomized proportion ($p_{\text{rand}} = 24/1035154$) and calculated the Z-score and its associated p-value (Methods).

The resulting $p=7e-03$ indicates that the observed difference is statistically significant, supporting the notion that the observed number of significant genes (48) exceeds what would be expected by chance (24).

We have changed the following sentence in the manuscript to clarify this point:

“This is in contrast to the 24 significant associations found by randomizing the iNMDeff estimates (Supp. Fig. S16A)” →

“This is significantly higher compared to the 24 significant associations found by randomizing the iNMDeff estimates ($p = 7e-03$, Supp. Fig. S23A), indicating that the observed number of significant candidates exceeds the expected baseline under randomization, supporting the presence of non-random associations between iNMDeff and germline rare variants.”

Method Section - NMD Efficiency Metric: The formula for pNMDeff (p.41, l.7) could benefit from clarification. I understand that a score of 0 means no mRNA degradation, but why does a score of

1 represent complete heterozygous mRNA degradation? A score of 1 would imply that the mutant RNA-seq allele counts are half of the wild-type RNA-seq allele counts. Please clarify this point.

This was correctly noted. A score of 1 implies that the MUT allele has half the counts of the WT allele, while in fact, a complete heterozygous mRNA degradation approaches infinity, not 1. We have corrected and clarified this in the methods section:

“A pNMDeff score of 0 indicates no mRNA degradation, while a score of 1 indicates complete heterozygous mRNA degradation via NMD.” → “A pNMDeff score of 0 indicates no mRNA degradation, a score of 1 indicates that the MUT RNA-seq allele counts are half of the WT allele, while a complete heterozygous mRNA degradation approaches infinity.”

Minor Specific Comments:

Figure 1: Explain the abbreviations used in the figure within the legend.

Corrected, thank you.

Figure S1B: It seems there is a typo; ">=uORF 30 bp?" should likely read ">=2 uORF 30 bp?"

Corrected, thank you.

Page 8, Line 17: A reference to Figure 2B is missing.

Corrected, thank you.

Figure 2C - Variants: The presented variants (4 and 10 for the two individuals) should be clarified. Do these represent all detected PTC variants in the samples?

These represent the remaining PTC variants after the filterings we perform to obtain a robust set of PTCs used for the ASE iNMDeff method (see Methods section, in brief, we exclude a PTC if: homozygous variant, coverage < 5 reads, variant population allele frequency > 20%, has a co-occurring somatic truncating mutation and/or CNA in that tumor sample, located in a non-coding transcript or a transcript with only a single exon; located in a gene from a set of 331 positively selected genes, a set of negatively selected genes based on first decile by LOEUF score). This has been clarified in the revised text, as follows:

“For individual “TCGA-F4-6856”, who was labeled as having a high iNMDeff, we observe four PTC variants with notably low RNA-seq counts of their alternative alleles, relative to the reference allele.” →

“After applying our ASE filtering criteria for germline PTC variants (see Methods) we retain four PTC variants with notably low RNA-seq counts of their alternative alleles, relative to the reference allele, for the individual “TCGA-F4-6856”, who was classified as having a high iNMDeff”

Page 10, Line 6: A reference to Figure S4D is missing.

Thank you very much, corrected.

Figure S6A-B: It would be helpful to explain what the yellow and green-ish bars represent in the Pearson correlation plots. Additionally, is there any trend observable in these correlations?

In the revised manuscript, this barplot was removed and the overall pan-tissue/pan-cancer correlations between log₁₀(TPM) gene expression (*RBFOX3* and *AQP4*) and ETG iNMDeff shown instead (Supp. Fig. S11A-B for TCGA and GTex). We observed that the neuron-specific marker *RBFOX3* is negatively correlated with iNMDeff (TCGA: $R = -0.31$, $p = 4.4e-15$; GTex $R = -0.3$, $p < 2.2e-16$), whereas the glia-specific marker *AQP4* is positively correlated with iNMDeff (TCGA: $R = 0.15$, $p = 2.2e-4$; GTex $R = 0.32$, $p < 2.2e-16$). This supports that neural cells, rather than glia or other cells present in these tissue samples, are the cell types with lowered NMD activity in brain cancers/tissues. We have now mentioned the *RBFOX3* and *AQP4* associations, as markers of neuron and glia populations respectively, as support for that specifically neurons rather than glia are the cells with lower NMD efficiency:

“The ETG iNMDeff correlations with estimated fractions of neurons across these cancer subtypes were negative (Fig. 3E; R of -0.3 , $p = 1.5e-14$, in ETG), supporting that neural cells, rather than glia or other cells infiltrating these tumor samples, are the cell types with lowered NMD activity (Supp. Fig. S6A-B).” →

*“The ETG iNMDeff correlation with estimated fractions of neurons across these cancer subtypes was also negative (Fig. 3F; $R = -0.3$, $p = 1.5e-14$, in ETG). As an additional analysis, we assessed the expression of the neuron-specific marker *RBFOX3*, which was negatively correlated with iNMDeff (TCGA: $R = -0.31$, $p = 4.4e-15$; GTex $R = -0.30$, $p < 2.2e-16$), whereas the glia-specific marker *AQP4* was positively correlated with iNMDeff (TCGA: $R = 0.15$, $p = 2.2e-4$; GTex $R = 0.32$, $p < 2.2e-16$), supporting that neural cells, rather than glia or other cells infiltrating these tumor samples, are the cell types with lowered NMD activity in the central nervous system (Supp. Fig. S11A-B).”*

Page 14, Lines 42-44: The sentence about reduced variance in NMD efficiency is difficult to understand. Consider rephrasing it for clarity.

We agree this was unclear. We have rephrased it as follows:

“The reduced variance over random is considerable in both tests, suggesting that there is considerable between-PTC as well as, more interestingly for the matter at hand, inter-individual variation in NMD efficiency.” →

“Compared to the randomized controls, we observe a significant reduction in variance for both tests, reflecting the underlying rules that govern NMD efficiency. This finding underscores two key points: first, that there exists unexplored systematic variation in NMD efficiency between different PTCs even after accounting for known NMD rules, and second - more relevant to the focus of our study - that there is substantial variation in NMD efficiency between individuals.”

Page 14, Lines 46-50: This sentence may be missing a reference to Supplemental Figure S6C.

Added the reference, thank you. We also swapped the order of the panels (C and D) because they were misordered.

Page 24, Line 12: "In specific" should be changed to "Specifically."

Found in two places and corrected both, thank you.

Page 30, Lines 31-35 - Sentence Structure: The sentence discussing MEN1 gene and NMD activity should be split into two sentences for clarity.

We have split the sentence:

“A previous study in mice monitored the levels of MEN1 gene and based on this reported NMD activity in brain, testis, ovary and heart, our comprehensive analysis across many human tissues suggests that considering multiple genes and mutations is necessary to reach robust conclusions on tissue specificity.”

→

“A previous study in mice monitored NMD activity through the expression levels of a single gene, MEN1, across seven tissues including brain, testis, ovary and heart. However, our comprehensive analysis across many human tissues suggests that considering multiple genes and mutations is helpful to reach robust conclusions about tissue-specific NMD activity.”

Page 31, Line 15: The phrase "...also contains NMD factors SMG5, SMG7..." should be modified to "...also contains genes encoding NMD factors SMG5, SMG7..."

Corrected, thank you.

Figure S15B+D: The y-axis label might represent the negative log₁₀ of the FDR values. Please verify and clarify.

Sorry for the confusion, it is corrected now, thanks.

Methods Section - NMD-Related Genes: A table summarizing all NMD-related genes used in the study should be included.

Indeed. We have added this table (Supplementary Table S7) of the 112 NMD-related genes sourced from Alexandrov et. al. 2017, Baird et. al. 2018 and general knowledge about core NMD factor genes (see reviews doi: [10.1007/s00018-015-2017-9](https://doi.org/10.1007/s00018-015-2017-9) and doi: <https://doi.org/10.1038/s41580-019-0126-2> as examples). We have additionally included new supplementary tables (all tables were reordered accordingly), see below:

S1: ASE and ETG individual NMD efficiency (iNMDeff) estimates for all TCGA samples. Table includes covariates and samples metadata

S2: ASE and ETG iNMDeff estimates for all GTex samples. Table includes covariates and samples metadata

S3: ETG-based estimates of cell NMD efficiency (cNMDeff) for UPF1 knock-downs and WT cell types S4:

our "Consensus" set of genes, listing the NMD target transcripts and paired control transcripts thereof

S8: ETG-based predictions of cNMDeff for CRISPR KO data from the Cancer Cell Line Encyclopedia (CCLE) (DepMap project)

Page 46, Line 59: The word "redefinment" should be changed to "redefinition."

Corrected, thank you very much.

Reviewer #3:

In this paper, the authors analyzed thousands of tumor and healthy tissue samples to examine variations in nonsense-mediated mRNA decay (NMD) efficiency across different tissue types and between individuals. Their findings suggest that NMD efficiency is lower in the nervous and reproductive systems compared to the digestive system, with potential genetic influences from copy number changes in core NMD genes and germline mutations in genes like KDM6B. However, there are significant concerns about the accuracy and reliability of their NMD efficiency measurements. These concerns raise questions about the robustness of the authors' conclusions.

We thank the reviewer for the summary of the main findings of our work. We aimed to address the various concerns of the reviewer below, by performing additional analyses (incl. involving additional external datasets). These new data do support that our findings on NMD efficiency variation between tissues and individuals are, indeed, robust. Please see our point-by-point response below.

The major concerns include:

(1) Their definition of NMD efficiency may not really capture the strength of NMD targeting on mRNA degradation. There is no validation/proof to show their calculated measurements really represent NMD efficiency. Their proposed NMD efficiency is roughly defined as the average expression negative log ratio of NMD targeted transcripts vs. control transcripts for both the approach using endogenous target gene and the one using allele-specific expression.

The reviewer here summarizes our methods for assessing NMD efficiency: (i) “ETG” (endogenous target gene) which is negative log ratio of mRNA levels for NMD target transcript vs control transcript of the same gene; and (ii) “ASE” (allele-specific expression) which is negative log-ratio of mRNA levels of mutated allele vs wild-type allele.

We would in principle agree that differences between the mRNA levels of transcripts could have been caused by other factors apart from NMD. Crucially, however, the transcripts have been chosen such as to reflect specifically the known NMD features:

(i) for the ETG (“endogenous target gene”) method, we used specifically genes which were shown, in previous experimental work, to respond to NMD factor perturbation. We selected genes that were reported in at least 2 out of 4 comprehensive studies (Colombo et. al. 2017, Tani H. et. al. 2012, Karousis E. D. et. al. 2021, Courtney et. al. 2020), called the “NMD Consensus” set used for most of our analysis. Within this known NMD target gene set, for every ETG gene we selected one an isoform which bears known NMD-promoting features (such as uORF at 5’UTR or intron within 3’UTR, as per ENSEMBL definition). In brief (see Methods for details) we normalize the mRNA level of the NMD-feature bearing transcript to another transcript of that same gene that does not bear obvious NMD-promoting features. Overall, in our ETG method the choice of genes based on prior experiment, and the choice of transcript based on NMD features, strongly enrich for the NMD signal.

(ii) for the ASE (“allele-specific expression”) method, again there are two ways in which we have enriched for the NMD signal. First is simply the fact that we are measuring allelic imbalance in mRNA levels (rather than overall mRNA levels) of a certain gene -- this means that factors that act on gene expression *in trans* are controlled for, since they are likely to affect both alleles similarly. Secondly, we use only nonsense

mutations in our analysis, and of those only the nonsense mutations that reside in the NMD-triggering regions of a gene. This means that the effect of these mutations on mRNA level are likely to be exerted via nonsense-mediated mRNA turnover, rather than e.g. disruption of an intragenic transcriptional enhancer caused by the same mutation (which indeed we cannot rule out formally, in all individual cases, however another study from the group [<https://doi.org/10.1101/2024.09.07.611780> and Fig 5C therein] reports, using the Puffin-D neural net predictive model, that coding exonic mutations in cancer extremely rarely change transcription levels to the extent comparable to e.g. known *TERT* promoter driver mutations).

Finally, we would like to note that by use of these two methods ASE and ETG, and also two independent datasets (one tumor, one healthy) to which both of the methods are applied, we ensure that the readouts do fairly faithfully represent NMD efficiency in a given individual or tumor. Below we provide additional data to support this is the case.

Note that expression measurements from RNA-seq are just static snapshots, which do not capture the full dynamic processes of transcription, splicing, and degradation. As a result, it is problematic to claim that NMD efficiency is lower in the nervous system. For example, if transcription in nervous tissues generates more NMD-targeted transcripts, they could appear in high quantities even if NMD efficiency is actually high, which would artificially produce a lower NMD efficiency score under this definition.

We thank the reviewer for drawing attention to that differential RNA-seq levels are an aggregate of various stages in the mRNA life cycle: transcription, splicing, and degradation (via NMD or otherwise).

Our analyses aimed to isolate the NMD signal in the differential mRNA expression, by drawing on the various rules, listed in our answer above (ETG method: use of experimentally validated NMD target genes, and restricting to NMD-feature bearing transcripts; ASE method: use of allelic imbalance to control for trans effects, and use only of NMD-triggering nonsense mutations not other mutations).

More generally, we do agree that computational analyses of large-scale genomic data require careful validation, lest they be confounded. In our original study we presented multiple lines of evidence supporting the robustness of our methodology (listed in 1-3 below).

In the revised study, we have added 2 additional analyses (listed as points 4-5 below) to further support that what we are measuring is, in fact, largely NMD signal.

Analyses supporting that we are observing NMD and not differences in e.g. transcriptional output or splicing, which were already in the original manuscript and we here provide an overview for convenience:

1- We compared the NMD readout with three negative control sets: (i) genes without NMD-triggering features in transcripts (“Random Genes”) for the ETG method; (ii) synonymous mutations and (iii) NMD-evading PTCs for ASE method. The *iNMD*eff estimates using these sets were close to 0 (scaled), meaning there was no difference in average negative log expression between NMD target transcripts (for (i)) or between mutations and controls (for (ii) and (iii)) (Fig. 2A-B, Supp. Fig. S2A-C). This lack of differences with control genes/mutations (i-iii) contrasts with significant differences observed using known NMD target genes/transcripts from NMD “Consensus” set in ETG method, which contains validated NMD targets from ≥ 2 experimental studies, or with PTC mutations (ASE method), which are located in gene regions sensitive to NMD. Presumably, mutations in the control set of mutations (ii, iii) would likely have similar chances of

affecting transcription output (e.g. via enhancer perturbation, etc) and/or splicing, as the NMD affecting PTCs, but their ability to trigger NMD differs, because they are not nonsense mutations at all (for ii) or they are in NMD-evading regions (for iii).

2- Out-of-sample marker gene expression supports our analysis. We demonstrated that individuals with high iNMD_{eff} show consistently lower expression of two well-established NMD marker genes (*RP9P* and *GAS5*; Fig. 2C and Supp. Fig. S2D), which were not part of our ETG “Consensus” set of genes. *GAS5* was first described upon deletion of *UPF2* in two cell types (Weischenfeldt J. et. al. 2008), and also identified in a transcriptome profiling of knockdowns and rescues of several NMD factors (Tani et. al. 2013). *RP9P* was found in the transcriptome profiling of KDs (Karousis E. D. et. al. 2020), and confirmed in RT-qPCR from *SMG1* KD cells (Soumasree De et. al. 2022), and in *SMG7* KO (Nasif S. et. al. 2023).

3- We explicitly tested associations of iNMD_{eff} estimates with a variety of potential confounding factors, both biological (e.g. mutation burdens) or technical (e.g. RNA-seq library sizes) and found little or no correlation with these known factors.

Neither the ETG nor the ASE method showed strong correlations with age, sex, tumor mutation burden (TMB, for TCGA only), tumor indel burden (TIB, for TCGA only), tumor nonsense burden (TNB, for TCGA only), tumor purity (for TCGA only), MSI score (for TCGA only), RNA-seq sample library size, leukocyte fraction (for TCGA only), CNA burden (for TCGA only), days to death (TCGA), death group (GTex), and number of NMD targets that passed filtering criteria and that were used to estimate iNMD_{eff} (number of PTCs for ASE, or number of transcripts for ETG) (Supp. Fig. S6B-C). For the few variables that did register any appreciable effect, the effect was small (CNA burden R = -0.10 in TCGA; RNA-seq sample library size R = -0.13-0.16; age R = -0.15 in GTex), and as a precaution we included them as covariates in all our downstream association analysis.

In the revised study, we have performed 2 additional analyses to validate that our estimates are indeed capturing the strength of mRNA degradation by NMD in specific, rather than e.g. the transcriptional output the reviewer mentions as an alternative explanation.

4- We analyzed NMD efficiency in additional RNA-seq datasets from various cell lines with *UPF1* knockdown (KD), as *UPF1* is a central NMD factor, comparing ETG method estimates between *UPF1* KD and the isogenic matched wild-type cell line. We analyzed RNA-seq data from three cell lines (HeLa, HepG2, K562) sourced from different studies (see table below), processing the raw data using the same methodology as our main analysis. That means, we performed *fastq* short read mapping to human genome, transcript-level RNA-seq quantification and gene annotation with the same methods and versions as in TCGA and GTex datasets (see Methods).

Cell Line	GEO accession	Wild-type N replicates	UPF1 KD N replicates	KD type	Source
HeLa	GSE152435	5	5	siRNA	DOI: 10.1101/gad.338061.120
HeLa	GSE86148	3	3	siRNA	DOI: 10.1261/rna.059055.116
K562	GSE88140	0	2	shRNA	DOI: 10.1038/nature11247

K562	GSE88266	2	0	shRNA	DOI: 10.1038/nature11247
HepG2	GSE88148	2	0	shRNA	DOI: 10.1038/nature11247
HepG2	GSE88466	0	2	shRNA	DOI: 10.1038/nature11247

Using our NMD Consensus ETG set of NMD target genes (n=130 target-control transcript pairs i.e. 260 transcripts were considered), we applied negative binomial regression to estimate cell line NMD efficiency (cNMDeff), paralleling the ETG iNMDeff approach in our main analysis (see Methods). We observed a remarkably clean separation of cNMDeff scores between *UPF1* KD and wildtypes; see new panel in Figure 2D and also provided below. As predicted, *UPF1* knockdown significantly reduced NMD efficiency compared to wild-type cells in HeLa ($p = 1.4e-06$), which had 8 replicates available. Moreover, we saw a consistent and strong effect in the other two cell lines that had 2 experimental replicates each, K562 ($p = 2.2e-2$) and HepG2 ($p = 0.11$). Notably, every *UPF1* KD experiment in any cell line showed a lower value of our cNMDeff score than any instance of a wild-type cell line, thus the inter-cell type variation was minor compared to the extent of the NMD signal captured by our ETG method.

D

Fig. 2. Individual-level quantification of NMD efficiency.

[...]

D, Cell line ETG NMD efficiency (cNMDeff) estimated in HeLa (n = 8), HepG2 (n = 2), and K562 (n = 2) cell lines using the “NMD Consensus” gene set (n = 130). cNMDeff was calculated using negative binomial

regression, similar to our *iNMDeff* main analysis (see Methods). Comparison between *UPF1* knockdown (KD) and wild-type (WT) conditions showed consistent reduction in NMD efficiency across all cell lines. Barplots and 95% confidence intervals represent the mean across cells.

[...]

These additional results support that the ETG method robustly distinguishes between depleted and functional NMD activity across individuals, using an external dataset from controlled experiments, where confounders (tissue, subtype, environment, genetic background etc.) did not play a role. As an additional analysis to confirm the robustness of the ETG Consensus NMD gene set, we iteratively removed one gene at a time from our NMD Consensus set and repeated the analysis with the remaining genes. This process, repeated 130 times (once for each gene in the set), consistently reproduced the differences between *UPF1* KD and wild-type cells, demonstrating that our results are not driven by outlying genes (see plot below) but rather stem from a property of multiple genes in the NMD consensus gene set.

A

Supp. Fig. S3. Validation of the ETG *iNMDeff* method

A, Cell line ETG NMD efficiency (*cNMDeff*) in HeLa ($n = 8$), HepG2 ($n = 2$), and K562 ($n = 2$) cell lines using the “NMD Consensus” gene set. *cNMDeff* was calculated using negative binomial regression, similar to our *iNMDeff* analysis (see Methods). A leave-one-out validation was performed on 130 NMD Consensus genes, with sequential removal of one gene at a time followed by *cNMDeff* recalculation. Comparison between *UPF1* knockdown (KD) and wild-type (WT) conditions showed consistent reduction in NMD efficiency across all cell lines, independent of gene outliers. Barplots and 95% confidence intervals represent the mean across cells. **B**, [...]

We have incorporated these findings into Figure 2 panel D and Supp. Fig. S3A and mentioned these data in the first results section (section “Robustness of the individual NMD efficiency estimates”), see text below.

Results:

“To validate our ETG method for iNMDeff estimation, we analyzed RNA-seq data from three cell lines (HeLa, HepG2, K562) sourced from different studies^{5,53,54} and processed the raw data using the same methodology as our main analysis (see Methods). Using our NMD Consensus set of NMD target genes, we applied negative binomial regression to estimate cell line NMD efficiency (cNMDeff), paralleling the iNMDeff in our main analysis (see Methods, Supp. Table S3). We observed a remarkable separation of cNMDeff scores between UPF1 KD and wild-type conditions (Fig. 2D). As predicted, UPF1 knockdown significantly reduced NMD efficiency compared to wild-type cells in HeLa cells ($p = 1.4e-06$, $n = 8$ replicates), with consistent effects in K562 ($p = 2.2e-02$, $n = 2$) and HepG2 ($p = 0.11$, $n = 2$). Notably, every UPF1 KD cell showed lower cNMDeff than any wild-type cell. As an additional analysis to confirm the robustness of the ETG Consensus NMD gene set, we performed leave-one-out analysis on our 130 NMD Consensus genes, consistently reproducing the UPF1 KD versus wild-type differences (Supp Fig. S3A), demonstrating that our results are not driven by an outlying gene but rather stem from a consistent property of the NMD target gene set.”

Methods:

We also described the analysis in detail in the Methods section (“Validation and robustness of ETG iNMDeff estimates”), please refer to the attached manuscript with tracked changes for new text added (omitted here for brevity).

By using external datasets from controlled experiments, we ensure that confounders (e.g. tissue, subtype, environment and genetic background) do not play a role. As an additional analysis to confirm the robustness of the ETG Consensus NMD gene set, we iteratively removed one gene at a time from our NMD Consensus set and repeated the analysis with the 129 remaining genes. Therefore, this process was repeated 130 times (once for each gene in the set).

5- In an additional analysis, we ruled out differential transcriptional output as a confounding factor in our estimation of NMD efficiency from mRNA levels. Based on the reasons we listed above, we think this should not be a common occurrence: there is not a prior that nonsense mutations would cause changes in transcription levels preferentially over causing NMD (as we measure in ASE method). Nor is there an expectation that transcription levels would be different (as measured in the ETG method) between the 2 transcripts based on whether a transcript has known NMD-enhancing features (e.g. 5'UTR uORF or intron at 3'UTR) due to some non-NMD effect.

We set out to more directly address the reviewer's concern that higher transcription rates could potentially lead to artificially lower NMD efficiency estimates, by producing more mRNAs of NMD-targeted transcripts.

To address this, we downloaded PRO-seq (Precision Run-On and Sequencing) data from three different cell lines (A549, U2OS, and SH-SY5Y). This is a high-resolution genomics method that measures active transcription by RNA polymerase II across the genome by capturing nascent RNA. PRO-Seq provides a direct measurement of transcription rates at nucleotide resolution. We used PRO-seq levels near the gene's promoter (TSS) and downstream region as the measure of promoter activity in driving mRNA transcription activity.

We applied this to the specific concern by the reviewer that the brain-specific (likely, neuron-specific) signal of lower NMD efficiency might be in part due to higher gene transcription rates of some transcripts. Here, we compared PRO-seq signal: on the one side, we compared SH-SY5Y neuroblastoma cells, which are of

neural crest origin and express various neuronal protein markers, thus serving as a model for neurons. On the other side, we compared the SH-SY5Y against A549 (lung) and U2OS (bone) cells. Specifically, we considered the genes in our ETG Consensus set of known NMD targets, whether the PRO-seq signal was different between the transcripts that were NMD targets and matched transcripts from the same gene which were not NMD targets, according to our definition of the ETG measure.

Additionally, we assessed if the transcription rate effects could confound specifically the brain-associated differential NMD efficiency (inferred via ETG method from mRNA levels in brain tissue). To this end, we stratified our analysis by comparing brain-upregulated versus non-brain-upregulated genes, using TPM gene expression from GTex (Supp. Fig. S3B upper panel) and TCGA (Supp. Fig. S3B bottom panel) to stratify genes.

There was no significant difference in PRO-Seq signal between NMD target transcripts and non-NMD transcripts, in either cell line. Moreover, this pattern held consistent regardless of whether we considered brain-enriched genes or non-brain-enriched genes. Importantly, this lack of significant difference in transcription rates (via PRO-seq) was observed neither in the neural-model cell line SH-SY5Y, nor similarly so in the other two cell lines (lung, bone). See new Supplementary Fig. S3B also shown below.

Cell Line	GEO study	Replicates	Type	Source
A549	GSM5169137	1	Lung cancer epithelial	ENCODE
U2OS	GSM5169140	1	Bone sarcoma	ENCODE
SH-SY5Y	GSE214243	2	Neuroblastoma	DOI: https://doi.org/10.1016/j.yexcr.2023.113536

B**Supp. Fig. S3. Validation of the ETG iNMDeff method**

[...] **B**, Transcriptional output analysis using PRO-seq scores in A549 ($n = 1$), U2OS ($n = 1$), and SH-SY5Y ($n = 2$ experiments; SH-SY5Y_1 and SH-SY5Y_2) cell lines. Promoter activity was compared between NMD-target and non-target (control) transcripts from the ETG Consensus gene set, measuring PRO-seq signals at TSS and downstream regions (see Methods). Brain-specific gene expression enrichment was assessed by comparing mean gene TPM values between merged normal brain subregions and all other tissues in GTex (top row) or between LGG/GBM tumors and other cancers in TCGA (bottom row) to define brain-enriched genes (performing Mann-Whitney U test, alternative = greater, $p < 0.05$). P-values on plot are by Mann-Whitney U test test.

Overall, the above suggests that our ETG estimate method, drawing on the “Consensus” set of NMD target genes that we use through the study, does not measure transcription output differences. In specific, our observation of differences between NMD efficiency in the brain (neurons in specific) versus other tissues using ETG method do not appear to be due to differences in transcription rates.

We have added the results of the new analysis as Supp. Fig. S3B, and added a brief description of these data to the Results (see section: “Robustness of the individual NMD efficiency estimates”) with an extended description in the new Supp. Text. S1 in the revised manuscript. We hope this addresses the reviewer’s concern. See below the new text:

Results:

PRO-seq analysis of nascent RNA transcripts from three cell lines^{55,56} showed no significant differences in transcription rates between NMD and non-NMD target transcripts, including when stratified by brain expression levels (Supp. Fig. S3B), confirming that our ETG measurements reflect NMD activity rather than transcription rates (see Supplementary Text S1).

Supplementary Text S1:

changes are omitted here for brevity; please refer to Supp. Text S1 file.

Methods:

We also described the PRO-seq analysis steps in detail in the Methods section (“Validation and robustness of ETG iNMDeff estimates”), please refer to the attached manuscript with tracked changes for new text added (omitted here for brevity).

(2) The agreement between the two methods the authors used to measure NMD efficiency is not satisfactory: only about 75% of the tissues/cancers have positive correlation. And the correlation based on pan-cancer or pan-tissue is low, at 0.14 and 0.19, respectively (as shown in Supp. Fig. S3). This low level of correlation raises concerns about the reliability of their NMD efficiency estimates.

This moderate albeit significant correlation between NMD scores obtained by ETG and ASE methods can be attributed to certain design choices that were deliberate.

Namely, there is a tradeoff between the precision in the measurements of the ASE estimate, and the number of data points available, since the ASE method relies on fairly uncommon events (nonsense variants, PTCs) to estimate NMDeff. When using more stringent criteria, restricting to samples with larger numbers of PTCs and requiring that these PTCs are rare in population genomics (to reduce effects of selection), then the correlations of ASE with ETG estimates increase strongly (see below, R was >0.75 in some settings). However this would be accompanied by a substantial decrease in the number of remaining samples, therefore we opted to keep the current setup.

We describe our justification for the tradeoff made, in much detail below. We also show data to support that our ETG and ASE methods are in fact fairly reliable in their estimation of NMD efficiency of individuals because they do correlate strongly under certain scenarios. Additionally, with the current setup, the tissue-level ASE and tissues ETG do correlate at R=0.69 already (in GTex: now shown as Fig. 2F and included below for convenience), implying that tissue-level NMDeff estimates are fairly reliable as they were provided in the original analysis.

Fig. 2. Individual-level and tissue-level quantification of variable NMD efficiency.

[...] **F**, Spearman correlation between tissue-level NMD efficiency rankings based on median ETG iNMDeff and median ASE iNMDeff values per tissue, for the GTex cohort. Tissues are grouped based on cell-of-origin: Nervous system-related tissues (Pan-nervous), Kidney-related tissues (Pan-kidney), Reproductive system tissues (Pan-reproductive), Gastrointestinal tissues (Pan-GI), and those originating from Squamous cells (Pan-squamous).

Next, moving from tissue-level estimates (which pool multiple samples) to individual-level NMDeff estimates, here are more observations about ETG and ASE method details relevant to their agreement:

Of the ETG and ASE methods, we believe that ETG method is likely more accurate -- not only it is based on known set of NMD target genes, previously experimentally demonstrated in multiple studies ("Consensus" set), but we now see that the ETG method is empirically performing very well on the validation analysis with new data from controlled experiment on various *UPF1* k.d. cell lines (see above and new Fig. 1D). The ETG method relies on classifying isoforms based on assumed NMD-triggering features (3'UTR splice site and/or uORF at 5'UTR), and we acknowledge that presence of these features doesn't always guarantee NMD targeting (e.g. various human genes have uORFs). Nonetheless the fact that we focus the analysis to only ~100 genes that are known NMD targets, and select the NMD-triggering feature bearing transcripts only in these genes, should greatly increase discriminatory power for obtaining NMD efficiency readouts.

In contrast, we think the ASE method is likely the less precise one when applied to some individual samples (however it works well in others). This is because it relies on fairly rare events -- PTC variants which might occur only a small number of times in a genome -- which means that the readouts for the few-PTC genomes are noisy. We will see below that in fact the few-PTC genomes are fairly common.

This ASE readout noise due to the small number of mutations per genome can in fact be ameliorated by excluding samples, however there is a tradeoff with coverage and consequently, statistical power. Making the criteria for minimum number of PTCs per genome more stringent also means we will have fewer samples available for the association analysis (with CNAs, or with germline variants etc) downstream. To strike a balance between noise reduction and data retention, the filtering criteria in our original analysis (and as they currently stand) have excluded 47% of TCGA samples and 25% of GTEx samples in ASE analysis, because they contained fewer than 3 PTCs (that met our criteria for sequencing read coverage at the PTC).

Crucially, requiring more PTCs per sample will increase the correlations between ASE and ETG methods substantially. Here, we analyzed the relationship between this threshold and ASE-versus-ETG correlation values, in both TCGA and GTEx datasets. Our current threshold of requiring ≥ 3 PTCs for retaining a sample (maximizing number of retained samples) yielded sample-wise ASE-ETG correlations of 0.19 in GTEx and 0.14 in TCGA in a pan-tissue analysis.

More stringent thresholds for minimum number of PTCs consistently improved correlations of the ASE method with the ETG, with both TCGA and GTEx datasets having ASE-ETG correlations ~ 0.23 at higher thresholds (5). (note: we took care that this analysis, shown in new Supplementary Fig. 5A, and also below for convenience, rigorously controlled for the varying sample sizes across different thresholds, by a subsampling to generate a matched-sample size baseline for each threshold). The plot is also shown below; see the upper panels where population allele frequency (AF) of PTCs threshold was ≤ 0.2 as applied in the original study (lower panels show an alternative AF setting with yet higher correlations, discussed below).

Supp. Fig. S5. Stringent ASE filterings enhance correlations with ETG iNMDeff

A, Pearson correlations (R) between ASE and ETG iNMDeff methods, at individual sample-level, across different filtering criteria in GTex (pan-tissue, left panel) and TCGA (pan-cancer, right panel). Two filtering parameters were varied: i) minimum number of germline PTC variants per sample required for ASE iNMDeff estimation (X -axis), and ii) PTC population allele frequency (AF) threshold (AF \leq 0.2 in upper panels; AF \leq 0.01 for strictly rare variants in bottom panels). Numbers above bars indicate the total number of samples remaining after applying filters. Under current thresholds (≥ 3 PTCs per sample and AF \leq 0.2), 13053 (75%) of GTex samples and 5216 (53%) of TCGA samples are retained. Statistical significance: *** $p < 0.001$, ** $p < 0.01$, * $p < 0.05$, ns = non-significant. **B**, Tissue-specific analysis using the same sample filtering criteria as in (A), showing correlations calculated separately for each normal tissue (GTex) or cancer type (TCGA). We show the median correlation with Q1-Q3 range as whiskers, and the proportion of tissues/cancers displaying positive correlation as barplots. Both results (A and B) demonstrate a trade-off between precision and sample retention.

Raising the threshold to ≥ 6 PTCs per sample, would however have required excluding approximately 65% of GTex samples and 85% of TCGA samples for the ASE method. We felt this choice was not prudent as

it would have severely limited our statistical power for downstream association analyses. Nonetheless it demonstrates that higher ETG-ASE agreement is possible to obtain.

When analyzing ASE-ETG correlations separately by tissue type: In GTex, raising the minimum threshold in the ASE method from 3 to 6 PTC would increase the proportion of tissues showing positive correlations from 75% to 86%, however, again with the trade-off with statistical power in downstream analyses in our study Supp. Fig. S5B and see above plot (panel B, upper subpanels).

Importantly, in addition to this filtering in the ASE method, another filter was also applied to the variants: population allele frequency (AF). In particular, each sample contains a distinct set of germline PTCs (see histograms below). We specifically excluded very common PTCs (i.e. high population AF) due to them being under negative selection, which results in constraints on NMD efficiency (Lindeboom R. et. al. 2019, Young-gon Kim et. al. 2024). The AF threshold that we applied in practice ($AF \leq 0.2$) again represents a compromise, so as to ensure we had enough variants for NMDeff estimation using ASE method. However, it is true that also the many retained variants below AF 0.20 are under negative selection, as the AF threshold is not stringent.

AF <= 0.2:

AF <= 0.01:

Reviewer response Figure R1. Number of PTCs available per sample, with different AF criteria.

When we tested using a more strict definition of rare germline PTCs i.e. more stringently removing effects of selection ($AF \leq 0.01$, instead of $AF \leq 0.2$ as in the original analyses), we had considerably fewer PTC variants available per sample for the ASE calculation (and thus, automatically, fewer samples were available for analysis, because of requiring a certain number of variants per sample). However, the pan-tissue/cancer ASE-ETG correlation on this smaller ASE dataset substantially improved (see the plot above, bottom 2 panels where $AF \leq 0.01$, also provided in new Supplementary Fig S5A, bottom panels): in GTex, the individual-level ASE-ETG correlations ranged from 0.24 to 0.70 (for thresholds 5-10 required PTC variants per sample). In TCGA, correlations ranged from 0.28 to ~0.50 (for thresholds 5-9 required PTC

variants per sample; threshold 10 was not useful for estimating R as only n=3 samples remained). Correlations within each tissue also improved for the AF<0.01 setting (see Supp. Fig. S5B, bottom panels, showing medians and Q1-Q3 of tissues). We note that pushing the ASE-ETG correlations to values of >0.50 for individual-level analysis would have required reductions in the number of available samples (Supplementary Fig. S5), so is not ideal for performing downstream association testing with genetic variants. However this analysis that varies stringency cutoffs does demonstrate that NMD efficiency can be estimated with ASE method accurately in some samples.

In conclusion, with using more restrictive thresholds for variant inclusion (population AF) and for sample inclusion (minimum PTC variants per sample), we would have been able to have substantially better correlations between the ASE and ETG methods, of up to ~0.70. This demonstrates that the ETG method is in fact quite reliable. The ASE method may be “tuned” differently, according to desired outcomes -- achieving coverage vs precision tradeoffs. With current data set sizes, we reasoned that coverage is a better strategy, particularly given that ASE is used largely as a replication method to the ETG analysis, i.e. no conclusions are made on ASE alone but ETG support is required.

We have now mentioned this briefly in the manuscript Results section (“Agreement between the two methods to estimate NMD efficiency”) and Supp. Text S2: “Optimizing sample size and precision in ASE- based NMD efficiency estimation”.

Results:

“Correlations between the ETG and ASE estimates of iNMDeff across individuals are variable increasing (from ~0.2 to ~0.7 on GTEx, see Supp. Fig. S5A-B) if more stringent filtering criteria are applied to the ASE variants, presumably due to noise stemming from low numbers of PTC variants in some samples (see Supp. Text S2)”

Supp. Text S2:

We also described the Methodology of the analysis in detail in the Supplementary Text S2, please refer to the attached manuscript with tracked changes (text omitted here for brevity).

Finally, we would like to re-emphasize that, when comparing aggregated samples to assess differences in NMD efficiency across tissues, then the ETG and the ASE method do correlate very well already at the default (high-coverage) ASE filtering settings: R=0.69 in GTEx (p=7e-9, Fig. 1F) and R=0.49 in TCGA (p=4e-4, Supp. Fig. S6A). The robustness of tissue-level estimates are further bolstered by that our tissue- specific NMD efficiency rankings correlate well with those recently reported by Teran *et. al.* (2021), who used an independent ASE approach (there was no ETG method in that study), and they applied it to the GTEx data only (TCGA not tested). This Teran *et. al.* tissue-wise NMD score correlated R = 0.75 across tissues for our ASE estimates and R = 0.72 for our ETG estimates, Supp. Fig. S8A-B, supporting our findings on tissue differences in NMD efficiency, and is briefly mentioned in the Results manuscript (section “Agreement between the two methods to estimate NMD efficiency”):

“When pooling samples by tissue, we observe the correlation between ETG and ASE methods of R=0.69 (GTEx, Fig. 2F) and R=0.49 (TCGA, Supp. Fig. S6A), supporting the concordance between the two NMD efficiency estimation methods, and suggesting systematic differences between tissues, which will be more rigorously tested below.”

(3) In GTEx, the sample sizes of brain tissues are generally smaller than those of other tissues. This discrepancy could confound the authors' observation that NMD efficiency is lower in the nervous system.

We agree that brain subregions in GTEx have smaller sample sizes compared to other tissues, though the difference is not extreme (brain tissues $n=139-255$, mean = 203; non-brain tissues $n=4-802$, mean = 359). Additionally, our estimates of NMD efficiency should not be biased by sample size. To address this empirically, we conducted a reanalysis by subsampling all GTEx tissues to match the sample size of the brain subregion with the smallest number of samples (BRNSNG, $n=139$) and discarding the additional samples (chosen randomly). We maintained the original sample sizes only for tissues that already had fewer samples than this threshold (KDNMDL, CVXECT, FLLPNT, CVSEND, BLDDER, KDNCTX).

Using these subsampled datasets, we performed our Tissue iNMDeff Deviation (TND) analysis to identify tissues with significantly higher or lower iNMDeff compared to a baseline, using a randomization test (as in original analysis; see Methods). The TND metric indicates higher tissue-specific NMD efficiency when positive and lower efficiency when negative. Importantly, the results from this subsampled analysis, although there were some differences in some particular tissues, were generally consistent with our original findings: both brain and reproductive system tissues maintained significantly lower iNMDeff compared to other tissues, and this pattern was robust across both ETG and ASE iNMDeff measurements (brain tissues: original median FDR = $3.3e-2$; subsampling median FDR = $2.7e-2$, across both iNMDeff methods) (see plot below, also provided as new panels in Supp. Fig. 10). Individual brain regions showed comparable FDR values between analyses, see some examples for ETG iNMDeff: BRNACC (original: FDR = 0.15 vs subsampling FDR = 0.29), BRNAMY (FDR = $9.4e-4$ vs $6.8e-3$), BRNCHA (FDR = $9.4e-4$ vs $1.4e-3$), BRNCHB (FDR = $9.4e-4$ vs $1.4e-3$), BRNHPT (FDR = 0.22 vs 0.33). Lastly, there is a similar proportion of brain tissues showing significant differences (FDR < 0.05) in both analyses: 8/13 brain tissues in the subsampling analysis compared to 7/13 in the original analysis.

These results have been briefly mentioned in the Results (section: “Lower NMD efficiency in the tissues of the nervous system”):

“Notably, the Pan-nervous group of tissues exhibited lower observed iNMDeff (Supp. Fig. S7A-B) compared to randomized expectations (Fig. 3C-D), and this result remained robust when equalizing the number of samples across tissues (Supp. Fig. S10)”

Supp. Fig. S10. GTex intra-tissue variability of NMD efficiency after subsampling

Tissue iNMDeff Deviation (TND) analysis performed on subsampled tissues as in the original (Supp. Fig. S9), in order to control for sample size differences between brain subregions ($n=139-255$, mean=203) and the rest of tissues ($n=4-802$, mean=359). Results showed a consistent proportion of brain tissues with significant differences ($FDR < 0.05$) between subsampled (8/13 brain tissues) and original (7/13) analyses (Supp. Fig. S9). The subsampling was done per tissue as to match the sample size of the smallest brain subregion (BRNSNG, $n=139$), except for tissues already below this threshold (KDNMDL, CVXECT, FLLPNT, CVSEND, BLDDER, KDNCTX).

(4) Why the replicated association between the CNA-PCs and NMD efficiency at the pan-cancer level is usually not observed at the cancer type level?

In the original study, we analyzed pan-cancer copy number alteration (CNA) patterns using sparse principal component analysis, which identified 86 CNA principal component signatures (CNA-PCs), which commonly correspond to large-scale segmental CNA (Supp. Fig. S14B-C). Analysis of associations between these CNA-PCs and iNMDeff at both pan-cancer and cancer-type levels revealed 6 pan-cancer and 3 cancer-type specific associations (with CNA-PC signatures in UCEC and PCPG cancers) that were significantly associated with iNMDeff ($FDR < 10\%$) in ASE and that were also significant in ETG. To achieve replication, we further retained cases where direction of effect was consistent between ETG and ASE iNMDeff methods, and we thus identified 3 robust associations with iNMDeff (all pan-cancer; the cancer type specific

associations were discarded due to this criterion): CNA-PCs 3, 52, and 86, those mentioned in the manuscript (Fig. 4A). To briefly recap: the CNA-PCs 3 and 86 corresponds to 1q gains, with a prominent focus in the 1q21.1-23.1 region (especially for CNA-PC86), and is associated with a lowered iNMDeff; the CNA-PC52 exhibited CNA gain peaks at chromosome 2p and 2q (2q31.1-2q36.3) and is associated with a reduced iNMDeff. These 3 associations, statistically significant in the pan-cancer analysis, also demonstrated broadly consistent directions across multiple cancer types sometimes with effect sizes exceeding the pan-cancer effect size (Supp. Fig. S15, also provided below).

Supp. Fig. S15. Associations between replicated pan-cancer CNA-PCs and iNMDeff in TCGA

Cancer-type and pan-cancer associations between three replicated pan-cancer CNA-PCs and iNMDeff, employing both ETG (left panels) and ASE (right panels) methodologies, under a 10% FDR threshold. The implicated CNA-PCs are CNA-PC 3, CNA-PC 52, and CNA-PC 86, each demonstrating a significant link with iNMDeff. The effect size of the associations (beta coefficient from the linear model) means that higher values of a given CNA-PC signature correlate with increased iNMDeff, and vice versa.

We have clarified this in the revised manuscript in Results (section: “Association analysis of somatic pan-cancer CNA signatures”):

“Utilizing ASE method for discovery and ETG for validation, we identified 3 pan-cancer CNA-PCs significantly associated with iNMDeff at FDR < 10% replicating in both methods with the same direction of NMD effect (Fig. 4A): CNA-PC 3 (ASE $p = 1.3e-2$; ETG FDR = $1.2e-37$), CNA-PC 52 ($2.2e-2$; $3.1e-9$), and CNA-PC 86 ($8.7e-2$; $8.6e-37$).”

→

“Utilizing ASE method for discovery and ETG for validation, we identified 6 pan-cancer and 3 cancer-type (UCEC and PCPG) CNA-PCs significantly associated with iNMDeff at FDR < 10%. We further retained cases where direction of effect was consistent between ETG and ASE iNMDeff methods, and we thus identified 3 robust pan-cancer associations with iNMDeff (Fig. 4A): CNA-PC 3 (ASE $p = 1.3e-2$; ETG FDR = $1.2e-37$), CNA-PC 52 ($2.2e-2$; $3.1e-9$), and CNA-PC 86 ($8.7e-2$; $8.6e-37$).”

As for why we did not observe more cancer-type significant hits, the individual cancer type analyses have substantially smaller sample sizes compared to pan-cancer, therefore power is more limited. For instance, breast cancer has ~1000 samples, which is roughly 10 times smaller than the pan-cancer dataset. Further, the required exclusion of ~50% of samples (in the whole dataset) due to stringent ASE iNMDeff filtering criteria (see above) further limited statistical power for tissue-specific association analysis.

We note there are additional associations of CNA-PCs with iNMDeff that are significant with ASE in individual cancer types but do not reach significance in ETG analyses (FDR < 10%), and we discarded them out of an excess of caution. We chose to focus only on the 3 robust, replicating results, which are all pan-cancer.

We have included a mention of this in the discussion (new text in blue):

“[...] We next extended the somatic variant analysis to CNAs, by a bespoke sparse-PCA-based method to control for genetic linkage in association studies, here applied to NMD efficiency variability. Limited cancer-type specific findings can be attributed to smaller sample sizes in individual cancer analyses and broad distribution of CNA signatures across cancer types, suggesting shared rather than cancer-specific mechanisms. Three CNA-PCs reflecting large-scale gains were robustly associated with reduced NMD activity, most strongly CNA-PC3 and 86 located in regions 1q and 1q21-23.1. The 1q gain is found in ~25% of cancers and it has been proposed to be selected through increasing dosage of the MDM4 oncogene phenocopying TP53 mutation⁶³, or of MCL-1 antiapoptotic factor⁹⁶ or via upregulation of Notch genes⁹⁷. [...]”

→

*“[...] We next extended the somatic variant analysis to CNAs, by a bespoke sparse-PCA-based method to control for genetic linkage in association studies, here applied to NMD efficiency variability. **Cancer-type specific significant findings were limited, which can be attributed to smaller sample sizes in individual cancer type analyses, thus our analyses focussed on shared rather than cancer-specific mechanisms.** Three CNA-PCs reflecting large-scale gains were robustly associated with reduced NMD activity, most strongly CNA-PC3 and 86 located in regions 1q and 1q21-23.1. The 1q gain is found in ~25% of cancers and it has been proposed to be selected through increasing dosage of the MDM4 oncogene phenocopying TP53 mutation⁶⁶, or of MCL-1 antiapoptotic factor⁹⁹ or via upregulation of Notch genes¹⁰⁰. [...]”*

Once again, we thank the reviewer for their thoughtful critique on analytical points (1)-(4) above, which we have hopefully addressed well with various additional data.

Next, below we provide responses to their various minor comments on the text:

(5) Some figures and legends in the paper are unclear or seem inconsistent with the data. Such inconsistency could lead to misunderstandings. For example, the legend for Fig. 4A mentions six CNA-PCs, though only three of these are actually replicated.

We acknowledge and apologize for the inconsistency in the figure legends, particularly regarding Fig. 4A description of CNA-PCs. To clarify: while we initially identified 6 significant pan-cancer CNA-PCs associated with ASE iNMDeff that were also significant using ETG iNMDeff, we ultimately identified 3 of them (CNA-PC 11, 27, and 71) as replicated after requirements on effect size across the two methods, see answer (4)

from above. We have now thoroughly revised the entire manuscript to ensure accurate representation of the data and clear communication of our analytical decisions. See the following examples:

-Fig. 4A: we have removed the 3 non-replicated CNA-PCs from the plot

-The classification of candidate genes and NMD-related genes from our CNA-PCs analysis was not consistent between some of the plots, including the number of candidates. We have changed the classification to “Candidates”, “Candidates NMD” (NMD-related genes that are potential candidates because they are located in the peak CNA gains), “NMD-related” (NMD-related genes that are not candidates) and “Controls” (genes from outside the peak CNA gains), and modified the text, figures and legends accordingly. Please see our answer to question (7) below for explanation of these categories. See Fig. 4D-F and Supp. Fig. S20 and S22 as examples.

(6) In Fig. 4E, there is no statistically significant difference in NMD efficiency between the high and mild groups for CNA-PC3, which further highlights the unreliability of their NMD efficiency measurements.

We think that the lack of statistical significance between "High" and "Mid" groups for CNA-PC3 in Fig. 4E (now Fig. 4C) actually supports rather than undermines our findings. Let us clarify this important point.

The "Mid" group by CNA-PC3 represents a focal Chr 1q gain strongly enriched in the 1q21-23.1 region (as shown in Fig. 4B), which contains our candidate genes hypothesized to reduce iNMDeff. The "High" group by CNA-PC3 represents a larger segmental Chr 1q gain, including this segment 1q21-23.1 but also the rest of the chromosome arm.

The confusion arises because the samples groups are labelled “Mid” and “High” by the CNA-PC3 score level, which captures both the extent (breadth) of the CNA gain and the level of the gain (samples with “Mid” scores have only slightly lower gene dosage than the “High” scores in the 1q21-23.1 region; instead, the main difference is breadth, where the “Mid” has smaller breadth and the signal is more focussed on 1q21-23.1 and then drops off quickly, while “High” is more widespread across 1q -- please see Fig 4B, also provided below for convenience).

Fig. 4. Somatic pan-cancer CNA signatures are associated with NMD efficiency.

[...] **B**, CNA-PC3 reflects gene amplifications across chromosome 1, ordered by genome location, plotted along the X-axis, with amplifications assessed by averaging GISTIC CNA scores for each gene across TCGA participants (Y-axis). Samples are categorized into bins based on scores from pan-cancer CNA-PC3 signature: 'High' showing complete chromosome 1q arm-level gain (gene dosage scores ~1), 'Mid' with more localized gain in the 1q21-23.1 region (gene dosage ~0.8, with remaining chromosome at ~0.4), and 'Low' showing no notable alterations in chromosome 1q (gene dosage scores ~0). [...]

Therefore, the similar iNMDeff levels between "Mid" and "High" groups (ETG: $p=0.76$; ASE: $p=0.7$) is expected and consistent with our hypothesis that specifically the 1q21-23.1 region gain (captured in both "Mid" and "High"), rather than gains in other 1q parts (captured mostly in "High") drives reduced NMD efficiency. We have added a brief clarifying note to the Fig 4B (see above) and Supp Fig 12 legend (see below), stating that "Mid" and "High" on CNA-PC3 only partly reflect gene dosage in the 1q21-23.1 segment (new text in blue):

"B, CNA-PC3 reflects gene amplifications across chromosome 1, ordered by genome location, plotted along the X-axis, with amplifications assessed by averaging GISTIC CNA scores for each gene across TCGA participants (Y-axis). Samples are categorized into three bins based on scores from pan-cancer CNA-PC3 signature: 'High' showing complete chromosome 1q arm-level gain (gene dosage scores ~1), 'Mid' with more localized gain in the 1q21-23.1 region (gene dosage ~0.8, with remaining chromosome at ~0.4), and 'Low' showing no notable alterations in chromosome 1q (gene dosage scores ~0).

The more informative comparison is against the "Low" group, which represents samples with no substantial copy number gains. These comparisons reveal significant differences between "High" and "Low" groups for both ETG iNMDeff ($p<2.22e-16$, Fig. 4E) and ASE iNMDeff ($p=4.8e-2$, Supp. Fig. 16B-C).

Furthermore, we validated these findings using an independent dataset of 1450 human cell lines from the Cancer Cell Line Encyclopedia (CCLE). This not only confirmed our TCGA findings with 1q gains, but also revealed an opposing trend in cell lines with 1q CNA deletions, which showed increased cell line NMD efficiency (cNMDeff) compared to the decreased efficiency observed with 1q amplifications (see Fig. 4C for TCGA-based ETG model and Supp. Fig. 17B for GTex-based model).

(7) In Fig. 4F, there are only six, not 30 candidate genes as mentioned in the figure legend and main texts.

To recap for context, our 30 candidate genes are those that passed our thresholds: correlation between gene expression and ETG iNMDeff; and correlation between gene expression and CNA. From these, 20 genes were identified in the 1q21.1-23.1 region (Fig. 4C-D, now D-E) –now called "Candidates"– including 2 NMD-related genes (*SMG5* and *INTS3*) –now called "Candidates NMD"– as potential causal genes of iNMDeff observed via CNA gain at 1q (Fig. 4C and E, now E and C). To further prioritize these candidates, we analyzed CRISPR KO data from CCLE (DepMap), hypothesizing that if these genes influence NMD efficiency, they should have genetic interactions; these are ascertained if a gene's knockout fitness effects should correlate with the fitness effects of NMD factors like *UPF1*, *UPF2*, and *SMG1* (total $n=21$ NMD-related). This "gene co-dependency" principle has been successfully applied to infer functional links from large-scale CRISPR screening data (see doi: 10.15252/msb.202311657).

To arrive at the 6 candidate genes shown (Fig. 4F) rather than the full original set of 20, we first performed a clustering of CRISPR score profiles (fitness across CCLE cell lines) containing all candidate genes (18 “Candidates”, 2 “Candidates NMD” plus 2 new ones: *SF3B4* and *RBM8A*, despite not meeting ETG iNMDeff thresholds, were included as an additional “Candidates NMD” via manual curation based on their related functions in NMD –spliceosome and EJC, respectively– and their location within 1q21.1-23.1), along with 21 known NMD-related genes. Additionally we included 18 random negative control genes located on chr 1q but outside the 1q21.1-23.1 region. Note that *KIAA0907* was excluded from the analysis due to lack of CRISPR data.

After observing that 6 of these candidate genes clustered with the known NMD-related genes, we created a simplified visualization (Fig. 4F) showing just these 6 genes along with the 21 known NMD-related genes. For transparency, we have included the complete clustering heatmap of CRISPR profiles as Supplementary Figure S20B; this includes all 20 candidates (plus *SF3B4* and *RBM8A*) genes, 21 known NMD-related genes, and 18 negative control genes.

We apologize for the confusion and have clarified this in figure legends (Fig. 4, Supp. Fig. S20 and S22) and in the Results text (section: “Identifying NMD-associated genes via focal CNA analysis”); please see text with tracked-changes to check the edits made, which were omitted here for brevity.

(8) The manuscript is quite lengthy, which can detract from the key findings. To improve clarity, the authors could consider moving less critical sections and methods to supplementary materials, focusing on the most significant findings in the main text. This would make the paper more accessible and ensure that the main conclusions are clearly highlighted.

We appreciate this suggestion and we agree with the reviewer. The manuscript has been streamlined by relocating several side findings to the new Supplementary Texts S1-S6, while leaving a brief mention of each in the main text. We think this restructuring will help readers focus on the key findings of our study. See the following changes:

Result section “Robustness of the individual NMD efficiency estimates”, some parts were moved as Supp. Text. S1 as “Additional support for robustness of the NMD efficiency measures”.

Based on our answer to question (2) above, we included the analysis in Supp. Text S2 as “Optimizing sample size and precision in ASE-based NMD efficiency estimation”.

Result section “Inter-individual NMD efficiency variability” was shortened and moved to Supp. Text S3, which included an extended version with more tissues analyzed.

Result section “Association analysis of somatic mutations with iNMDeff” was moved to Supp. Text S4.

Result section “Somatic chromosome 2q gain also associates with reduced NMD efficiency” moved to Supp. Text S5.

Result section “NMD efficiency modulates selection of somatic nonsense mutations”, moved as Supp. Text. S6.

Additionally we have changed Figures in interest of brevity and to keep the referencing order correctly:

-Merged Figure 6 and 7. Figure 6 panel B was moved to Supp. Fig. S26.

-Figure 1 D and F panels are new. Current C panel was shortened. Current E was previous C. Three panels were moved to Supplementary figures S2 and S4

-Figure 3 panels were rearranged to follow referencing order.

-Reordered and improved supplementary figures: Supp. Fig. S4, S7, S11, S14, S16, S18, S20, S26

-New Supp Figures (either new data or split old Supp Figures): Supp Fig. S3, S5, S6, S8, S10, S19, S21, S22, S24

Second round of review

Reviewer 2

The authors have made a satisfactory revision.

The addition of data from cell lines depleted for UPF1 and PRO-seq data strongly supports the methodology.

I agree that the original manuscript was too lengthy, so it's good that the revised version is shorter.

Publication recommended.

Minor comments:

Please explain cancer-related abbreviations when first mentioned in the text. Examples:

p. 8, l. 12: COAD_MSI

p. 14, l. 15: LUAD and BRCA

p. 16, l. 43: LUSC and OV

p. 17, l. 10-11: The reference should be to Fig. 4D, not Fig. 4C.

Fig. 4D: The legend at the bottom right ("Other, Candidates, ...") is not consistent with what appears in the plot.

p. 17, l. 16-30: The reference should be to Fig. 4E, not Fig. 4D.

Figure 5D is not referenced in the text.

Fig. 6A: Please include at least a comment on the apparent positive selection of genes with NMD-evading mutations in samples with low iNMDeff.

p. 24, l. 10: "above-mentioned iNMDeff proxy model" — this term is not explicitly used earlier, making it difficult to identify what the authors are referring to.